# Origins of chromosome instability unveiled by coupled imaging and genomics

Marco Raffaele Cosenza[1], Alice Gaiatto[1], Büşra Erarslan Uysal[1,2,3,4,5], Álvaro Andrades[1], Nina Luisa Sautter[1], Marina Simunovic[1], Michael Adrian Jendrusch[1], Sonia Zumalave[6], Tobias Rausch[1,7], Aliaksandr Halavatyi[8], Eva-Maria Geissen[9], Joshua Lucas Eigenmann[1], Thomas Weber[1,9], Patrick Hasenfeld[1], Eva Benito[1], Catherine Stober[1], Isidro Cortes-Ciriano[6,10], Andreas E. Kulozik[2,3,4,5], Rainer Pepperkok[8,11] & Jan O. Korbel[1,2,6,12 ✉]

Somatic chromosome instability results in widespread structural and numerical chromosomal abnormalities (CAs) during cancer evolution[1–3]. Although CAs have been linked to mitotic errors resulting in the emergence of nuclear atypia[4–7], the underlying processes and rates of spontaneous CA formation in human cells are underexplored. Here we introduce machine-learning-assisted genomics and imaging convergence (MAGIC)—an autonomously operated platform that integrates live-cell imaging of micronucleated cells, machine learning on-the-fly and single-cell genomics to systematically investigate CA formation. Applying MAGIC to near-diploid, non-transformed cell lines, we track de novo CAs over successive cell cycles, highlighting the common role of dicentric chromosomes as initiating events. We determine the baseline CA mutation rate, which approximately doubles in *TP53*-deficient cells, and observe that chromosome losses arise more frequently than gains. The targeted induction of DNA double-strand breaks along chromosome arms triggers distinct CA processes, revealing stable isochromosomes, coordinated segregation and amplification of isoacentric segments in multiples of two, as well as complex CA outcomes, influenced by the chromosomal break location. Our data contrast de novo CA spectra from somatic mutational landscapes after selection occurred. The experimentation enabled by MAGIC advances the dissection of DNA rearrangement processes, shedding light on fundamental determinants of chromosomal instability.

Cancer genomes are shaped profoundly by somatic chromosomal abnormalities (CAs)[1–3]. According to pan-cancer studies, CA driver events outnumber base substitution drivers in cancer genomes, and the cumulative burden of CAs is linked strongly to adverse clinical outcomes[2,3,8,9]. Recent studies have shed light on patterns and classes of CAs present in cancer genomes[10–13]. However, unlike for base substitutions[10,14], specific contributions of CA formation processes to the mutational spectrum in cancer, as well as the baseline rate at which CAs emerge, are poorly understood. Consequently, our understanding of the role of CA formation in driving karyotype evolution in cancer remains incomplete.

Recent reports have established that a single DNA lesion can trigger a cascade of alterations, resulting in chromosomal instability and promoting complex CA formation processes[5,6,15]. Mitotic errors serve as intermediate steps for these cascades[4–7], resulting in nuclear atypia such as micronuclei and chromatin strings[2,16]. Live-cell microscopy combined with (semi)-manual cell selection and single-cell sequencing, has causally linked nuclear atypia to the formation of complex CAs, and shed light on mechanisms underlying chromothripsis[4–6,15,17,18]. Yet, owing to their labour-intensive nature, only a limited number of single-cell genomes have been investigated, leaving important gaps in our understanding of chromosomal instability processes linked to aberrant mitoses.

To address these limitations, we devised a platform that couples autonomous confocal microscopy in live cells, machine learning for on-the-fly assessment of nuclear atypia, targeted cell photolabelling and cell sorting. Through automated imaging-based cell selection, target cells are isolated precisely from a heterogeneous cell population. The isolated cells are then subjected to single-cell sequencing and systematic phenotype analyses, thus enabling investigation of the cellular context, mutation rates and triggers of spontaneous CA formation in cell line models that mimic particularly early steps in tumour evolution.

## Investigating de novo CA formation with MAGIC
### Mitotic error profiles in non-transformed cells
To investigate de novo CAs arising in a human cell, we devised MAGIC—a platform coupling automated microscopy with targeted photolabelling

[1]Genome Biology Unit, European Molecular Biology Laboratory (EMBL), Heidelberg, Germany. [2]Molecular Medicine Partnership Unit (MMPU), EMBL, University of Heidelberg, Heidelberg, Germany. [3]Department of Pediatric Oncology, Hematology, and Immunology, University of Heidelberg, Heidelberg, Germany. [4]CCU Pediatric Leukemia, German Cancer Research Center (DKFZ), Heidelberg, Germany. [5]Hopp Children's Cancer Center Heidelberg, Heidelberg, Germany. [6]European Bioinformatics Institute (EMBL-EBI), Hinxton, UK. [7]Genomics Core Facility, EMBL, Heidelberg, Germany. [8]Advanced Light Microscopy Core Facility, EMBL, Heidelberg, Germany. [9]Data Science Centre, EMBL, Heidelberg, Germany. [10]Cancer, Ageing and Somatic Mutation Programme, Wellcome Sanger Institute, Hinxton, UK. [11]Cell Biology and Biophysics Unit, EMBL, Heidelberg, Germany. [12]Bridging Research Division on Mechanisms of Genomic Variation and Data Science, German Cancer Research Center (DKFZ), Heidelberg, Germany. ✉e-mail: Jan.Korbel@embl.de

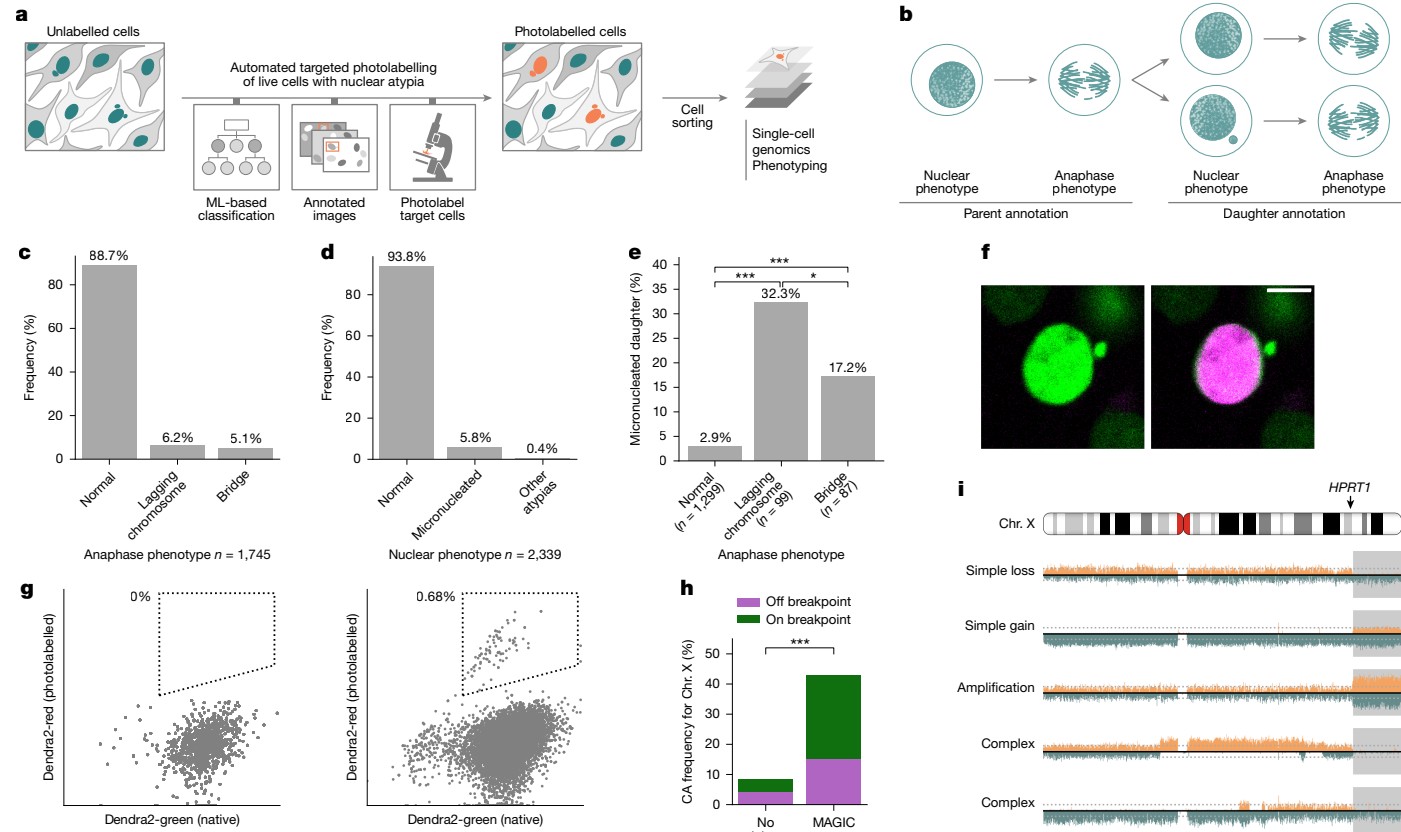

**Fig. 1 | MAGIC enables characterization of de novo CAs at scale. a**, Overview of MAGIC. Left to right, automated microscope analyses of a cell population with a heterogeneous morphology, performing targeted photolabelling using a machine-learning-based (ML-based) classifier that outputs a micronuclei detection probability for each cell, identification of areas for targeted photolabelling and automatic microscope laser photolabelling of target cells. The steps are repeated to reach a desired yield. Photolabelled cells are then isolated for further analysis. **b**, Annotation strategy for long-term live-cell imaging of MCF10A H2B-Dendra2 cells. Anaphase and nuclear phenotypes were assessed over two generations, to observe at least one cell cycle; first generation cells are referred to as parents, and second generation cells as daughters. A total of 821 genealogies were annotated, 765 of which generated progeny. **c**,**d**, Overall frequency of anaphase (**c**) and nuclear (**d**) phenotypes

over all genealogies; 1,750 mitoses and 1,795 nuclei were annotated, respectively. **e**, Frequency of daughter micronucleation associated with different anaphase phenotypes (Fisher's exact test). **f**, MCF10A cells constitutively expressing H2B-Dendra2 before in its native state (left, green) and after photolabelling (right, magenta). **g**, FACS profile of control (left) and photolabelled (right) cell conditions. Native (green) versus photolabelled (red) fluorescence is plotted. The sorting gate is represented by a dashed line. **h**, Frequency of Chr. X CAs after targeted DSB induction in randomly sampled single cells, or in cells selected using MAGIC (Fisher's exact test). On-breakpoint CAs (green bar) refer to the *HPRT1* locus. **i**, Examples of CAs involving a breakpoint at the *HPRT1* locus. Grey shading, acentric fragment created through targeted DSBs (*$P < 0.05$, **$P < 0.01$, ***$P < 0.001$; Methods). Scale bar, 10 μm.

and single-cell genomics to gain insights into CA formation from studying nuclear atypia (Fig. 1a; Methods). To investigate CA formation landscapes during an initial stage of tumorigenesis, we selected two non-transformed cell lines maintaining a relatively stable karyotype[4,5,19,20]: MCF10A cells, derived from normal breast tissue and spontaneously immortalized; and hTERT RPE-1 cells (RPE-1) of retinal pigment epithelial origin. During mitosis, both cell lines occasionally form micronuclei[21–23], the collapse of which can result in complex CAs, including chromothripsis[2,4,6].

To generate pilot data for setting up MAGIC, we manually annotated nuclear and mitotic phenotypes across two generations in MCF10A cells (Fig. 1b). We find that spontaneously arising anaphase bridges and lagging chromosomes are the predominant type of mitotic error, occurring in 5.1% and 6.2% of all mitoses, respectively (Fig. 1c). During interphase, 6.2% of cells have nuclear atypia, with micronuclei (5.8%) being by far the most common type (Fig. 1d). We find that mitotic errors result in the formation of at least one micronucleated daughter cell in 32.3% and 17.2% of cases for lagging chromosomes and chromatin bridges, respectively, demonstrating that both types of mitotic error converge on micronucleation (Fisher's exact test, Fig. 1e). Furthermore, micronucleated cells are around 9.5 times more likely to generate a

micronucleated daughter cell, compared with cells with normal nuclei (Supplementary Fig. 1a). Accordingly, we detect widespread anaphase defects in daughter cells originating from abnormal anaphases (Supplementary Fig. 1b). This 'self-propagating' nature of micronucleated cells implies that mitotic errors can result in nuclear atypia formation over consecutive cell cycles, which could trigger episodic chromosomal instability.

We examined micronucleated MCF10A cells also with respect to their propensity to generate viable daughters. Micronucleated cells exhibit a significantly longer cell cycle duration, and a notably delayed cell cycle compared with normal cells (Supplementary Fig. 1c–e). Irrespectively, relevant subsets of micronucleated cells continue dividing, and some eventually regain normal nuclear morphology, facilitating the automated isolation of viable cells subject to de novo CAs.

## Machine-learning-enabled adaptive feedback loop

MAGIC systematically selects micronucleated cells using adaptive feedback microscopy ('smart microscopy'), driven by a computational loop integrating machine learning, image analysis and photolabelling (Fig. 1a and Supplementary Fig. 2a). In brief, a confocal image is acquired and examined on-the-fly by machine learning; if a cell of

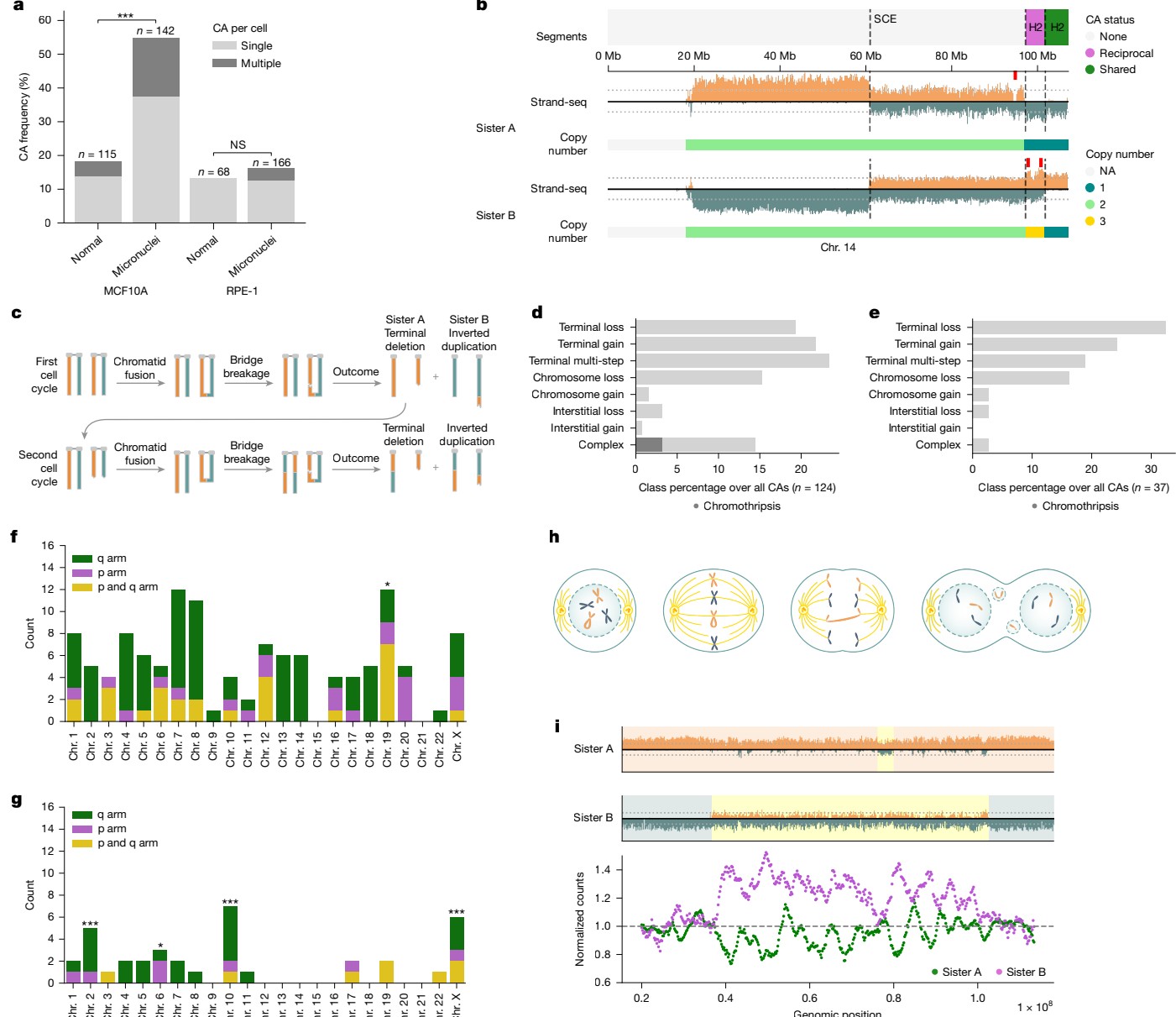

**Fig. 2 | CA landscape of spontaneous micronucleated cells in near-diploid human cell lines. a**, CAs detected per cell (*P* values on the basis of Fisher's exact test). NS, not significant. **b**, Reciprocal CAs between sister cells. Segment annotation: dashed lines indicate an SCE, and demarcate shared and reciprocal segments. Red ticks mark points of inferred complex CA formation. NA, not available. **c**, Mitotic history reconstruction for reciprocal CAs in **b**. Orange and teal colours correspond to Watson and Crick template-strand orientations, respectively. **d**,**e**, Breakdown of CA classes in micronucleated MCF10A (**d**) and RPE-1 (**e**) cells, class percentage over all CAs. See Methods for CA classification criteria. **f**,**g**, CA count per chromosome in micronucleated MCF10A (**f**) and RPE-1 (**g**) cells (binomial testing was used to identify enrichments). In RPE-1, the highest CA counts are seen for Chr. 10 and Chr. X engaging in an unbalanced der(X) t(X;10) translocation—a derivative chromosome that could be particularly susceptible to inclusion in micronuclei—in this cell line[29]. **h**, Scheme showing mitosis with a dicentric chromosome. Sister chromatid fusion generates a dicentric. During anaphase, the dicentric chromosome forms a bridge, which can rupture and potentially form micronuclei in daughter cells. **i**, Chromothripsis affecting Chr. 13, with reciprocal segment inheritance into sister cells seen for a large fraction of the affected homologue. Top, Strand-seq data. Bottom, smoothened, normalized read counts along chromosomal positions.

interest is identified, information on its location is used to photolabel its nucleus automatically with the microscope laser. Then, the next image is assessed, re-initiating the adaptive feedback loop. MAGIC operates autonomously for up to 24 h, examining tens of thousands of cells to photolabel hundreds of live cells exhibiting the desired nuclear morphology for downstream investigation.

We trained machine-learning-classifiers for micronuclei using manually annotated images. We used an extreme gradient boosting-based machine learning framework (XGBoost; Methods) for its model explainability, streamlined implementation, and the relatively few training examples it requires. The whole classification pipeline achieves a precision of at least 90% and a recall of 50%, offering an acceptable balance between specificity and sensitivity (Supplementary Fig. 2b and Supplementary Methods).

## Photolabelling dyes and cell sorting

Photolabelling leverages fluorescent markers with unique characteristics. The expression of the Dendra2 protein allows tagging cells of interest through the stable transition of emitted fluorescence from green to red[7,24], upon gentle and targeted illumination with a 405 nm laser. We

engineered MCF10A and RPE-1 cells to stably express H2B-Dendra2, facilitating nuclear morphology visualization and allowing photolabelling (Fig. 1f). The photolabelling efficiency increases with the illumination up to a maximum dependent on the amount of H2B-Dendra2 expression (Supplementary Fig. 2c,d). We fine-tuned photolabelling conditions to achieve a red fluorescence increase of roughly 22-fold after illumination, without any detectable phototoxicity (Supplementary Fig. 2e,f). As an alternative dye, we synthesized DACT-1 (Supplementary Methods)—a small molecule used for cell tracking[25]—allowing for photo-activation under similar conditions (Supplementary Fig. 2g) while bypassing the need for genetic manipulation. Using fluorescence-activated cell sorting (FACS), we observe distinct populations that represent photolabelled cells with either dye (Fig. 1g and Supplementary Fig. 2h), confirming that target cells are sorted efficiently.

## Targeted CRISPR–Cas9 manipulation reveals de novo CAs

Having demonstrated the effectiveness of MAGIC in cell sorting, we next explored its utility for identifying de novo CAs. To verify experimentally that discovered CAs originate from chromosomes entrapped in micronuclei, we generated DNA double-strand breaks (DSBs) at the *HPRT1* locus on Chr. X, by applying CRISPR–Cas9 in MCF10A cells (Methods). Cas9-mediated DSBs have been reported previously to result in acentric fragments incorporated into micronuclei[26] (Supplementary Fig. 3a). Quantifying nuclear defects upon targeted DSB generation, we find that micronucleation rises by approximately 4.8-fold, indicating an increase in CA formation.

To enable the discovery of de novo CAs resulting in copy-number imbalances, we coupled MAGIC with single-cell template-strand sequencing (Strand-seq)[27]. We performed single-cell genomic sequencing on 85 and 93 cells, respectively, with and without ('control') automated selection for micronucleated cells. We identified CAs using the strandtools algorithm (Methods). We observe a strong increase in CAs in the selected cell fraction (Supplementary Fig. 3b), with 37 (45%) of micronucleated cells showing at least one CA on the X chromosome, corresponding to a fivefold enrichment over the control. Moreover, 24 of 37 (64.9%) of the CAs in the enriched sample contain a breakpoint at the *HPRT1* locus (Fig. 1h), consistent with CAs arising directly at the cut site.

Notably, we find that the cut site gives rise to a diversity of CA classes (Fig. 1i and Supplementary Fig. 3c). Within the micronucleated cells, 18 of 37 (48.6%) CAs show an isolated loss or gain of the cut fragment, indicating abnormal segregation of the acentric fragment. Moreover, 12 CAs comprise terminal deletions from 18 Mb to 70 Mb in size, and 6 are terminal duplications ranging from 22 Mb to 76 Mb in size. We also find evidence for complex CAs in nine cases, all mapping to a single homologue as resolved by haplotype analysis[27], and observe amplifications of the cut acentric fragment in a further nine cases (Supplementary Fig. 3c–e). These data demonstrate the capability of MAGIC to selectively isolate cells undergoing de novo CAs.

## Verification of de novo CAs from sister cell pairs

Reciprocal template-strand inheritance and sister chromatid-exchange (SCE) events[27] from Strand-seq offer uniquely identifiable records of sister cell relationship (Supplementary Fig. 3f). Harnessing these records, we devised an approach to confidently identify sister cell pairs directly from single-cell sequencing data (Supplementary Methods). Using this approach, we find two sister cell pairs in the enriched sample. In one of these pairs, we observe directly a de novo CA. Analysis of this pair reveals a cut fragment inherited asymmetrically in the sisters, thus verifying CA formation (Supplementary Fig. 3g).

## Spontaneous CA formation landscapes

### CA formation in spontaneously micronucleated cells

Whereas biochemically or genetically induced micronuclei have been used to study CA formation[4–6,15], how spontaneously arising micronuclei

may trigger distinct classes of CAs is underexplored. Addressing this gap, we used MAGIC to isolate spontaneously micronucleated cells and investigate CAs landscapes. We first focused on MCF10A, sequencing 142 single-cell genomes from micronucleated cells. Single-cell genome analysis (Methods) revealed 124 CAs, with 54.9% micronucleated cells exhibiting at least one CA (Fig. 2a). When compared with 115 cells with a normal nucleus ('control'), we observe a threefold enrichment in CAs ($P = 1.75 \times 10^{-9}$; Fisher's exact test), indicating widespread CA formation in micronucleated MCF10A cells. We also investigated RPE-1 cells, sequencing the genomes of 166 micronucleated cells and 68 controls. Unlike for MCF10A, we find a non-significant enrichment of CAs in micronucleated cells ($P = 0.69$; Fig. 2a), indicating that RPE-1 exhibits a more stable karyotype.

## Reconstructing de novo CAs over consecutive cell cycles

To further corroborate CA formation, we focused on the sister cell pairs found among 142 micronucleated MCF10A cells (Extended Data Fig. 1a). Out of 12 sister pairs, 7 (58%) show reciprocal CA segregation, consistent with de novo CAs (Extended Data Fig. 1b). We identify three sister pairs with shared CAs; in two cases, these are also accompanied by reciprocal CAs, indicating CA formation across several recent divisions (Fig. 2b and Extended Data Fig. 1b). By contrast, among ten sister pairs of micronucleated RPE-1 cells, only two (20%) show reciprocal CAs and one pair shares a common CA (Extended Data Fig. 1a,b), consistent with a lower spontaneous CA rate in this cell line.

The concomitant presence of shared and reciprocal CAs affecting a single haplotype implies a multi-step process extending over successive cell cycles. To exemplify this, Fig. 2b depicts reciprocal CAs that, on the basis of our genomic reconstruction, arose over two consecutive breakage-fusion-bridge (BFB)[28,29] cycles. In the first cell cycle, sister chromatid fusion followed by bridge breakage gave rise to two cells carrying an inverted duplication on Chr. 14 along with a terminal deletion on the same homologue (Fig. 2c). In the second cell cycle, the cell carrying the terminal deletion underwent a second fusion and bridge breakage on this homologue. We also find evidence for complex CAs arising during the most recent cell cycle near the second bridge-breakage site (Fig. 2b; addressed further below). These data show how MAGIC enables reconstruction of CA processes over successive cell cycles.

## De novo CA landscapes in near-diploid cell lines

We next performed a comprehensive analysis of the CA landscape of spontaneously micronucleated cells, initially focusing on MCF10A (Fig. 2d and Extended Data Fig. 1c). We find that the most common CA class comprises a simple gain or loss of terminal chromosome segments ('terminal CAs'), representing 21.8% and 19.4% of CAs, respectively. By comparison, simple, interstitial CAs represent only 4% of the CAs detected. Whole-chromosome aneuploidies represent 16.9% of all CAs, with chromosomal losses ($N = 19$) being significantly more frequent than gains ($N = 2$; $P = 0.007$, permutation test; Supplementary Table 10). We also find 29 CAs (23.4%) that seem to have arisen from multi-step rearrangements affecting terminal segments of a single homologue (Fig. 2d and Extended Data Fig. 1b). Furthermore, we find 18 examples of clustered CAs unrelated to terminal multi-step events. Leveraging the haplotype resolution of Strand-seq, we confirm that the respective CAs are on the same homologue in line with complex CA formation, except for one instance where both homologues are affected. These complex CAs include four chromothripsis cases[30,31] with extensive rearrangements spread across the respective homologues (Extended Data Fig. 1e).

Analysis of the CA landscape in micronucleated RPE-1 cells revealed a similar range of CAs, with terminal CAs occurring most frequently. Yet, unlike in micronucleated MCF10A, we observe that complex CAs are essentially absent in RPE-1 cells (Fig. 2e and Extended Data Fig. 1d). Application of MAGIC to two more non-transformed cell lines—BJ-5ta and IMR-90 (Methods)—confirm widespread terminal CA formation

in spontaneously micronucleated cells, with complex CAs remaining comparably infrequent (Extended Data Fig. 2a–f). Furthermore, we compared these data with CA landscapes from MCF10A and RPE-1 cells exposed to the mitotic kinase MPS1 inhibitor reversine (Methods), which exhibit pervasive whole-chromosome aneuploidies both in the presence and absence of micronucleation[32–34], alongside a relatively low frequency of terminal CAs (Extended Data Fig. 3a–f). These data show that CA landscapes can vary substantially depending on whether CAs are induced biochemically, or arise spontaneously in non-transformed cells, highlighting the utility of MAGIC in distinguishing specific sources of chromosomal instability.

### Genomic contexts associated with spontaneous CAs

We next examined the genomic features of spontaneously arising de novo CA across the chromosome sets of MCF10A and RPE-1. Although we do not observe recurrent CA breakpoints, we find an uneven density of CAs across each karyotype. Analysing region-specific properties associated previously with somatic structural variants (Supplementary Notes), we observe significant overrepresentation of regions forming G4-quadruplexes, as well as both early and late-replicating regions (Extended Data Fig. 4a; adjusted $P < 0.05$; permutation test). Furthermore, under the assumption that each homologue acquires CAs with equal probability, we observe an enrichment of CAs on chromosome 19 in MCF10A (adjusted $P < 0.05$, binomial test; Fig. 2f). As most CAs are terminal to a chromosome arm and thus comprise the telomeres, we conducted long-read sequencing on an MCF10A-derived clone ('clone 7') to infer arm-specific telomere lengths (Methods). We observe a significant inverse correlation between the chromosomal arm CA frequency and telomere length estimates (Pearson's $R = -0.39$; $P = 0.0073$; Extended Data Fig. 4b), indicating that shortened telomeres[5,35–37] can foster mitotic errors resulting in CA formation.

By comparison, RPE-1 Chr. 2, Chr. 6, Chr. 10 and the X chromosome each exhibit elevated CAs (Fig. 2g, adjusted $P < 0.05$). Overall, we note a bias towards CAs affecting larger chromosomes in RPE-1, but not in MCF10A ($P < 0.05$; Extended Data Fig. 4c). This trend in RPE-1 is consistent with an earlier report using this cell line[32] (Extended Data Fig. 4d) and might originate from differences in the tendency of large versus small chromosomes[32,38,39] to be included in micronuclei in both cell line models.

### Post-selection CA landscape

CAs contributing to tumorigenesis must be maintained in the cell population. To investigate the potential of CAs to propagate clonally, we used MAGIC to isolate micronucleated cells and test their clone-forming capability. We observe a significantly reduced success rate in generating clones from micronucleated compared to normal cells (MCF10A: 0.73-fold reduced; RPE-1: 0.41-fold reduced; $P < 0.001$, Fisher's exact test; Extended Data Fig. 5a). We subjected 27 single-cell-derived clones from MCF10A (18 from micronucleated and 9 from control cells) and 11 RPE-1 clones (all from micronucleated cells) to low-pass whole-genome sequencing (WGS) (Methods). We find 13 clonally propagated CAs in the clones seeded from micronucleated MCF10A cells (Supplementary Table 1), with 9 out of 18 expanded cultures containing at least one CA (Extended Data Fig. 5b). These CAs compromise both simple ($N = 12$) and putatively complex ($N = 1$) events (Methods). By comparison, three clones grown from the controls each contain one simple CA. In RPE-1, none of the 11 micronucleated cell-derived clones exhibited CAs (Extended Data Fig. 5b; $P < 0.0052$, Fisher's Exact test versus MCF10A). Furthermore, we note reciprocal CAs are less frequent in RPE-1 sister cells with spontaneous micronuclei (Extended Data Fig. 5b), indicating that selective constraints limit CA propagation in the RPE-1 cell line.

We compared clonally propagated CAs with the de novo CA landscape of MCF10A cells. Notably, we observe a prevalence of losses on the 7q arm including simple and complex events, affecting 5 of 13 (38.5%) of all propagated CAs—a fourfold enrichment compared with the de novo

CAs ($P = 0.0043$; Bonferroni-corrected Chi-square test; Extended Data Fig. 5c,d). Although 7q-losses are common in breast cancer[8,40] (Supplementary Note), these CAs may either be subject to positive selection or could have persisted as selectively neutral events during clonal expansion, implying selective pressures[2] influence genomic CA landscapes. To investigate clonally maintained 7q events in a specific case, we analysed the long-read sequencing data generated for MCF10A clone 7, for which low-pass WGS indicated complex CA formation involving Chr. 7. Long-read analysis confirmed the existence of these complex CAs, uncovering a chromothripsis event accompanied by isochromosome formation, which ultimately resulted in 7q loss (Extended Data Fig. 5e,f). These data illustrate how MAGIC can be used to select cells undergoing CAs for phenotypic analyses and clone-based sequencing.

## CA processes acting in micronucleated cells
### Pivotal role of dicentric chromosomes

Harnessing the combined Strand-seq data generated for MCF10A and RPE-1, we next systematically inferred de novo CA processes. We first investigated terminal CAs, which represent 64.6% and 75.5% of all CAs seen in MCF10A and RPE-1, respectively. Out of the 49 terminal gains observed in both cell lines involving either parts of or an entire chromosomal arm, 42 (85.7%) show a configuration where the segment gained at the terminus has a strand-state opposite to that of its homologue (Extended Data Fig. 6a). This karyotypic pattern could arise from a terminal inverted duplication arising during a BFB cycle (Fig. 2h), yet may alternatively reflect acentric fragments entering mitosis unrepaired and undergoing asymmetric segregation (Extended Data Fig. 6c).

Although both scenarios would yield reciprocal gain–loss in sister cells, BFB cycles typically result in sequential CAs affecting the same homologue. Among several CAs mapping to the same chromosome, where at least one involves a terminal segment, 86.2% can be traced back to the same homologue, consistent with BFBs (Fig. 2h and Extended Data Fig. 6b). These data are further bolstered by our analysis of three sister cell pairs harbouring at least one shared and one reciprocal CA on the same homologue, in each case supporting the occurrence of multi-step BFBs (Fig. 2b and Extended Data Fig. 1b). These data highlight the pivotal role of dicentrics in facilitating spontaneous CA formation.

### Complex CAs

We next focused on other CA processes. We observe two distinct types of complex CA implicating chromothripsis. Among all 54 CAs involving a terminal deletion or inverted duplication in spontaneously micronucleated MCF10A cells, 11 (20.4%) exhibit a localized copy-number oscillation pattern near the internal breakpoint (Extended Data Fig. 6d,e). This pattern is further corroborated by its similar occurrence frequency in MCF10A control cells, and is likewise detected in RPE-1 cells (Extended Data Fig. 6e). Pooling examples of this pattern across all examined conditions, we observe a single oscillation in most cases (11 of 16), characterized by troughs and crests of similar size (averaging 1.6 Mb, with the whole oscillation pattern spanning from 2 to 7 Mb; Extended Data Fig. 6f–h). Assuming chromatin bridge breakage as the source of this pattern, the location of these complex CAs corresponds to the point of rupture, indicating a link to bridge resolution. The position and oscillatory characteristics of this pattern resemble previously described instances of chromothripsis associated with dicentric breakage, mediated by cytosolic enzyme activity[5], implicating this mechanism in spontaneous complex CA formation.

### Chromosome pulverization

In MCF10A cells, but not in RPE-1, we observed four instances of whole-chromosome or whole-arm-level chromothripsis that we subjected to more detailed analysis. The respective rearrangements are confined to a single homologue and show evidence for random fragmentation, in line with established chromothripsis criteria[30] (Fig. 2i and Extended

Data Figs. 1e and 5i). In two instances, we identified the corresponding sister cell from the Strand-seq data (this included a single sister cell not initially passing quality control). In both instances, we observe anti-correlated read counts (Fig. 2i and Extended Data Fig. 5g–i), in line with the reciprocal segregation of pulverized chromosome fragments. These CA patterns closely mirror previous reports of chromothripsis linked to micronucleus entrapment, observed in TP53-depleted RPE-1 cells following monastrol washout[4]. Altogether, up to 13% of CAs in spontaneously micronucleated MCF10A cells can be attributed to chromothripsis, considering both focal and chromosome-wide patterns.

## TP53 status affects de novo CA formation

### Analysis of TP53−/− cells

Disruption of TP53, causing loss of the p53 tumour suppressor, is the most common driver mutation in cancer, and associated with a range of genomic instability patterns[41–44]. Yet, the potential roles of TP53 deficiency in CA mutational rates and in determining the de novo CA landscape remains underexplored. By performing single-cell transcriptomics coupled with MAGIC in MCF10A and RPE-1, we find strong evidence for cell cycle arrest and over-expression of TP53 or its targets in micronucleated as opposed to normal cells (Supplementary Note and Extended Data Fig. 7a–c), indicating the DNA damage response may constrain CA formation. To explore the effect of TP53 in CA formation, we used isogenic TP53−/− models of MCF10A and RPE-1 (Supplementary Fig. 4a; Methods). Using microscopy, we find a general increase in nuclear atypia in both of these TP53−/− cell lines compared with their unmutated ('wild-type') counterparts (Supplementary Fig. 4b,c). For example, about one-third of TP53−/− MCF10A cells exhibit micronuclei (Fig. 3a)—an increase accompanied by a high frequency of anaphase bridges (36.8%; Fig. 3b). Moreover, the probability of anaphase errors to result in a micronucleated daughter is increased to 73.3% and 69.7% for anaphase lagging chromosomes and chromatin bridges, respectively (Supplementary Fig. 4d). Similar to their wild-type counterparts, micronucleated TP53−/− cells are prone to generate a micronucleated daughter cell (Supplementary Fig. 4e,f). Yet, unlike wild-type cells, the duration of the cell cycle is not prolonged and TP53−/− cells do not effectively enter cell cycle arrest[42] (Supplementary Fig. 4g,h). These observations hint at an elevated rate of spontaneous CAs in TP53−/− cells, with the absence of efficient cell cycle arrest potentially promoting CA formation.

To investigate the effect of TP53 disruption on CAs, we subjected both TP53−/− cell lines to MAGIC. Analysis of 300 single-cell genomes indicates a marked increase in de novo CAs (Fig. 3c), with TP53−/− micronucleated cells exhibiting significantly more CAs than wild-type micronucleated cells (Fig. 3d; $P < 2.65 \times 10^{-10}$ for MCF10A; $P < 9.36 \times 10^{-6}$ for RPE-1). We next conducted an analysis of CA classes arising in both TP53−/− cell lines (Fig. 3e,f and Supplementary Fig. 5a,b). Although the CA spectra seem very similar between TP53−/− and wild-type cells (Supplementary Fig. 5c), a notable exception is the marked increase in complex CAs seen in micronucleated TP53−/− RPE-1 cells compared with wild type (from 2.7% to 16.5%; $P < 0.05$ Fisher's exact test; cf. Fig. 2e and Fig. 3f), which includes chromothripsis events. This finding is consistent with TP53 status exerting a particularly strong effect on complex CA formation[41].

### CA mutation rates

Accurately estimating the baseline mutational rate of somatic CAs was previously unfeasible due to technological limitations[2]. Harnessing the imaging and genomic data generated in our study, we devised a statistical agent-based model (Methods) where simulated cells transition between nuclear atypia and normal mitoses on the basis of probabilities derived from live-cell long-term imaging (Figs. 1b and 3g). This model allows simulating the CA rate associated with three mitosis types: normal (with lagging chromosomes ('laggard')) and with chromatin bridges ('bridge'). We selected MCF10A, given that CAs

arise in both TP53−/− and wild-type contexts in this line. The simulation closely recapitulates our empirical data (Supplementary Fig. 6a,b), enabling estimation of the mitosis type-specific CA rate (Supplementary Methods). We calculated that 3.7% of normal cell divisions result in a CA for wild-type cells, whereas this value increases to 92.5% and 84.4% for laggard and bridge mitoses in wild-type cells, respectively (Fig. 3h). In TP53−/− cells, mitosis type-specific CA rate estimates are very similar—2.9% for normal, 82.8% for laggard and 83.2% for bridge mitoses per cell division—indicating that the underlying processes by which CAs form through nuclear atypia are not affected by TP53 deficiency. Finally, by considering the relative contribution of mitosis types, we estimate the basal CA rate for MCF10A, which is 13.3% in wild-type cells per cell division, and approximately doubles to 30.4% in TP53−/− cells. This increase seems to be driven particularly by the higher proportion of chromatin bridges in TP53−/− cells (Fig. 3b), consistent with dicentrics representing an important trigger for CA formation.

## Chromosome region determinants for de novo CAs

### Modelling CA formation with targeted DSBs

Understanding the mechanistic origins of chromosomal instability requires clarifying how CAs arise from initial DNA lesions, particularly DSBs. Considering the patterns observed in our data, we reasoned that following an initial DSB trigger, the size and nature of fragments generated and whether they result in a dicentric or acentric chromosome (Fig. 2h and Extended Data Fig. 6c) are likely to influence their fate, implying that the chromosomal DSB location could have an important role in determining CA processes. Reanalysis of X chromosomal HPRT1 data reveals that most segmental CAs (53.5%; 23 of 43) involve centrally oriented alterations near the cut site, indicating that BFB cycles are triggered frequently by CRISPR–Cas9 treatment at this locus. To explore the potential relationship between DSB location and CA process, we devised a MAGIC experiment generating DSBs on Chrs. 2 and 7 (Methods), with each chromosome targeted at specific sub-centromeric, sub-telomeric and central sites of the q arm (Fig. 4a and Supplementary Table 2). After subjecting MCF10A cells to targeted DSBs, we sequenced 361 single-cell genomes from micronucleated cells. Although we observe different CA induction efficiencies for each targeted DSB (from 20% to 60%), in each case most CAs originate from the gRNA-directed DSB sites (Fig. 4b).

Analysing the CA spectra separately by DSB site, we observe a wide diversity of CA classes (Fig. 4c,d), including cases of chromothripsis (Extended Data Fig. 8f). However, the relative proportions of CA classes differ substantially by cut location, with patterns largely consistent between 7q and 2q (Fig. 4c,d and Extended Data Fig. 8a,b). For example, we find terminal CAs affecting the q arm—seen with 29.4% and 80.0% for 7q—when targeting the central and sub-telomeric site, respectively, whereas whole-arm CAs arise exclusively from sub-centromeric DSBs. Furthermore, when focusing on those CAs initially annotated as either terminal or complex, we find several cases of terminal deletions with an inverted duplication centrally located relative to the DSB (Fig. 4d and Extended Data Fig. 8a,c), consistent with BFBs; these bridge-related CAs are enriched more than eightfold in central and sub-telomeric cuts compared with sub-centromeric cuts (Fig. 4e and Extended Data Fig. 8d).

Moreover, when targeting the sub-centromeres, we observe a notable frequency (10.5% and 9.1% for 7q and 2q, respectively) of whole-arm CAs sharing a distinctive pattern characterized by a p arm gain in inverted orientation coupled with q arm loss (Fig. 4d and Extended Data Fig. 8a,g). This pattern is indicative of isochromosome formation, reflecting a derivative chromosome structure recurrent in different cancer types[8,45]. By comparison, neither central nor sub-telomeric cuts result in isochromosomes (Fig. 4f and Extended Data Fig. 8e). Taken together, these data provide strong evidence that different DSB sites can promote distinct CA processes.

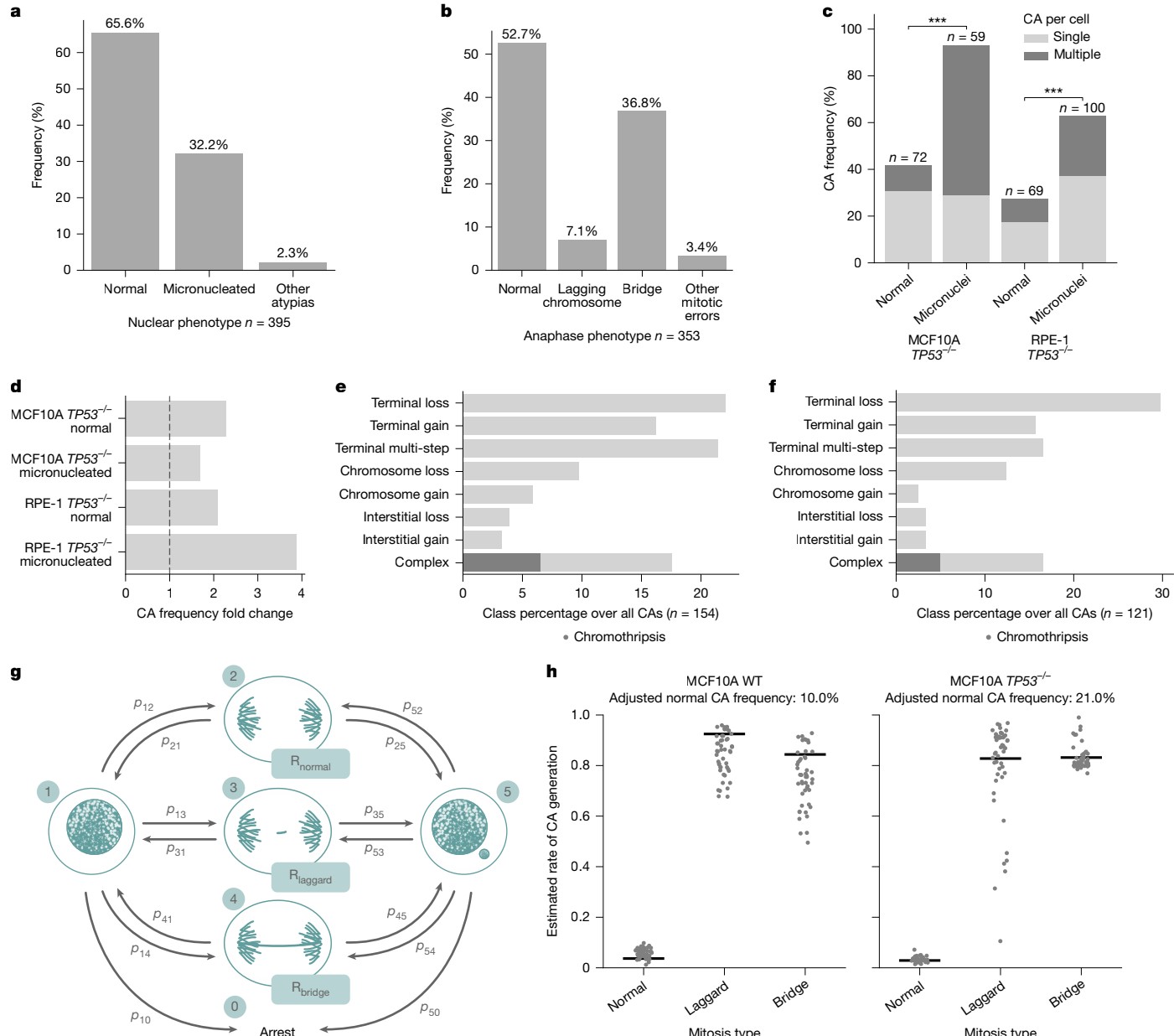

**Fig. 3 | Effect of *TP53* disruption on de novo CA formation, and modelling basal CA rates. a,b,** Frequency of nuclear (**a**) and anaphase (**b**) phenotypes in MCF10A *TP53*−/− cells from long-term live-cell imaging. **c**, Observed CA frequency by cell (Fisher's exact test). **d**, Fold change in CA numbers per cell for *TP53*−/− versus the respectively matched wild-type cell line model, shown across cell line models and nuclear phenotypes. **e,f,** Breakdown of CA classes in micronucleated MCF10A *TP53*−/− (**e**) and RPE-1 *TP53*−/− (**f**) cells, class percentage over all CAs. **g**, Agent-based model for estimating the de novo CA rate. Scheme representing available states (numbers in teal circles; 1, normal cell; 2, normal mitosis; 3, laggard mitosis; 4, bridge mitosis; 5, micronucleated cell) and available transitions (arrows) between states, with $P_{ij}$ being the empirically measured transition probabilities. Parameters are estimated for three mitosis types with $R$ representing estimated CA rates. **h**, Mitosis type-specific CA rate estimations in MCF10 wild-type (WT) and *TP53*−/− models. Each data point represents the estimated value from an optimization run. Black lines represent the average weighted by the residual error of each optimized simulation.

## Acentric fragments result in distinctive CA patterns

Across cut sites, we also observe several cases of amplification of the generated acentric fragment. This pattern accounts for up to 40% of all CAs, depending on the cut site (Supplementary Fig. 7a–c), and is characterized by the simultaneous gain of both Watson and Crick templates[27] in the Strand-seq data (Supplementary Fig. 7d): particularly, among 18 acentric gains with a copy-number increment of two, the Watson/Crick ratio remains 1:1 in all 18 cases, with Crick/Crick and Watson/Watson configurations missing entirely ($P < 7.63 \times 10^{-6}$, binomial test; Supplementary Methods). This peculiar strand pattern implies that these duplicated acentric segments are integrated into the same derivative chromosome, promoting their co-segregation in multiples of two, with the segments arranged in inverted orientation (Supplementary Fig. 7e). This inference is corroborated by sister cell pair analysis demonstrating the joint segregation of gains in multiples of two (Fig. 4g,h). In summary, coupling MAGIC with targeted DSBs provides evidence for a CA process that enables the co-segregation of amplified acentrics.

## Verification by fluorescent in situ hybridization

To further investigate these reconstructed CA patterns, we conducted a further round of targeted DSB experiments, this time coupled with fluorescent in situ hybridization (FISH). We used a two-probe strategy labelling the sub-centromeric regions of 7p and 7q, respectively.

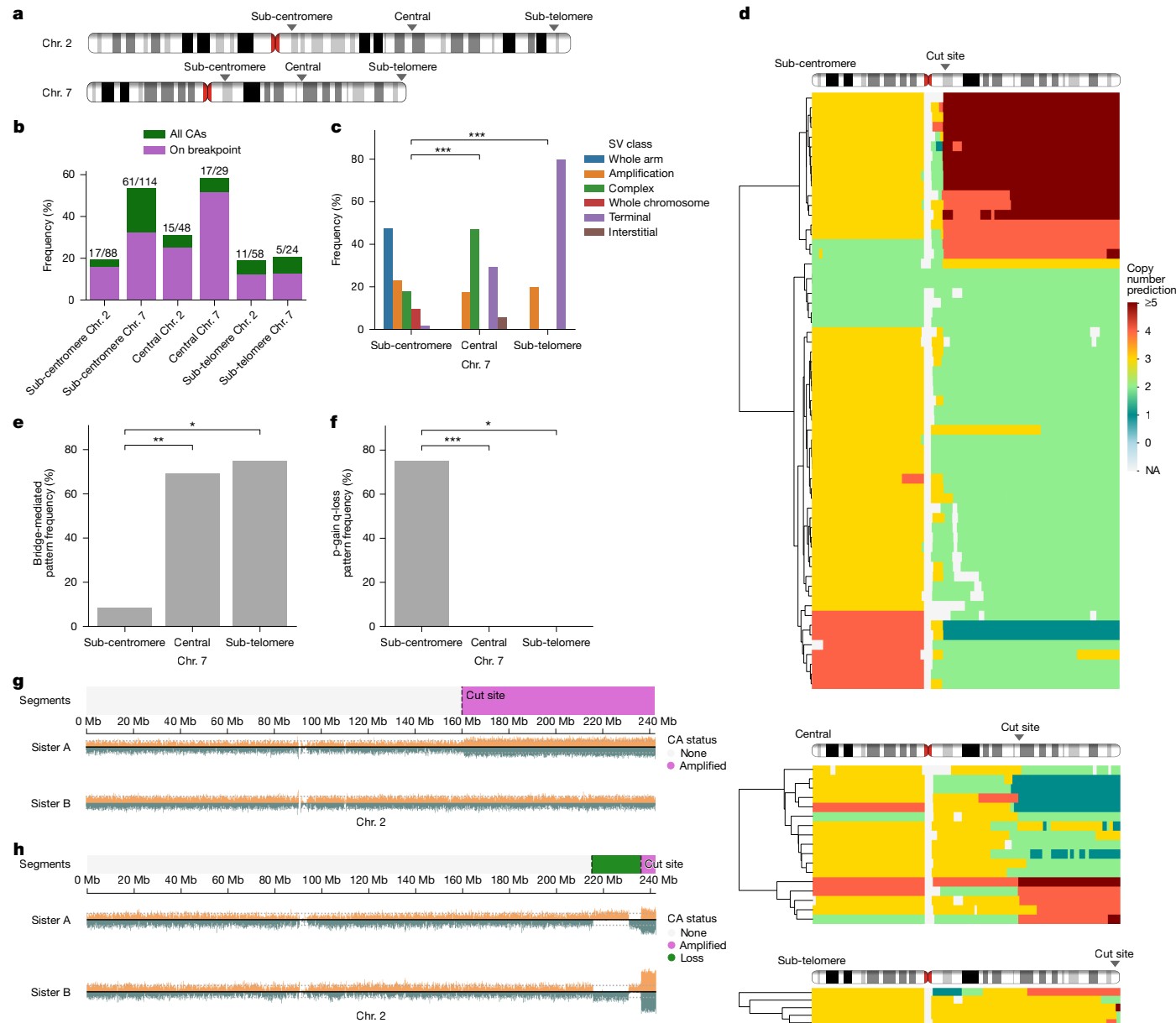

**Fig. 4 | CA landscape following targeted DSB induction along chromosome arms. a**, Scheme showing the locations of targeted DSBs. **b**, CA frequency per cell across the different chromosomal cut sites. **c**, Frequency of CA types for different cut sites on the Chr. 7 q arm (Fisher's exact test). SV, structural variant. **d**, Overview of CAs for the Chr. 7 q arm, expressed as copy-number states. Sub-centromere, central and sub-telomere target loci are depicted from top to bottom. (Note the consensus copy number for Chr. 7 in MCF10A H2B-Dendra2 cells is three–trisomy). **e,f**, Enrichment for bridge-mediated CAs (**e**) and coupled p arm gain/q arm loss indicative for isochromosome formation (**f**) (Fisher's exact test). Only terminal and complex CAs were considered in this analysis. **g,h**, Sister cells showing one-sided (**g**) and asymmetric (**h**) inheritance of the acentric fragment in multiples of two. The sister cell examples shown here are from the Chr. 2 central cut and HPRT1 datasets, respectively.

The gRNA cut site, thereby, is located within our designed q-arm FISH probe, facilitating the analysis of CA outcomes (Fig. 5a). Following centromeric cuts, we find that 28% of metaphases have an abnormal sub-centromeric probe signal indicative for CAs, an outcome not observed for sub-telomeric cuts (Supplementary Fig. 8a).

Systematic analysis of metaphase spreads reveals CA patterns confirming those identified through Strand-seq. Following sub-centromeric cutting, we observe loss of the long arm at the DSB site in 33.3% of all spreads with an abnormal Chr. 7 (centric fragment; Fig. 5b). Isochromosomes account for 11.6% of all abnormalities, as visualized by two sub-centromeric p-arm signals surrounding a single sub-centromeric q-arm signal (Fig. 5b,c). These data validate isochromosome formation following targeted DSB generation, resulting in a

derivative chromosome that, despite comprising two centromeres, seems to represent a chromosomally stable structure.

Notably, these FISH experiments also highlight abnormalities affecting acentrics. Acentrics appear as isolated chromosome fragments, with a q-arm signal at one extremity, in 30.4% of spreads with an abnormal Chr. 7 (Fig. 5b and Supplementary Fig. 8b). In 18.8% of cases, we detect 7q acentric fragments that have doubled in size, bearing a sub-centromeric q-arm probe signal located at the middle (Fig. 5b,c and Supplementary Fig. 8c). These visualized chromosomal derivatives represent isoacentrics, characterized by two inverted acentric arms fused at the cut site, thus confirming our Strand-seq based genome reconstructions. Notably, the isoacentrics occasionally appear thin and elongated, indicating that they could be subject to abnormal chromatin

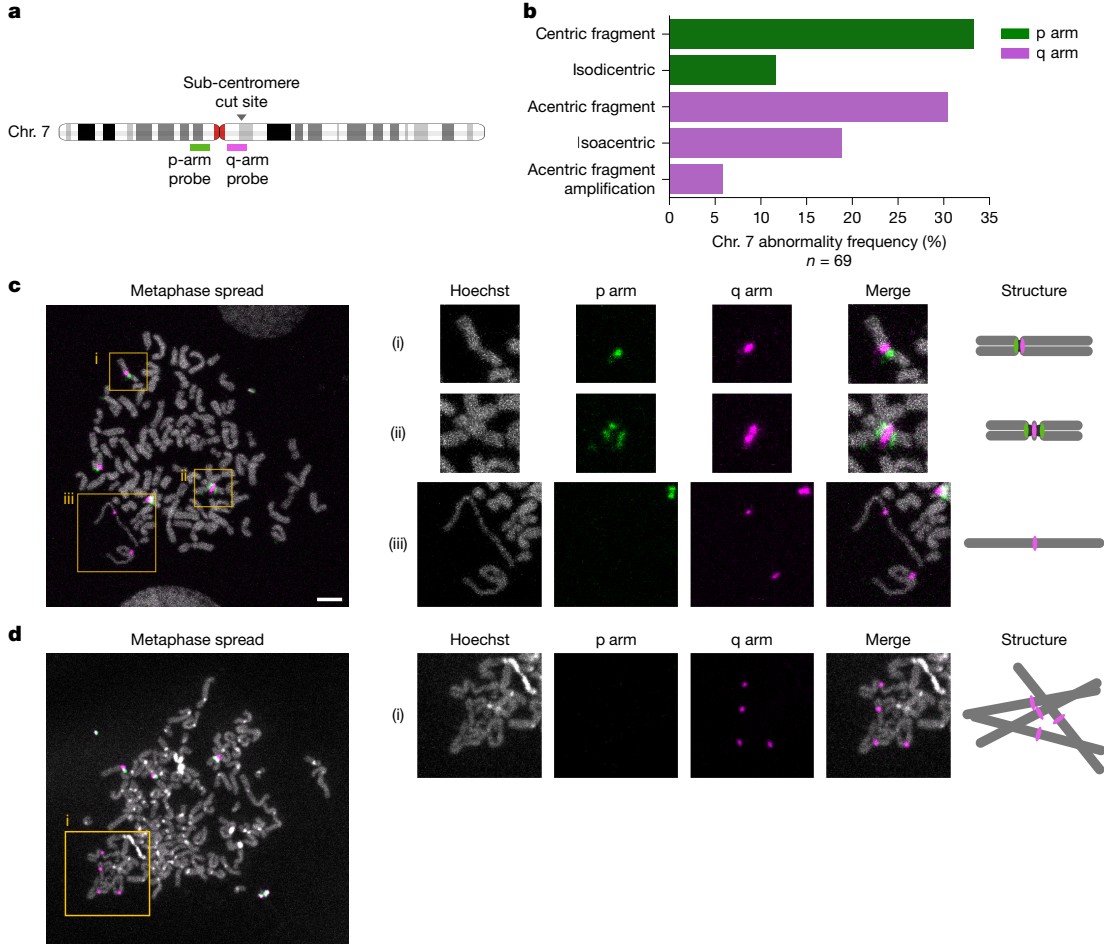

**Fig. 5 | Isodicentric and isoacentric derivative chromosome generation through targeted DSBs. a**, Scheme depicting hybridization sites for sub-centromeric p-arm and q-arm FISH probes. The targeted DSB site, 'splitting' one of the FISH probe locations to facilitate the examination of chromosome rearrangement outcomes, is indicated by an arrowhead. **b**, FISH-based quantification of Chr. 7 abnormalities showing an abnormal probe pattern in metaphase spreads. A total of 69 abnormal Chr. 7s were analysed over 155 metaphases. **c**, Metaphase spread example revealing different derivative chromosomes. Left, regions of interest (ROIs) are marked by yellow squares.

Right, magnified ROIs, with breakdown per fluorescent channel and putative chromosomal structures. ROI (i), normal Chr. 7; ROI (ii), isodicentric chromosome with a q-arm signal surrounded by two p-arm signals, indicating the presence of two adjacent centromeres and duplication of 7q; ROI (iii), isoacentric derivative chromosomes with signs of premature chromatin condensation. **d**, Metaphase spread example with amplified isoacentrics. ROI (i) shows visibly clustering and amplified isoacentrics, with several interspersed q-arm signals clearly visible. Scale bars, 5 μm.

condensation (Fig. 5c). Such morphology has been associated previously with premature chromatin condensation[46,47], and could reflect under-replication due to micronucleus entrapment[47]. Furthermore, in 5.8% of metaphase spreads with an abnormal sub-centromeric probe signal, we observe further amplified isoacentrics, which present as clusters of condensed DNA with interspersed sub-centromeric q-arm signals (Fig. 5b,d). It is intriguing to speculate that these condensed DNA structures might promote the co-segregation of highly amplified genetic material in multiples of two. These observations support the utility of MAGIC in leveraging targeted DSBs to dissect the origins of CA formation.

## Discussion

We show that MAGIC facilitates investigating spontaneously arising CAs, providing a representative view of the de novo CA landscape linked to micronucleation in non-transformed cell lines. Dicentric chromosomes represent key drivers of karyotypic diversification, capable of triggering homologue-specific changes across successive cell divisions. Similarities between the CA patterns identified here and those reported in advanced cancer stages[13] (Supplementary Fig. 9 and Supplementary Notes) imply a potentially significant role of micronuclei in shaping cancer genome evolution; however, our data also show marked differences between spontaneously arising CAs and the post-selection CA landscape.

Analysis of our dataset as a whole reveals a distinct bias for de novo whole-chromosome losses compared with chromosome gain, observed under different experimental conditions (Supplementary Table 10). These data are supported by recent findings implicating CRISPR-based genome manipulation specifically in the induction of chromosome losses[48,49]. Furthermore, in an analysis of 2,600 cancer genomes[8] we observe that chromosome losses predominate markedly (81.5%) over chromosome gains, even when excluding cases subject to whole-genome duplication (Extended Data Fig. 9a). The mechanism underlying this marked bias towards chromosome losses remains unclear. Although proteotoxic stress linked to trisomy can select against chromosome gains[50,51], our data indicate that the bias is established during CA formation, preceding proteotoxic effects. Furthermore, MPS1 inhibition, which can induce missegregation in the presence and absence of micronucleation[34], yields balanced chromosome gains and losses (Extended Data Fig. 3a–f), arguing against immediate proteotoxic selection. It therefore seems likely

that micronucleus-specific processes, such as DNA replication defects and DNA damage[22] as well as micronucleus elimination[52,53], or dicentric segregation into a single daughter[54], contribute to the chromosome loss bias observed in our study.

MCF10A and RPE-1 cells show certain differences in their CA formation patterns. MCF10A cells frequently develop complex CAs and even chromothripsis, despite *TP53* wild-type status. This is potentially facilitated by immortalizing events that occurred on MCF10A, including gain of *MYC* and loss of *CDKN2A* and *CDKN2B*[55]. By contrast, RPE-1 cells maintain a relatively stable karyotype, and only rarely exhibit complex CAs unless *TP53* is lost. Irrespective of this, our sister cell analyses indicate that RPE-1 cells occasionally tolerate de novo CAs, implying p53 surveillance can be bypassed (Fig. 2e). Tolerance of de novo CAs is similarly observed following biochemical perturbation of chromosome segregation (Extended Data Fig. 3d–f). These findings indicate that MAGIC could provide a framework for uncovering how cell-intrinsic factors, including DNA repair activity and cell cycle regulation, influence chromosome instability and context-specific determinants of CA formation and tolerance.

Integrating CRISPR–Cas9 and MAGIC, we show that the location of initiating DSBs distinctly influences CA outcomes, resulting either in stable derivative chromosomes (particularly isochromosomes) or facilitating further chromosomal instability. Our data support a single-DSB U-type exchange mechanism for isochromosome formation, initiated by a sub-centromeric DSB, and followed by DNA replication and subsequent sister chromatid end fusion[56]. Compared with a process involving two independent DSBs, this mechanism offers a simpler, and thus more parsimonious, model.

With respect to isochromosomes, our results underscore the significance of the inter-centromeric distance of fused chromatids in determining CA outcomes (Fig. 5c and Supplementary Fig. 8d). Longer inter-centromeric distances enable dual kinetochore attachments, causing chromatin bridges and further chromosomal instability. By comparison, shorter distances can result in a single kinetochore attachment enabling stable mitotic segregation of dicentric isochromosomes (isodicentrics). Indeed, cancer genome analysis demonstrates isodicentrics are widespread in tumours (31% of samples in the Cancer Genome Atlas dataset; 55% in the Pan-Cancer Analysis of Whole Genomes dataset; Extended Data Fig. 9b,c and Supplementary Notes), with inter-centromeric distances occasionally exceeding 20 Mb in length.

Our targeted DSB experiments also reveal asymmetric segregation of acentric segments amplified in inverted orientation (isoacentrics). These derivative chromosomes probably form through fusion of an acentric fragment with its sister chromatid or by aberrant replication. They may facilitate rapid DNA segment amplification, and potentially explain the recurrent inheritance of chromosomal segments in multiples of two, observed recently from in vitro screens[57]. In spontaneously micronucleated MCF10A and RPE-1 cells, we detect isoacentric formation in up to 3% (Supplementary Fig. 7f,g). Upon targeted DSB induction, their relative frequency increases by approximately tenfold (Supplementary Fig. 7h). Likewise, we find that isochromosome formation is relatively frequent following targeted sub-centromeric DSBs, but occurs only occasionally in spontaneously micronucleated cells (Supplementary Fig. 7i). This indicates that chromosomal fragments emerging from internal unrepaired DSBs do not represent primary drivers of spontaneous CA formation in these cell lines, with a larger fraction of CAs appearing to spontaneously arise from lesions at (or near) the telomeres.

Furthermore, the inverted duplication architecture of isoacentrics observed in our study implies that fold-back inversions[11,58] may occasionally result from CA processes independent of classical BFB cycles, with implications for interpreting rearrangement patterns in cancer genomes. Mitotic clustering of isoacentric derivative chromosomes could facilitate their asymmetric segregation after subsequent rounds of isoacentric amplification[17,18]. It is intriguing to speculate that this might promote oncogene amplification, extrachromosomal DNA (ecDNA) formation, or complex rearrangements in cancer genomes when coupled with other CA processes.

MAGIC enables automated analysis of several tens of thousands of cells per experiment, permitting the isolation of rare cell morphologies at large numbers, and thus overcoming previous limitations in studying nuclear atypia. In total, we isolated 2,898 single cells and sequenced 2,192 single-cell genomes in this study, generating an unprecedented dataset for investigating de novo CAs. Nevertheless, methodological constraints remain: due to the intermediate coverage achieved, the single-cell sequencing approach we coupled with MAGIC (Strand-seq) is limited to detecting CAs larger than 200 kb. Furthermore, because Strand-seq requires BrdU incorporation, only dividing cells are sequenced, potentially underrepresenting CAs leading to immediate cell cycle arrest. Coupling MAGIC with complementary single-cell sequencing approaches[4,59,60] could allow studies of CA formation in non-dividing cells, enhance sensitivity for smaller genetic variants and ecDNAs and improve CA breakpoint-resolution to inform mechanistic analyses[2] of underlying CA formation processes.

Looking ahead, MAGIC holds promise for versatile future applications. Future studies could exploit MAGIC to target other nuclear atypia, or expand analyses to primary cell types. Integration of advanced deep learning-based nuclear segmentation approaches[61,62] would broaden morphological classification capabilities. The openly accessible computational workflows accompanying MAGIC (Methods) thereby allow optimization of resolution, experimental duration and cell yield. Further method advancements, including enrichment of sister cell pairs or linking single-cell sequencing data directly to cell images through automated cell picking, could facilitate the investigation of particular CA processes, albeit with potential trade-offs in throughput. Realization of such further method developments could facilitate comprehensive delineation of CA-associated mutational processes arising before Darwinian selection acts (Supplementary Note), enhancing our understanding of cancer evolutionary mechanisms.

In conclusion, MAGIC enables systematic investigation of sporadic CAs in non-transformed cells. Our results demonstrate that dicentrics drive chromosome instability, DSB location influences CA outcomes and *TP53* status shapes the CA mutation rate. These insights lay the groundwork for future research aimed at explaining tumorigenesis driven through somatic karyotype evolution.

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

## Methods

### Statistical analysis

Unless otherwise stated, we used the following system to indicate significance levels in the figure panels: *P* < 0.05; **P* < 0.01; ***P* < 0.001. Statistical tests used are indicated in the main text or figure caption, with specific tests for chromosome biases and breakpoint as well as SCE locations detailed in the Supplementary Methods.

### Cell culture and cell line development

MCF10A (CRL-10317, American Type Culture Collection) and RPE-1 (CRL-4000, American Type Culture Collection) cell line and their *TP53*[-/-] derivatives were cultured at 37 °C with 5% $CO_2$ atmosphere and 100% humidity, in DMEM/F12 medium (1:1) without phenol red (Gibco), supplemented as follows: RPE-1 medium was further supplemented with 10% FCS, 2 mM L-glutamine (Gibco) and antibiotics; MCF10A medium with 5% horse serum (Thermo Fisher Scientific), 2 mM L-glutamine (Gibco), 20 ng ml⁻¹ human EGF (Biotrend), 0.5 mg ml⁻¹ hydrocortisone (Sigma-Aldrich), 100 ng ml⁻¹ cholera toxin (Sigma-Aldrich), 10 µg ml⁻¹ recombinant human insulin (Sigma-Aldrich) and antibiotics. BJ-5ta (CRL-4001, American Type Culture Collection) were cultured at 37 °C with 5% $CO_2$ atmosphere and 100% humidity in a 4:1 ratio of DMEM (Gibco) and Medium 199 (Gibco) without phenol red, supplemented with 10% FCS, 2 mM L-glutamine (Gibco) and antibiotics. IMR-90 (CCL-186, American Type Culture Collection) were cultured at 37 °C with 5% $CO_2$ atmosphere and 100% humidity in Minimum Essential Medium containing Earle's salts and without phenol red (Gibco), supplemented with 10% FCS, 2 mM L-glutamine (Gibco), 1 mM sodium pyruvate (Gibco), 1× NEAA (Gibco) and antibiotics; cells were discarded after 15 population doublings. MCF10A *TP53*[-/-] cells were kindly provided by C. Scholl (Laboratory of Applied Functional Genomics, DKFZ), whereas RPE-1 *TP53*[-/-] variants were generated in a previous study from our laboratory[63]. All cell lines tested negative for mycoplasma contamination.

For experiments using H2B-Dendra2 as photolabelling strategy, a plasmid carrying H2B-Dendra2 (ref. 64) (Addgene, plasmid no. 75283) was introduced by transfection: 20,000 cells were seeded in a glass-bottom slide (Nunc LabTek eight-well) and transfected with 20 µl of transfection mixture at 4:1 ratio of Fugene HD (Promega) to DNA in Opti-MEM (Thermo Fisher Scientific). Transfection success was assessed 48 h later by fluorescence microscopy, cells were transferred into two 10-cm dishes and G418 antibiotic was added at 200 µg ml⁻¹ (MCF10A) or 400 µg ml⁻¹ (RPE-1) for selection. Two weeks later, well separated, fluorescent colonies were visible and were isolated by pipetting, transferred to 24-well plates and grown into stable cell lines. Stable-transfectants for RPE-1 wild-type and RPE-1 *TP53*[-/-] were instead collected in pool and isolated by single-cell sorting using a BD FACSAria at 1.0 flow rate, with a 130 µm nozzle, dispensed in a flat-bottom, 96-well plate (Thermo Fisher, Nunc plates) with normal growth medium. In experiments designed to induce micronucleus formation biochemically, MCF10A and RPE-1 cells were treated with 0.5 µM of reversine (Sigma), a potent MPS1 inhibitor[65], 1 day after seeding. After 24 h of treatment, cells were washed gently four times with 1× PBS before being released into fresh medium.

### MAGIC: autonomous platform for de novo CA formation studies

MAGIC leverages machine learning and automated microscopy to perform targeted photolabelling of cells of interest, for subsequent fluorescence-activated cell sorting and downstream analysis, building on approaches coupling the imaging of visual phenotypes with precise optical tagging[66–68]. An MAGIC experiment with this adaptive feedback microscopy (Smart Microscopy) system comprises three phases: (1) the preparation phase of MAGIC, where cells are seeded and other treatments, such as targeted DSB induction or staining by DACT-1, can take place; (2) the photolabelling phase of MAGIC, where targeted illumination[66,67,69] takes place using automated microscopy

and (3) the cell collection phase of MAGIC, when cells are collected and isolated by FACS. These steps are outlined below, accompanied by further details presented in the Supplementary Methods.

**Preparation.** During this phase cells are prepared to undergo the targeted photolabelling procedure. Further treatments, such as targeted DSB induction, staining with live-cell dyes and adding BrdU for Strand-seq, can take place. To enable photolabelling, we engineered MCF10A and RPE-1 cell line models to constitutively express H2B-Dendra2—a monomeric fluorescent protein that undergoes irreversible photoconversion with 405 nm light, which also enables the visualization of nuclear atypia without affecting mitotic fidelity[70]. As an alternative, for RPE-1 wild-type cells, as well as BJ-5ta and IMR-90, we also used DACT-1—a photo-activatable cell tracking dye—that converts to a bright red-fluorescent state upon 405 nm light exposure (further details are available in the Supplementary Methods). Neither H2B-Dendra2 nor DACT-1 significantly altered micronucleus frequency in MCF10A cells (Supplementary Fig. 1f), indicating that these labelling approaches, by themselves, do not induce chromosomal instability under the conditions used.

Cells were seeded in up to four wells of a µ-slide eight-well dish (Ibidi). Seeding density was adjusted to have about 40,000 cells 1 day before experiment start. In the case of Strand-seq downstream analysis, BrdU (40 µM final concentration) was added to the cells before the start of photolabelling (a concentration previously reported not to cause genomic instability[71]). One control slide without BrdU was also prepared to adjust gating strategies during single-cell sorting. In the case of targeted DSB induction experiments, ribonucleoprotein (RNP) complexes were delivered by electroporation 48 h before the start of the experiment and up to two different sgRNAs were examined during a single experiment.

**Photolabelling.** Living cells were then transferred to an LSM 900 microscope (Zeiss) with confocal and widefield imaging capabilities, and an environmental chamber with temperature and $CO_2$ control. MAGIC relies on full microscope automation and computer vision for laser-assisted, phenotype-driven targeted illumination of single cells at scale. The system includes three software components: a microscope control script, an image analysis manager on the basis of AutoMicTools and a Python package, magic_tools, which we designed for advanced image processing.

The microscope control script automates autofocusing, micronuclei identification and photoconversion of target nuclei across several positions. Autofocus is achieved by detecting the glass-bottom dish reflection using a 639 nm laser and AutoMicTools analysis. For micronuclei identification, a *Z* stack image centred on the focused slice is analysed on an image analysis server driven by magic_tools. Photoconversion involves using micronuclei coordinates to define ROIs of the corresponding parental nuclei, which are then photolabelled selectively with a 405 nm laser. Pre- and post-experiment images are acquired before the microscope moves to the next position.

We ran the photolabelling experiment overnight and up to 24 h, to achieve a yield of 700 to 2,000 photolabelled cells, depending on the experimental conditions. A detailed description of the automation software and the image analysis pipeline can be found in Supplementary Methods.

**Cell collection.** Following photolabelling, cells were collected and target cells were isolated by single-cell sorting. In case of Strand-seq experiments, at the end of the photolabelling phase, cells were stained for 1 h with Hoechst 33342 at 5 µg ml⁻¹. Cells were collected with 0.25% trypsin (Gibco) and resuspended in buffer (8% FBS in 1× PBS, supplemented with Hoechst 33342 5 µg ml⁻¹ and BrdU 40 µM). Single cells were sorted using a BD FACSAria in purity mode with a 100-µm or 130-µm nozzle and dispensed into lysis buffer or fresh medium in a flat-bottom 96-well plate (Thermo Fisher, Nunc plates). We used the

following gating strategy: we selected first the general population in forward and side scatter and we excluded doublets. Then, cells were sub-gated for photolabelled cells as shown in Fig. 1g for H2B-Dendra2 or Supplementary Fig. 2h for DACT-1. When using Strand-seq, the singlet population was further filtered to select cells with a quenched Hoechst signal that had thus incorporated BrdU[72]. Cells collected from control slides were used to optimally adjust gates to exclude false positives.

## Long-term live-cell imaging

The live-imaging experiment for nuclear and mitotic phenotype[16,73] scoring was carried out over the course of 72 h. MCF10A cells stably expressing H2B-Dendra2 were seeded at a 15–25% confluence on μ-slide eight-well dishes (catalogue no. 80806; Ibidi), and images were acquired every 10 min with a Plan-Apochromat ×20/0.8 M27 air objective using the LSM 900 confocal microscope (Zeiss). Manual annotation was performed with the assistance of a customized tool written in Python. Mitotic phenotype and nuclear morphology for parental cells and the first generation of daughter cells were annotated as described in Fig. 1b.

## Optimization of photolabelling parameters

MCF10A cells stably expressing H2B-Dendra2 were seeded on μ-slides (Ibidi) and imaged on an LSM 900 confocal microscope (Zeiss). To determine Dendra2 photoconversion dynamics, we performed five bleaching rounds, each with ten laser-scanning iterations with a ×20 objective and 405-nm laser, at scanning speed 8 and power at 0.5% in the low-intensity power range. Images in green and red channels were acquired at the beginning and end of each round. The fluorescence intensity of ten photoconverted nuclei and five non-photoconverted control nuclei per field of view was quantified on manually defined ROIs with ImageJ. Data were then processed and analysed with custom Python scripts. To assess phototoxicity from targeted illumination, MCF10A and RPE-1 cells seeded on μ-slides (Ibidi) were photoconverted with settings used in the MAGIC pipeline and followed by confocal microscopy. Images for native and photoconverted Dendra2 fluorescence channels were acquired with a ×20 objective over the course of 24 h. Cells were tracked manually and their fate annotated. No cell death was detected for the photoconverted cells within the timeframe analysed.

## Single-cell genomic sequencing with Strand-seq

Unlike other single-cell genomic techniques, Strand-seq uniquely preserves haplotype identity across an entire homologue[27,29], which enables sensitive detection of simple and complex CA classes at intermediate sequence coverage[29,74]. We performed cell sorting as in the original procedure[27] with important adjustments to accept whole cells as input, to avoid loss of cytoplasmic DNA material and micronuclei during nuclei isolation. Cells were incubated with Hoechst 33342 (5 μg ml$^{-1}$) for 60 min, as it is cell membrane-permeable. Cells were then collected with 0.25% trypsin (Gibco) and resuspended in buffer (8% FBS in 1× PBS, supplemented with Hoechst 33342 5 μg ml$^{-1}$ and BrdU 40 μM). Single cells were sorted using a BD FACSAria in purity mode with a 100 or 130 μm nozzle, and dispensed into a flat-bottom 96-well plate (Thermo Fisher Scientific, Nunc plates) containing freeze buffer supplemented with 0.2% NP-40 (Thermo Fisher Scientific) to ensure membrane lysis and DNA accessibility in subsequent protocol steps. Strand-seq libraries were prepared at large-scale using a liquid handling robotic platform as described previously[29]. Libraries were sequenced on a NextSeq5000 (MID-mode, 75 bp paired-end) followed by demultiplexing. Reads were aligned to GRCh38 reference assembly with BWA-MEM v.0.7.17, yielding a median of ~285,000 mapped unique fragments per cell, and further processed as described below.

## Single-cell de novo CA discovery and classification

We discovered a wide variety of de novo CA classes leading to chromosomal or segmental copy-number imbalances by integrating read coverage and Watson/Crick template ratios[29], enabling high-resolution CA calling in Strand-seq data. Extending the functionality of the previously released MosaiCatcher tool[29], we designed strandtools, which is tailored for the specific task of handling de novo CA discovery in single cells under diverse ploidy backgrounds (Supplementary Methods). To achieve high confidence CA classification, we integrated read depth, strand orientation and haplotype information in each cell[29], to characterize segmental alterations and assign them to one of the following CA classes: chromosome loss, chromosome gain, interstitial loss, interstitial gain, terminal loss, terminal gain, terminal multi-step, complex CA and chromothripsis (a complex CA subclass). Chromosome gains and losses affect a whole chromosome, from p-ter telomere to q-ter telomere. Interstitial gains and losses are isolated CAs between two breakpoints, within one chromosome arm. As terminal alterations, we refer to all CAs that involve a portion of a chromosome, from a breakpoint anywhere along a chromosome arm to the telomere of that same arm. Therefore, terminal gains and losses are simple CAs, with one isolated, altered segment spanning from a breakpoint to the telomere of one chromosome arm. Terminal gains are annotated as inverted duplications if the gained segment is in opposite strand orientation compared with that of the original homologue with the same haplotype[29]. Terminal multi-step CAs are a sequential combination of gains and losses that are affecting the terminal portion of a chromosome arm. The terminal multi-step class also includes all cases of localized oscillations arising alongside terminal gains and losses.

Complex CAs are defined as events that include more than two breakpoints, can affect either one or both arms of the same homologue and can be composed of non-adjacent, altered segments. As such, complex CAs cannot be resolved as terminal multi-step. Chromothripsis events extending over large chromosomal regions, such as a chromosome arm, are included under the complex CA class. These events show characteristic copy-number oscillation between typically two copy-number states, affecting one single haplotype and with oscillating segments allowed in either strand orientation[29,30]. With regard to experiments on targeted DSB induction along chromosome arms, we likewise considered all copy-number imbalanced CA classes. In addition, we specified whole-arm alterations in the case of isolated gains and losses affecting more than 90% of a chromosome arm, and amplifications in case of isolated gains with a copy-number increment of two or more compared to the baseline. All single-cell CA annotations are available in Supplementary Tables 7, 8 and 9.

## Targeted induction of DSBs

CRISPR components, designed as described in the Supplementary Methods, were delivered in the form of RNP complex using a Neon Electric Transfection System (10 μl kit; catalogue no.: MPK1096; Thermo Fisher). First, the RNP complex was formed by incubating 0.3 μl of Alt-R S.p. Cas9 Nuclease (catalogue no.: 1081059; IDT) with 0.2 μl Resuspension Buffer R (Neon 10 μl kit) and 1 μl of designed sgRNA for 20 min. Cells (500,000 per reaction) were prepared for electroporation as described in the manufacturer's manual. Concentration of Cas9 nuclease in the final RNP/cell suspension was 1.5 μM, and that of sgRNA was 3.6 μM. Electroporation parameters of 1,400 V, 20 ms and two pulses were used for both RPE-1 and MCF10A cells. Transfected cells were diluted in antibiotic-free cell culture medium and different amounts (between 36,000 and 72,000) were seeded into four central wells of μ-slides containing 300 μl of antibiotic-free medium. The medium was replaced with fresh medium containing BrdU (40 μM) at 48 h post-transfection to allow cells to recover, and the slide was transferred immediately into the confocal microscope for imaging. For determining how DSB location may determine CA processes, we selected chromosome 2q due to its low average repeat content facilitating gRNA design, and 7q due to the enrichment for clonally propagated CAs we observe for this arm.

### Clone generation from single cells

Cells were subjected to automated photolabelling, collected with 0.25% trypsin (Gibco) and resuspended in buffer (8% FBS in 1× PBS). Single cells were sorted using a BD FACSAria at 1.0 flow rate, with a 130-μm nozzle to minimize cell damage, and dispensed into a flat-bottom 96-well plate (Thermo Fisher, Nunc plates) with normal growth medium. Formation of viable colonies was assessed visually daily with a phase-contrast microscope from day 7 to day 14 post sorting. At the 2-week mark, clones were transferred to six-well plates, and grown to confluence to be frozen for future experiments and prepared for sequencing.

### Low-pass WGS of clones

A total of 27 MCF10A cell pellets (18 clones deriving from micronucleated cells, nine control clones) and 11 RPE-1 cell pellets (11 clones deriving from micronucleated cells) were subjected to bulk-cell low-pass Illumina sequencing (NextSeq2000, P3, 100 bp paired-end sequencing) at EMBL's Genomics Core Facility, to an approximate genomic coverage of 1× for screening purposes. Reads were aligned to the GRCh38 genome reference with BWA[75], and read depth based CA calling was determined with support of the Control-FREEC tool[76]. A single case of a potential complex CAs was inferred on the basis of the chromosomal clustering of CAs inferred by read depth analysis.

### Long-read WGS of clone 7

Clone 7 was re-established to obtain $10 \times 10^6$ cells for Oxford Nanopore Technologies (ONT) long-read sequencing. The library was prepared using the SQK-LSK114 ligation kit, and sequencing performed on PromethION flow-cells. The obtained coverage was 16×, and the reads showed an estimated N50 of 13.97 kb. Reads were aligned to the GRCh38 genome reference with minimap2 (ref. 77). Structural variant calling was performed with Sniffles[78] and Delly[79], and calls were curated manually to exclude false positives. Read depth profiles for the micronucleated clone 7 were generated using delly (cnv subcommand) with a window size of 25 kb and the standard GRCh38 mappability map. The read depth signal was segmented using the DNAcopy Bioconductor package. Somatic structural variants were called using sniffles2 and delly (lr subcommand). For both delly and sniffles2, we used another clone of MCF10A as a control to filter for somatic variants in the micronucleated clone 7. Subsequently, only candidate somatic structural variants called by both methods and larger than 10 kb were used. Single-nucleotide variants, as well as small insertions and deletions (indels), were called using Clair3. Haplotype phasing of the ONT reads was performed with WhatsHap to generate read depth plots by haplotype[80]. Telogator[81] was used for telomere length inference from ONT reads generated from a MCF10A-derived clone ('clone 7'), using the suggested '-r ont' parameter recommended for handling Nanopore reads.

### Modelling de novo CA rates

We developed an agent-based model[82] to simulate CA acquisition in a growing population of cells, considering mitotic errors and micronuclei generation. During the simulation, cell agents are allowed to move between the states depicted in Fig. 3g. The probability $P_{ij}$ of transitioning from state $i$ to state $j$ is derived from long-term live-cell imaging experiments. Each cell agent is designed to possess three main attributes: cell cycle status, micronucleus status and CA status. The micronucleus status captures whether the cell possesses a micronucleus or not. The cell cycle status keeps track of an internal clock that simulates advancing through cell cycle until mitosis. Cell cycle duration is set at the median cell cycle duration measured in imaging experiments. The CA status captures whether the cell possesses a de novo CA. When the internal cell cycle clock reaches the end, mitosis or arrest occurs: the cell agent can move from interphase to a mitosis state (normal, laggard or bridge) or arrest. To simulate cell division, the current agent is moved to the arrest state and two new cells are generated and assigned

to state 1 or 5, according to the transition probability associated with that specific mitosis type. Moreover, during mitosis, each cell has the possibility of acquiring a de novo CA according to the assigned rate $R$. Arrested cells are then removed from the simulation. Each simulation is initiated with an initial population of 50 cells and is stopped when the population reaches size 50,000, as we found empirically that the micronuclei and CA frequency usually stabilize by this time. Encouragingly, despite not being programmed explicitly into the model, the frequency of micronuclei stabilized at 5.0% and 38.3% for wild-type and $TP53^{-/-}$ cells respectively, closely mirroring our empirical data (Supplementary Fig. 6a,b). At the end of the simulation, we compute the sum of squared error between the simulated and target de novo CA frequencies. Details on how bound-constrained minimization was used to the CA rate estimation are the Supplementary Methods.

### Fluorescent in situ hybridization

MCF10A cells were seeded on coverglass slides, subjected to targeted DSB induction, and allowed to recover for 48 h. Metaphase spreads were then prepared in situ directly on coverslips, as described elsewhere[83]. Sub-centromeric probes for Chr. 7 p and q arms were purchased from KromaTiD (Biocat catalogue no.: CEP-0013-C-KTD, CEP-0014-A-KTD). FISH was performed according to manufacturer instructions. After post-hybridization washes, DNA was stained with Hoechst 33342 and slides were mounted in anti-fade medium (Vectashield, Vector Laboratories). FISH images were acquired on a LSM 900 confocal microscope (Zeiss) at ×40 magnification and signals were evaluated visually.

### Reporting summary

Further information on research design is available in the Nature Portfolio Reporting Summary linked to this article.

## Data availability

All genomics data generated in this study (Strand-seq, as well as short and long-read bulk WGS) are available at ENA under the following accession: PRJEB78885. Strand-seq processed count data are publicly available at Zenodo (https://doi.org/10.5281/zenodo.15262423)[84]. We re-analysed publicly available data from the PCAWG[8] and TCGA resources to compare our findings to those previously made in cancer genomes. The raw WGS data generated by TCGA can be accessed through controlled data access application using dbGAP under study accession code phs000178. Data links are available in Supplementary Table 17.

## Code availability

The software automation components and step-by-step instructions for running MAGIC experiments are available in the magic_automation repository (https://git.embl.de/cosenza/magic_automation). For image analysis, computer vision, and image processing, visit the magic_tools repository (https://git.embl.de/cosenza/magic_tools). Tools for analysing Strand-seq data and single-cell copy-number calling are provided in the strandtools repository (https://git.embl.de/cosenza/strandtools). A Docker container providing a unified environment to run the main computational pipeline behind MAGIC (strandtools, magic_tools and magic_automation) is available at https://git.embl.de/tweber/magic-container. The script for estimating basal CA rates is provided at https://git.embl.de/cosenza/ca_rates_estimation. A snapshot of all software repositories for MAGIC experiments and Strand-seq data analysis has been archived and is publicly available at Zenodo (https://doi.org/10.5281/zenodo.16631215)[85]. The code used in this study to analyse WGS data is available at: https://github.com/cortes-ciriano-lab/osteosarcoma_evolution. Software repository links are available in Supplementary Table 17.

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

**Acknowledgements** We thank D. Pellman and C.-Z. Zhang for providing valuable comments on an advanced version of our manuscript. *TP53*-/- MCF10A cells are a kind gift from C. Scholl. We acknowledge support from C. Hain in the telomere length analysis. Principal funding for this work came from the European Research Council (ERC Advanced grant (SEE-MAGIC) grant no. 101098056) to J.O.K., with additional support coming from an ERC Consolidator grant (MOSAIC; no. 773026) to J.O.K, the Volkswagen Foundation (VW–95826) to J.O.K, the Health + Life Science Alliance Heidelberg Mannheim with funding approved by the State Parliament of Baden-Württemberg, and from EMBL core funding. M.R.C. received support from the EMBL Interdisciplinary Postdoc (EIPOD4) program 4 under Marie Skłodowska-Curie Actions COFUND (grant agreement no. 847543), enabling interdisciplinary studies in the Pepperkok and Korbel groups. We acknowledge the EMBL core facilities and services for support in high-performance computing (IT), sequencing (GeneCore), chemical synthesis (Chemical Biology), imaging (Advanced light microscopy) and cell sorting (Flow cytometry core facility) and the German Cancer Research Centre genomics core facility.

**Author contributions** M.R.C., J.O.K. and R.P. conceived the project, with J.O.K. providing scientific direction and supervision. M.R.C. and A.H. developed the microscope automation software and integration with AutoMicTools. M.R.C. developed magic_tools. T.W. integrated the software components into a Docker image. M.R.C. designed and performed the photolabelling optimization experiments. M.R.C. and A.G. designed and performed the long-term live-cell imaging experiments, M.R.C., A.G. and N.L.S. analysed the image data. N.L.S. performed Western blotting experiments. M.R.C., A.G., N.L.S. and M.S. designed and performed MAGIC experiments with support from P.H., C.S and J.L.E. A.G. performed single-cell clone propagation experiments, low-pass WGS and long-read sequencing characterization of clones. Long-read data were analysed by A.G. and T.R. M.R.C. designed the FISH experiments, which were performed by A.G. M.A.J. developed the convolutional neural network. P.H., E.B. and C.S. prepared Strand-seq libraries. Single-cell data were analysed by M.R.C., A.G., N.L.S., A.A. and J.O.K. A.A. designed and performed copy-number pattern analysis in PCAWG data, with support from M.R.C. A.A. and S.Z. designed and performed the isochromosome analysis in primary cancer genomes, with guidance from J.O.K. and I.C.-C. B.E.U. designed and tested the CRISPR guides, under guidance of A.E.K. CRISPR experiments were performed by B.E.U. and A.G. M.R.C. developed the agent-based statistical model with support from E.-M.G. M.R.C. and J.O.K. wrote the core of the manuscript, with contributions from all authors. J.O.K., R.P., A.E.K., T.R., A.G., M.S., N.L.S., A.A., S.Z. and I.C.-C. critically reviewed and edited the manuscript.

**Funding** Open access funding provided by European Molecular Biology Laboratory (EMBL).

**Competing interests** J.O.K. has previously disclosed a patent application (no. EP19169090) that is relevant to the use of Strand-seq for somatic structural variation analysis. The other authors declare no competing interests.

**Additional information**
**Correspondence and requests for materials** should be addressed to Jan O. Korbel.

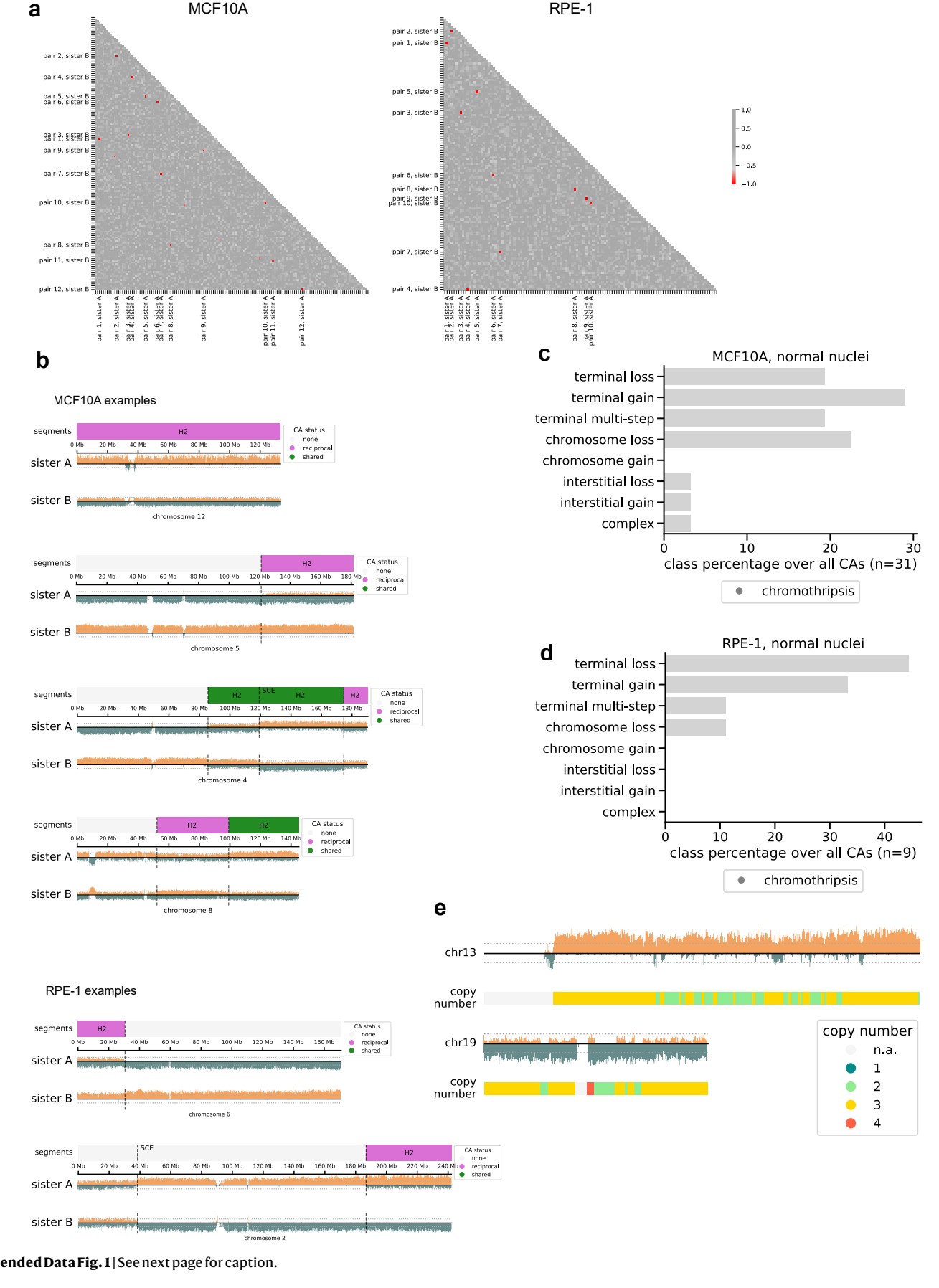

**Extended Data Fig. 1** | See next page for caption.

**Extended Data Fig. 1 | Sister cell analysis and CA examples in near-diploid non-transformed cells.** (a) Sister cell strand-state anti-correlation analysis for MCF10A (left) and RPE-1 (right) micronucleated cells (Pearson correlation coefficient) – revealing several sister cell pairs through single-cell genomic analysis. (b) Reciprocal CA examples in MCF10A and RPE-1 cells, with annotated segment boundaries marked by dashed lines. (c,d) Breakdown of CA classes in normal MCF10A (c) and RPE-1 (d) cells, with the class percentage shown relative to all CAs. (e) Chromothripsis examples from spontaneous micronuclei in MCF10A cells.

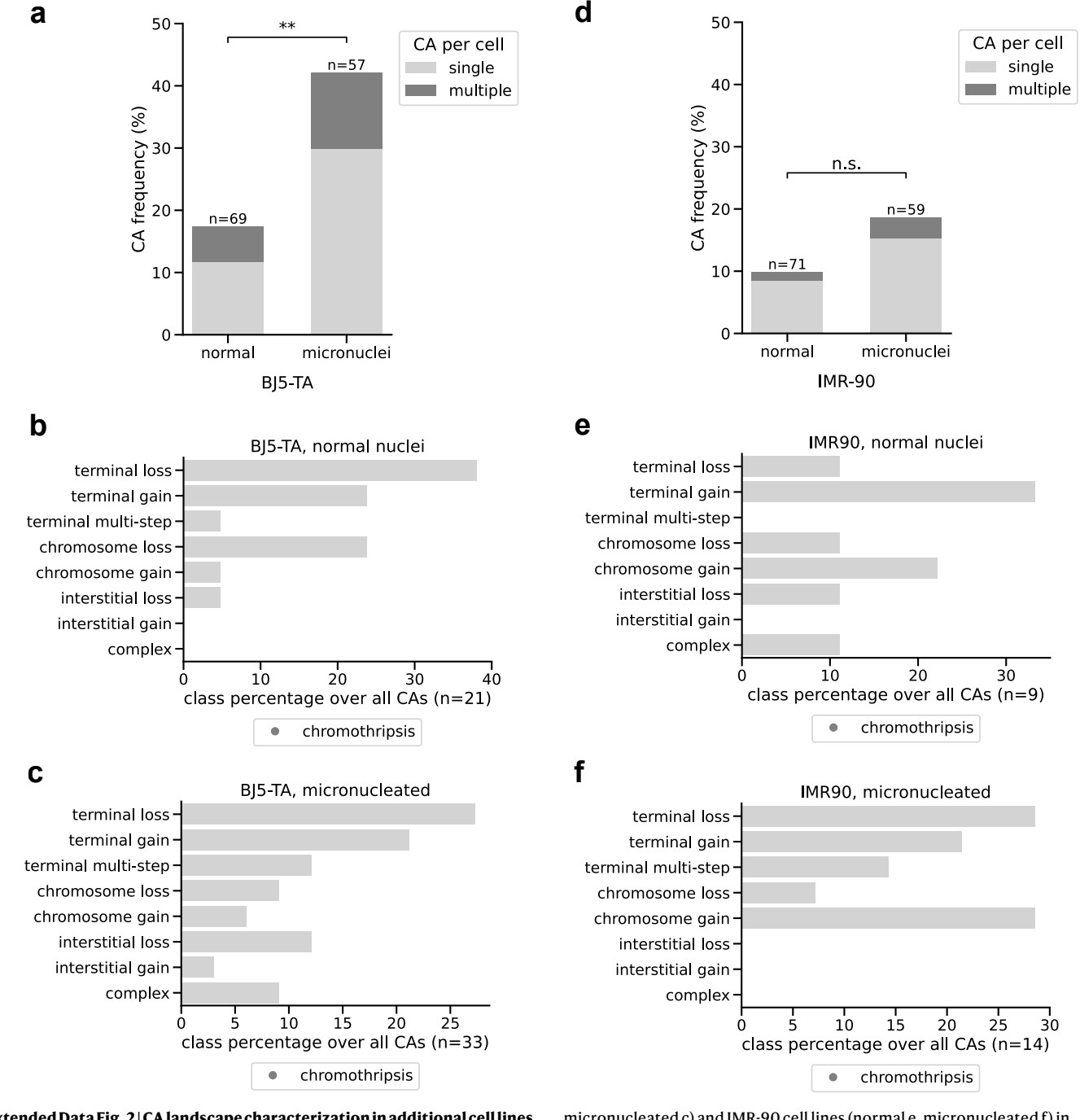

**Extended Data Fig. 2 | CA landscape characterization in additional cell lines.** CAs detected per cell in BJ-5ta (a) and IMR-90 (d) cell lines (without perturbation or treatment, Fisher's exact test). Breakdown of CA classes in the BJ-5ta (normal b, micronucleated c) and IMR-90 cell lines (normal e, micronucleated f) in absence of perturbation. See Methods section for CA classification criteria.

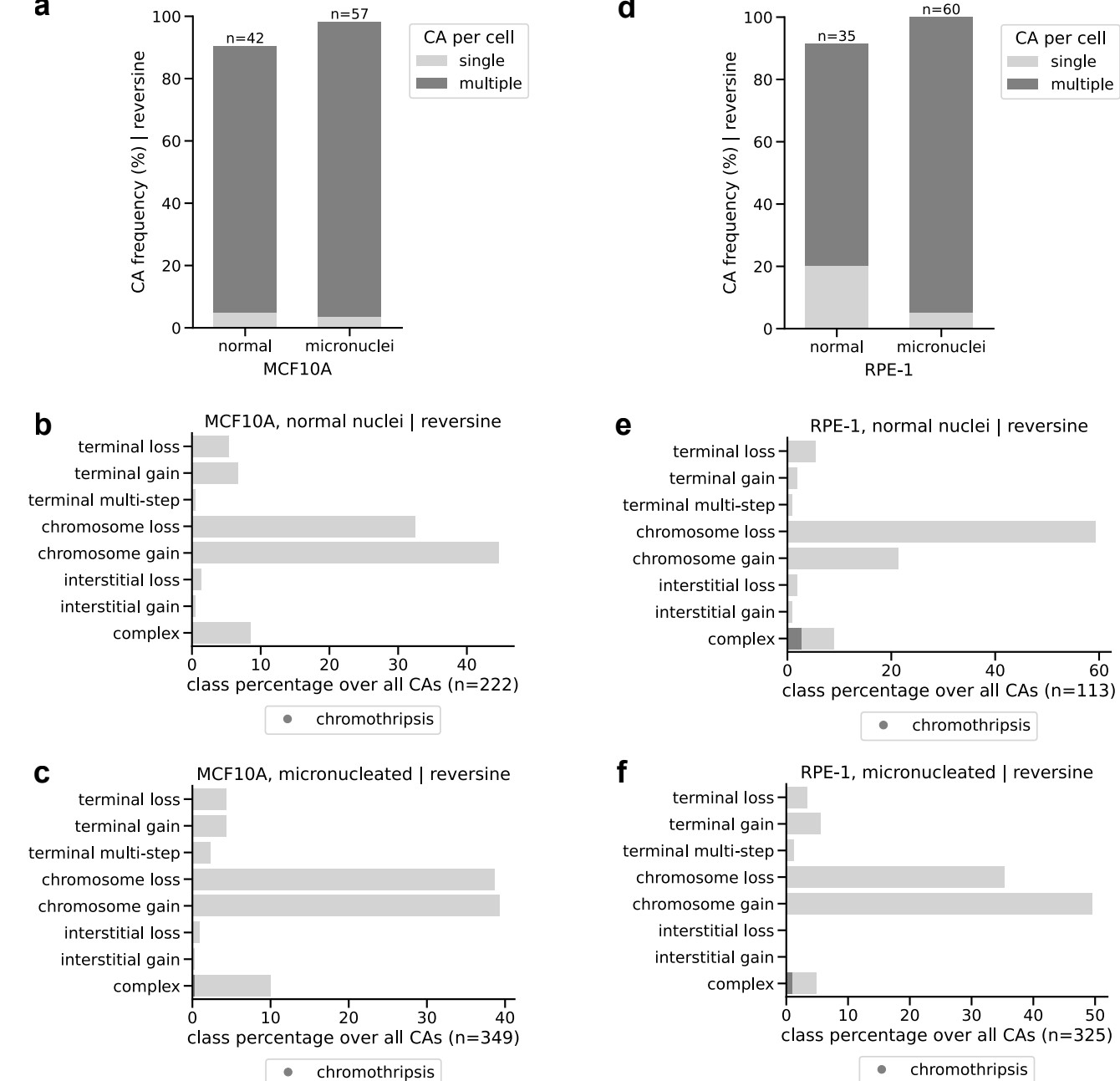

**Extended Data Fig. 3 | CA landscape characterization following perturbation of chromosome segregation.** CAs detected per cell in MCF10A (a) and RPE-1 (d) cell lines after reversine treatment (Methods). Breakdown of CA classes in the MCF10A (normal b, micronucleated c) and RPE-1 (normal e, micronucleated f) following treatment with reversine. See Methods section for CA classification criteria.

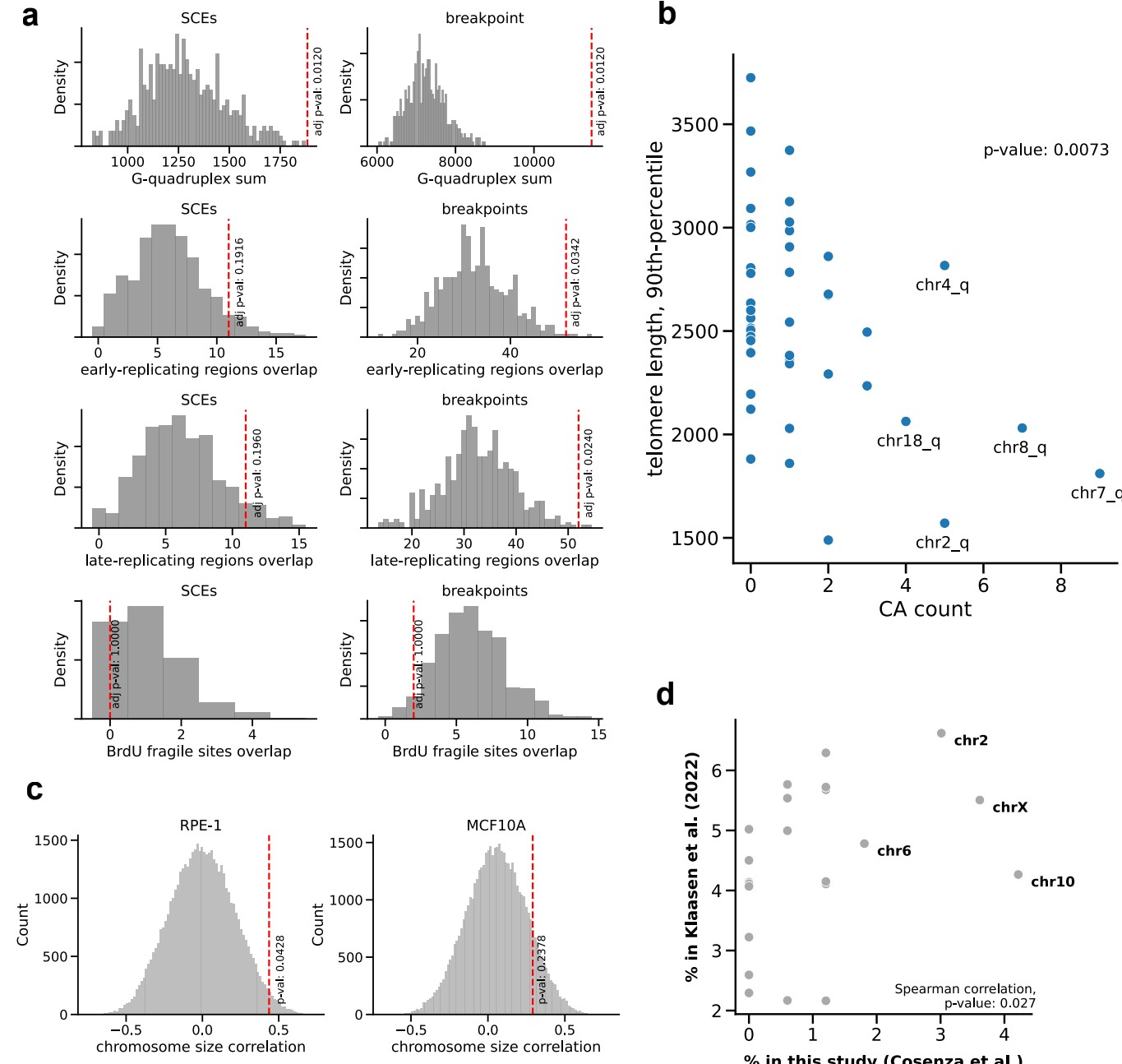

**Extended Data Fig. 4 | Breakpoint features of CAs and chromosome size correlation results from spontaneous micronucleation experiments.** (a) Distribution derived from permutation of breakpoint location against G-quadruplex sites (top row), early (second row) and late (third row) replicating regions, as well as BrdU fragile sites (bottom row). Red dashed line: observed statistic (see Methods). The analysis was conducted using MCF10A cells, from spontaneously micronucleated cells. (b) Correlation between CA count and ONT-sequencing based telomere length per chromosome arm for MCF10A, expressed as 90th percentile of read length (Pearson correlation; see Methods).

(c) Distribution derived from permutation of determined CA numbers in spontaneously micronucleated cells against chromosome size in RPE-1 and MCF10A. Red dashed line: observed statistic. (d) Percentage of aneuploid chromosomes in RPE-1 across studies. Correlation in aneuploid chromosome percentages in RPE-1 cells as measured from spontaneous micronucleation (our study) and after chemically induced missegregation from a prior study[32]. Dots represent chromosome-specific aberration frequencies in both studies (Spearman correlation).

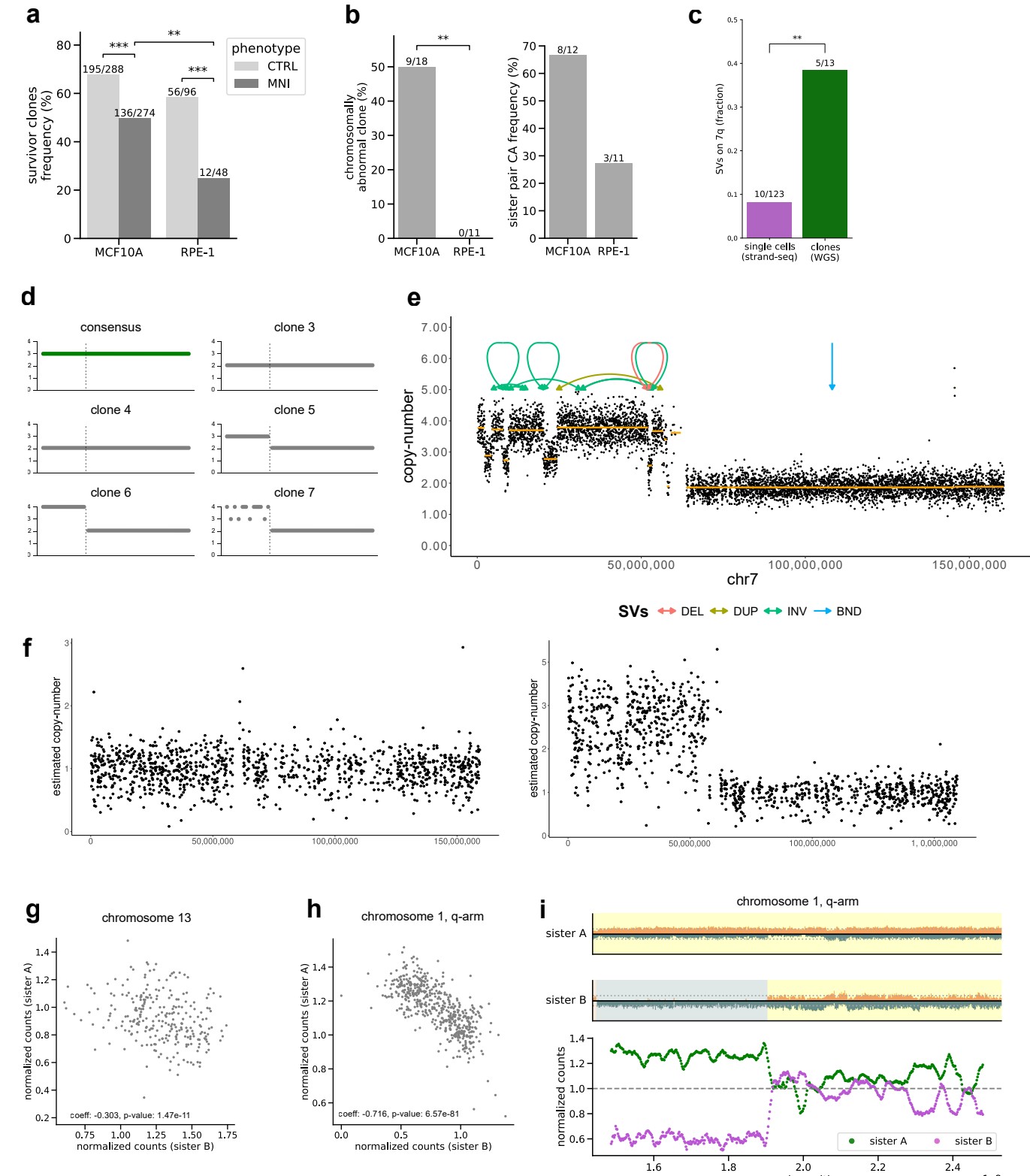

**Extended Data Fig. 5** | See next page for caption.

**Extended Data Fig. 5 | Analysis of chromosome-scale chromothripsis, and clonal propagation experiments.** (a) Fraction of surviving clones after MAGIC isolation (Fisher's exact test). (b) De novo CA formation in two cell lines measured from chromosomally abnormal propagated clones as well as sister cell pairs. Left, percentage of clones carrying at least one de novo CA, originating from micronucleated cells. Right, frequency of micronucleated sister cell pairs carrying at least one de novo CA (Fisher's exact test). (c) 7q loss enrichment in propagated clones (Fisher's exact test). (d) Schematic of 7q hits in micronucleated clones, which include 7q-arm loss, chromosome 7 losses, and isochromosomes resulting in 7q-loss. (e) ONT long-read sequencing based copy-number plot of chromosome 7 for the micronucleated clone 7; arrows on top represent the boundaries of the different classes of SVs (DEL: deletion; DUP: duplication; INS: insertion; INV: inversion; BND: translocation, see Methods). The chromosome presents a duplication of the p-arm coupled to the deletion of the q-arm, indicating isochromosome formation; in addition, a chromothripsis event is identified based on the presence of an oscillating copy-number pattern (in this case, between copy-number 3 (CN3) and CN4) and on the simultaneous occurrence of multiple SVs on the affected chromosome arm consistent with randomness of DNA fragment joins. As a consequence of isochromosome formation followed by chromothripsis, three copy-number states are seen across the chromosome 7 genomic coordinates. (f) Copy-number plot of chromosome 7 of the micronucleated clone 7, resolved by haplotype. The copy-number alterations seen are confined to haplotype 2 (see Methods). (g,h) Sister cell read-count anti-correlation for chromosome 13 (g) and chromosome 1 q-arm (h) (Pearson correlation coefficient), verifying the reciprocal segregation of shattered DNA fragments. (i) Chromothripsis in a sister cell pair, affecting chromosome 1, q-arm, determined in a complex ploidy background. Top, Strand-seq plots with oscillatory copy-number pattern. Bottom, smoothened and normalised counts along chromosomal positions.

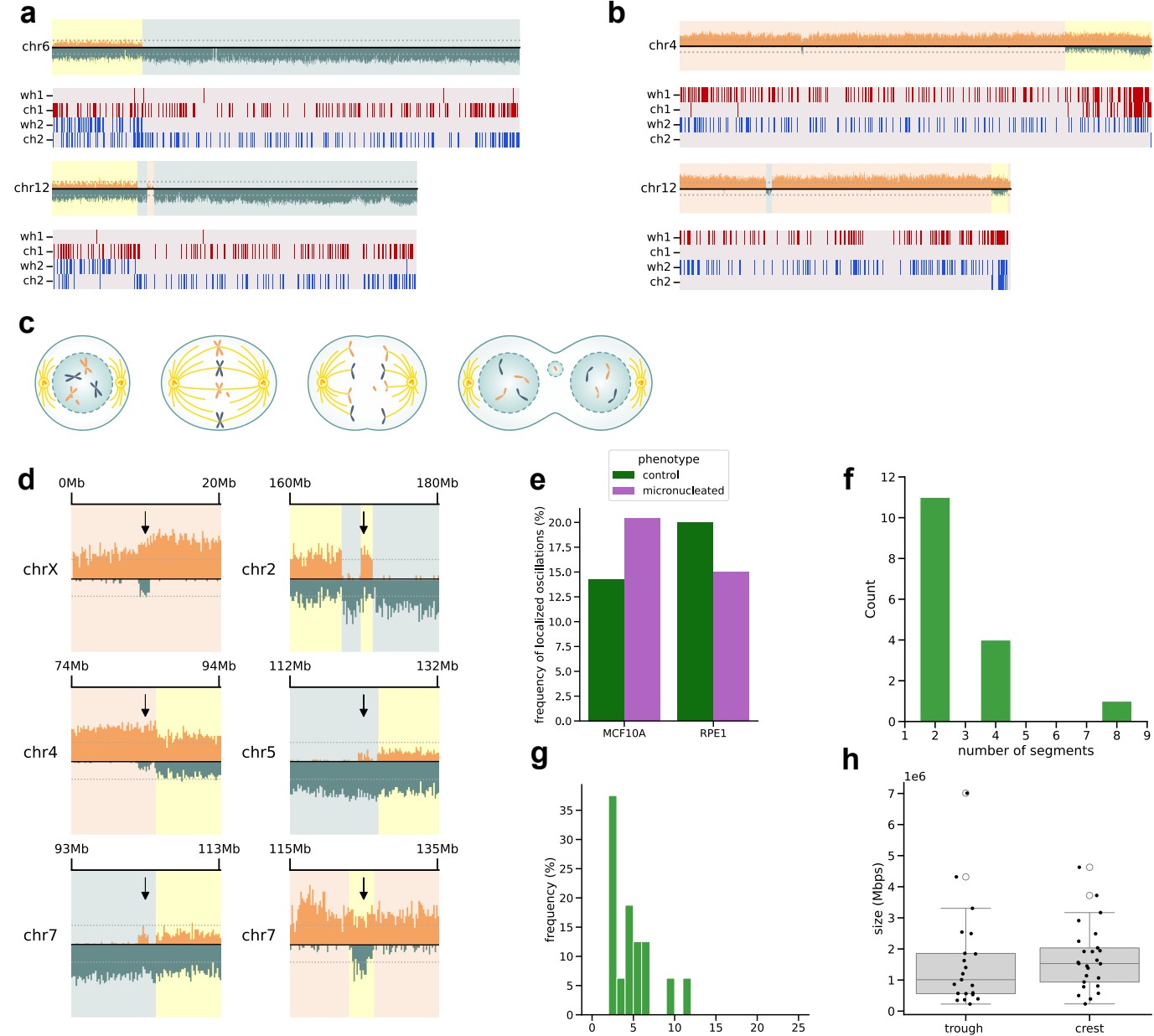

**Extended Data Fig. 6 | Analysis of spontaneous bridge-mediated CAs.**
(a,b) Strand-seq based haplotype-resolved examples of terminal inverted duplications (a) and terminal multi-step alterations (b). For each example, we depict at the top: Strand-seq plot, bottom: haplotag[29] localisation. In (a), inverted terminal duplications are characterised by the presence of haplotype 2 (H2) haplotags on the W strand for both chromosome 6 and 12 examples. In the (b) upper panel, chromosome 4 carries two adjacent copy-number gains, affecting haplotype 1 (H1) and both strands. In the lower panel, chromosome 12 carries one inverted duplication on H2, with an adjacent terminal deletion of the

sample haplotype. (c) Scheme showing mitosis entry with unrepaired DSB leading to micronucleus formation. (d) Localised copy-number oscillation examples indicative for small to intermediate scale complex SVs. Arrow: oscillation peak. (e) Frequency of the localised oscillation pattern in MCF10A and RPE-1 cells. (f,g,h) Features of the localised oscillation pattern: number of segments composing the oscillation (f), overall oscillation (g), trough and crest (h) oscillation size. The data depicted includes cases from both WT and TP53-/- cell line models. Center line, median; box limits, upper and lower quartiles; whiskers, 1.5x interquartile range; points, outliers.

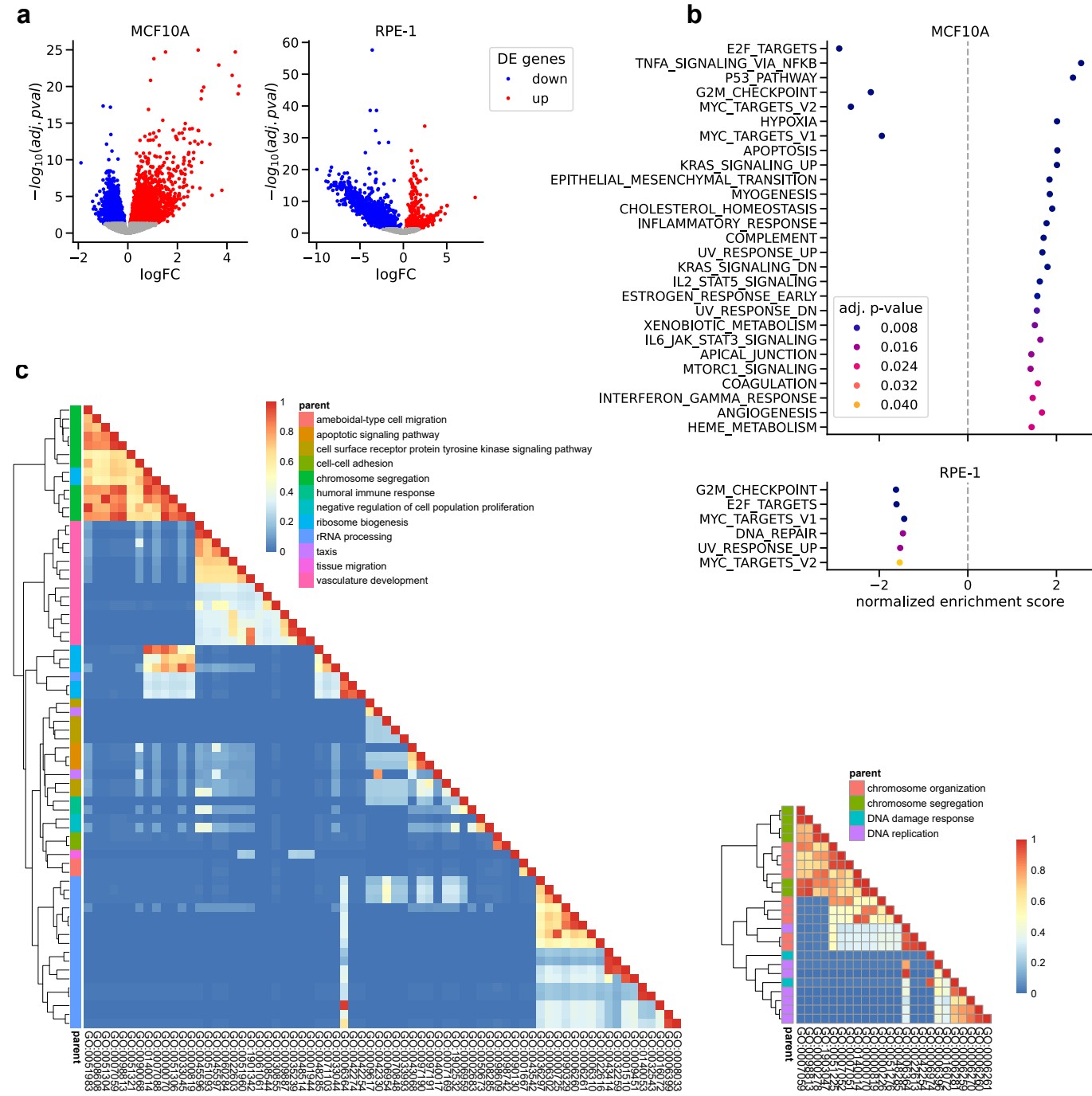

**Extended Data Fig. 7 | Differential expression analysis of micronucleated cells.** (a) Volcano plots of differentially-expressed genes identified by coupling the MAGIC platform with single-cell RNA sequencing, contrasting normal vs. micronucleated cells in MCF10A (left panel) and RPE-1 (right panel) wild-type cells. Differentially expressed (DE) genes are in blue if down-regulated or red if up-regulated. (b) Enrichment analysis for Hallmark gene sets from the Human Molecular Signatures Database (MSigDB). Upper panel MCF10A, lower panel RPE-1. (c) Heatmap plot summarizing gene ontology (GO) enrichment for MCF10A (left panel) and RPE-1 (right panel). Redundant GO terms were clustered by semantic similarity, indicated by the blue to red heatmap gradient.

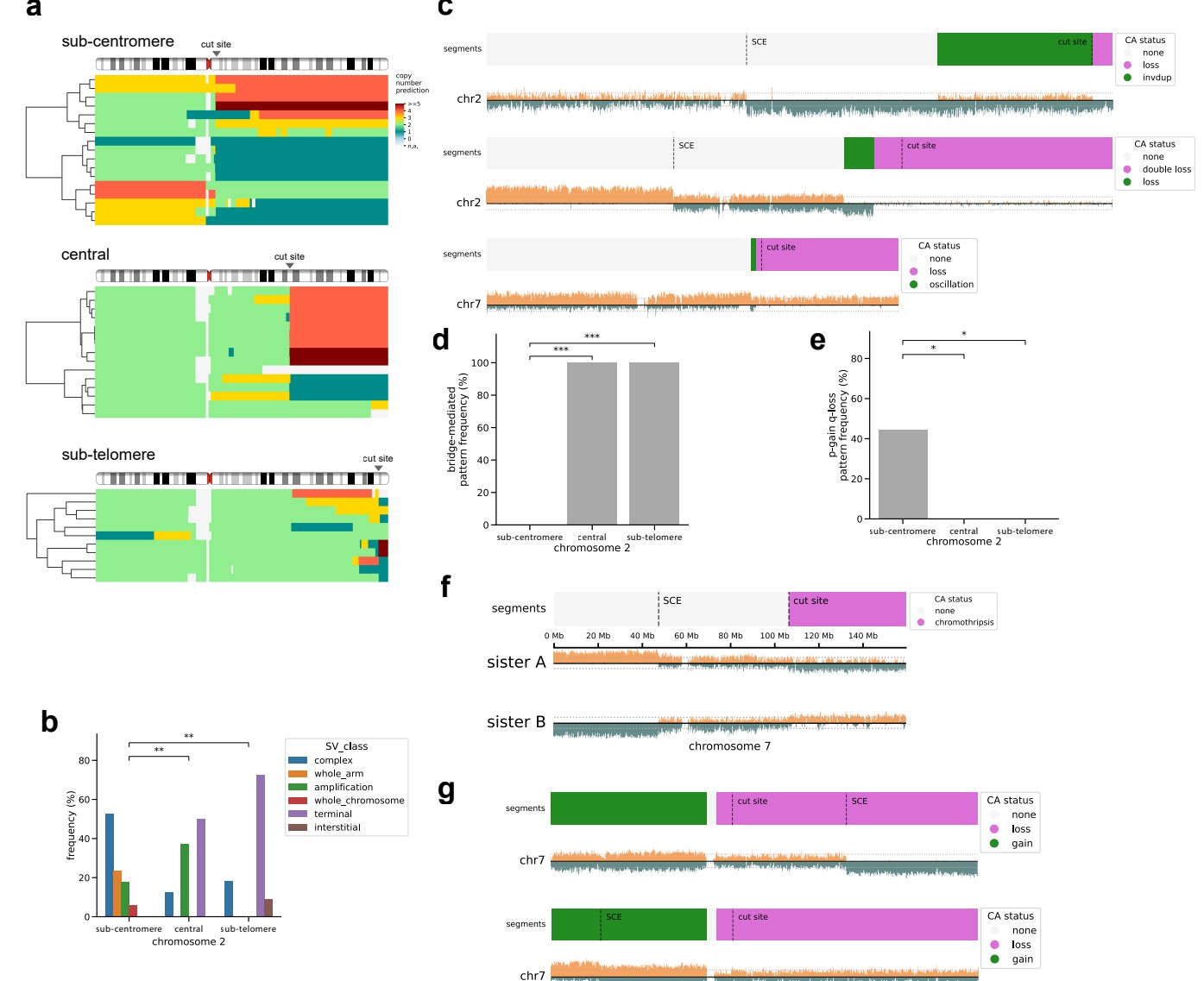

**Extended Data Fig. 8 | Detailed characterisation of CAs arising from targeted DSB induction.** (a) Overview of copy-number changes for chromosome 2 q-arm CAs, shown for sub-centromere (top), central (middle) and sub-telomere (bottom) target loci. (b) Types of CA observed for different cut sites on the chromosome 2 q-arm (Fisher's exact test). (c) Terminal multi-step CAs on targeted chromosomal arms. (d,e) Significant enrichment for bridge-mediated (d) and isochromosome-like (e) CA patterns, depending on the cut location. We considered all terminal and complex CAs in this analysis (P-values are based on Fisher's exact test). (f) Chromothripsis in sister cells, involving targeted DSBs at the central cut site. (g) Examples of inferred isochromosomes. Dashed lines correspond to SCEs and targeted cut sites. The chromosome 7 consensus copy-number of the MCF10 cell line is three.

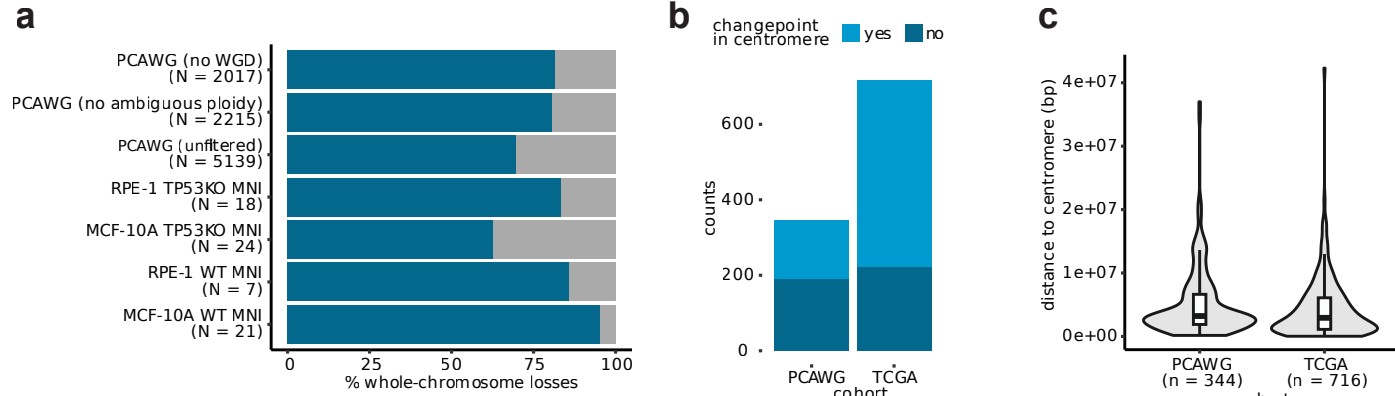

**Extended Data Fig. 9 | Aneuploidy and isochromosome analysis in primary cancer genome datasets.** (a) Proportion of whole-chromosome losses among all whole-chromosome aneuploidy events in spontaneous micronuclei models and the PCAWG[8] dataset. The "unfiltered" PCAWG dataset includes all representative aliquots (see Supplementary Notes). The "no ambiguous ploidy" dataset only includes those aliquots in which an integer ploidy could be assigned unambiguously (see Supplementary Notes). The "no WGD" dataset excludes aliquots affected by whole genome duplication (WGD). The total number of whole-chromosome gains and losses is shown for each dataset. MNI: micronuclei; TP53KO: knockout of TP53 gene; WT: wild type, for normal TP53 status. (b,c) Number of inferred isochromosomes (b) and distribution of distances between isochromosome changepoints and centromeres (c) identified in the TCGA and PCAWG datasets, considering only non-redundant donors (see Supplementary Notes). Center line, median; inner box limits, upper and lower quartiles; whiskers, 1.5x interquartile range; gray patch, kernel density estimate.

# Reporting Summary

## Statistics

For all statistical analyses, confirm that the following items are present in the figure legend, table legend, main text, or Methods section.

| n/a | Confirmed | |
|---|---|---|
| ☐ | ☒ | The exact sample size (*n*) for each experimental group/condition, given as a discrete number and unit of measurement |
| ☐ | ☒ | A statement on whether measurements were taken from distinct samples or whether the same sample was measured repeatedly |
| ☐ | ☒ | The statistical test(s) used AND whether they are one- or two-sided<br>*Only common tests should be described solely by name; describe more complex techniques in the Methods section.* |
| ☐ | ☒ | A description of all covariates tested |
| ☐ | ☒ | A description of any assumptions or corrections, such as tests of normality and adjustment for multiple comparisons |
| ☐ | ☒ | A full description of the statistical parameters including central tendency (e.g. means) or other basic estimates (e.g. regression coefficient) AND variation (e.g. standard deviation) or associated estimates of uncertainty (e.g. confidence intervals) |
| ☐ | ☒ | For null hypothesis testing, the test statistic (e.g. $F$, $t$, $r$) with confidence intervals, effect sizes, degrees of freedom and $P$ value noted<br>*Give P values as exact values whenever suitable.* |
| ☒ | ☐ | For Bayesian analysis, information on the choice of priors and Markov chain Monte Carlo settings |
| ☒ | ☐ | For hierarchical and complex designs, identification of the appropriate level for tests and full reporting of outcomes |
| ☐ | ☒ | Estimates of effect sizes (e.g. Cohen's *d*, Pearson's *r*), indicating how they were calculated |

*Our web collection on statistics for biologists contains articles on many of the points above.*

## Software and code

Policy information about availability of computer code

| Data collection | Python 3.11.5<br>ImageJ 1.53e,<br>ZEN Microscopy Software<br>MAGIC https://git.embl.de/cosenza/magic_automation<br>AutoMicTools https://git.embl.de/halavaty/AutoMicTools<br>Specific python packages are available through the pip requirement files in the relevant repositories |
|---|---|
| Data analysis | Python 3.11.5<br>R v4.1.1<br>Mosaicatcher https://github.com/friendsofstrandseq/mosaicatcher-pipeline<br>strandtools https://git.embl.de/cosenza/strandtools<br>https://git.embl.de/cosenza/ca_rates_estimation<br>https://github.com/cortes-ciriano-lab/osteosarcoma_evolution<br>Delly 1.2.6 https://github.com/dellytools/delly<br>Sniffles 2.2<br>Whatshap 2.2<br>Specific python packages are available through the pip requirement files in the relevant repositories |

For manuscripts utilizing custom algorithms or software that are central to the research but not yet described in published literature, software must be made available to editors and reviewers. We strongly encourage code deposition in a community repository (e.g. GitHub). See the Nature Portfolio guidelines for submitting code & software for further information.

## Data

Policy information about availability of data

All manuscripts must include a data availability statement. This statement should provide the following information, where applicable:

- Accession codes, unique identifiers, or web links for publicly available datasets
- A description of any restrictions on data availability
- For clinical datasets or third party data, please ensure that the statement adheres to our policy

All genomics data generated in this study (Strand-seq, as well as short and long-read bulk WGS) are available under the following accession: PRJEB78885. We re-analysed publicly available data from the PCAWG and TCGA resources to compare our findings to those previously made in cancer genomes. The raw WGS data generated by The Cancer Genome Atlas (TCGA) can be accessed through controlled data access application via dbGAP under study accession code phs000178.

## Research involving human participants, their data, or biological material

Policy information about studies with human participants or human data. See also policy information about sex, gender (identity/presentation), and sexual orientation and race, ethnicity and racism.

| | |
|---|---|
| Reporting on sex and gender | *Use the terms sex (biological attribute) and gender (shaped by social and cultural circumstances) carefully in order to avoid confusing both terms. Indicate if findings apply to only one sex or gender; describe whether sex and gender were considered in study design; whether sex and/or gender was determined based on self-reporting or assigned and methods used. Provide in the source data disaggregated sex and gender data, where this information has been collected, and if consent has been obtained for sharing of individual-level data; provide overall numbers in this Reporting Summary. Please state if this information has not been collected. Report sex- and gender-based analyses where performed, justify reasons for lack of sex- and gender-based analysis.* |
| Reporting on race, ethnicity, or other socially relevant groupings | *Please specify the socially constructed or socially relevant categorization variable(s) used in your manuscript and explain why they were used. Please note that such variables should not be used as proxies for other socially constructed/relevant variables (for example, race or ethnicity should not be used as a proxy for socioeconomic status). Provide clear definitions of the relevant terms used, how they were provided (by the participants/respondents, the researchers, or third parties), and the method(s) used to classify people into the different categories (e.g. self-report, census or administrative data, social media data, etc.) Please provide details about how you controlled for confounding variables in your analyses.* |
| Population characteristics | *Describe the covariate-relevant population characteristics of the human research participants (e.g. age, genotypic information, past and current diagnosis and treatment categories). If you filled out the behavioural & social sciences study design questions and have nothing to add here, write "See above."* |
| Recruitment | *Describe how participants were recruited. Outline any potential self-selection bias or other biases that may be present and how these are likely to impact results.* |
| Ethics oversight | *Identify the organization(s) that approved the study protocol.* |

Note that full information on the approval of the study protocol must also be provided in the manuscript.

# Field-specific reporting

Please select the one below that is the best fit for your research. If you are not sure, read the appropriate sections before making your selection.

☒ Life sciences    ☐ Behavioural & social sciences    ☐ Ecological, evolutionary & environmental sciences

For a reference copy of the document with all sections, see nature.com/documents/nr-reporting-summary-flat.pdf

# Life sciences study design

All studies must disclose on these points even when the disclosure is negative.

| | |
|---|---|
| Sample size | We aimed to strike a balance between available funding for sequencing and the need for comprehensive sequencing of single-cell genomes. Our design aimed at maximizing the identification of the most pronounced determinants while remaining experimentally feasible. We typically generate 96-well plates for the initial screenings from our automated Strand-seq library prepration set up, which dictates our batch size and ensures that we have sufficient coverage for robust analysis. In the estimation of sample size, we anticipated some sample loss during the process, as only single cells of acceptable quality will be selected for further analysis. |
| Data exclusions | We excluded low quality single-cell libraries that showed very low (<100,000 unique reads), uneven coverage, or an excess of 'background reads' yielding noisy Strand-seq data prior to analysis. |
| Replication | The overall number of samples collected in this study confirms the high robustness and reproducibility of the MAGIC platform. To ensure the reliability of our findings, we used two different cell lines and two isogenic p53 models. For conditions involving micronucleation, we aimed to sequence two plates whenever possible. Additionally, we verified our observations through analyses in sister cells, sequencing of clonally |

| | |
|---|---|
| | propagated cells, and by performing FISH. |
| Randomization | This aspect is not applicable in our study. For collection of normal and micronucleated cells, we subjected all cell lines to the same treatments and experiments. For targeted-DSB experiments, we focused on MCF10A cells as this cell line was more amenable to experimental manipulation by CRISPR and MAGIC. |
| Blinding | During analysis and annotation of the data, investigators were blinded whenever possible, with regards to experimental conditions |

# Behavioural & social sciences study design

All studies must disclose on these points even when the disclosure is negative.

| | |
|---|---|
| Study description | *Briefly describe the study type including whether data are quantitative, qualitative, or mixed-methods (e.g. qualitative cross-sectional, quantitative experimental, mixed-methods case study).* |
| Research sample | *State the research sample (e.g. Harvard university undergraduates, villagers in rural India) and provide relevant demographic information (e.g. age, sex) and indicate whether the sample is representative. Provide a rationale for the study sample chosen. For studies involving existing datasets, please describe the dataset and source.* |
| Sampling strategy | *Describe the sampling procedure (e.g. random, snowball, stratified, convenience). Describe the statistical methods that were used to predetermine sample size OR if no sample-size calculation was performed, describe how sample sizes were chosen and provide a rationale for why these sample sizes are sufficient. For qualitative data, please indicate whether data saturation was considered, and what criteria were used to decide that no further sampling was needed.* |
| Data collection | *Provide details about the data collection procedure, including the instruments or devices used to record the data (e.g. pen and paper, computer, eye tracker, video or audio equipment) whether anyone was present besides the participant(s) and the researcher, and whether the researcher was blind to experimental condition and/or the study hypothesis during data collection.* |
| Timing | *Indicate the start and stop dates of data collection. If there is a gap between collection periods, state the dates for each sample cohort.* |
| Data exclusions | *If no data were excluded from the analyses, state so OR if data were excluded, provide the exact number of exclusions and the rationale behind them, indicating whether exclusion criteria were pre-established.* |
| Non-participation | *State how many participants dropped out/declined participation and the reason(s) given OR provide response rate OR state that no participants dropped out/declined participation.* |
| Randomization | *If participants were not allocated into experimental groups, state so OR describe how participants were allocated to groups, and if allocation was not random, describe how covariates were controlled.* |

# Ecological, evolutionary & environmental sciences study design

All studies must disclose on these points even when the disclosure is negative.

| | |
|---|---|
| Study description | *Briefly describe the study. For quantitative data include treatment factors and interactions, design structure (e.g. factorial, nested, hierarchical), nature and number of experimental units and replicates.* |
| Research sample | *Describe the research sample (e.g. a group of tagged Passer domesticus, all Stenocereus thurberi within Organ Pipe Cactus National Monument), and provide a rationale for the sample choice. When relevant, describe the organism taxa, source, sex, age range and any manipulations. State what population the sample is meant to represent when applicable. For studies involving existing datasets, describe the data and its source.* |
| Sampling strategy | *Note the sampling procedure. Describe the statistical methods that were used to predetermine sample size OR if no sample-size calculation was performed, describe how sample sizes were chosen and provide a rationale for why these sample sizes are sufficient.* |
| Data collection | *Describe the data collection procedure, including who recorded the data and how.* |
| Timing and spatial scale | *Indicate the start and stop dates of data collection, noting the frequency and periodicity of sampling and providing a rationale for these choices. If there is a gap between collection periods, state the dates for each sample cohort. Specify the spatial scale from which the data are taken* |
| Data exclusions | *If no data were excluded from the analyses, state so OR if data were excluded, describe the exclusions and the rationale behind them, indicating whether exclusion criteria were pre-established.* |
| Reproducibility | *Describe the measures taken to verify the reproducibility of experimental findings. For each experiment, note whether any attempts to repeat the experiment failed OR state that all attempts to repeat the experiment were successful.* |
| Randomization | *Describe how samples/organisms/participants were allocated into groups. If allocation was not random, describe how covariates were controlled. If this is not relevant to your study, explain why.* |

| Blinding | *Describe the extent of blinding used during data acquisition and analysis. If blinding was not possible, describe why OR explain why blinding was not relevant to your study.* |
|---|---|

Did the study involve field work?  ☐ Yes  ☐ No

## Field work, collection and transport

| Field conditions | *Describe the study conditions for field work, providing relevant parameters (e.g. temperature, rainfall).* |
|---|---|
| Location | *State the location of the sampling or experiment, providing relevant parameters (e.g. latitude and longitude, elevation, water depth).* |
| Access & import/export | *Describe the efforts you have made to access habitats and to collect and import/export your samples in a responsible manner and in compliance with local, national and international laws, noting any permits that were obtained (give the name of the issuing authority, the date of issue, and any identifying information).* |
| Disturbance | *Describe any disturbance caused by the study and how it was minimized.* |

# Reporting for specific materials, systems and methods

We require information from authors about some types of materials, experimental systems and methods used in many studies. Here, indicate whether each material, system or method listed is relevant to your study. If you are not sure if a list item applies to your research, read the appropriate section before selecting a response.

### Materials & experimental systems

| n/a | Involved in the study |
|---|---|
| ☐ | ☒ Antibodies |
| ☐ | ☒ Eukaryotic cell lines |
| ☒ | ☐ Palaeontology and archaeology |
| ☒ | ☐ Animals and other organisms |
| ☒ | ☐ Clinical data |
| ☒ | ☐ Dual use research of concern |
| ☒ | ☐ Plants |

### Methods

| n/a | Involved in the study |
|---|---|
| ☒ | ☐ ChIP-seq |
| ☐ | ☒ Flow cytometry |
| ☒ | ☐ MRI-based neuroimaging |

## Antibodies

| Antibodies used | Nucleolin (D4C7O) Rabbit mAb, Cell Signaling, #14574<br>P53 (DO-1) mouse mAb, Santa Cruz, sc-126 |
|---|---|
| Validation | All antibodies were validated for the specific application by the manufacturer and validation data is available on the manufacturer's website.<br>All antibodies were validated for the specific application by the manufacturer and validation data is available on the manufacturer's website.<br>Nucleolin https://www.cellsignal.com/products/primary-antibodies/nucleolin-d4c7o-rabbit-mab/14574<br>P53 https://www.scbt.com/de/p/p53-antibody-do-1 |

## Eukaryotic cell lines

Policy information about cell lines and Sex and Gender in Research

| Cell line source(s) | MCF10A and RPE-1 were acquired from ATCC. RPE-1 TP53-/- was previously developed in our lab. MCF10A TP53-/- were a kind gift of Prof. Dr. Claudia Scholl. |
|---|---|
| Authentication | Genomic data for cell lines used matched their published karyotypes |
| Mycoplasma contamination | All cell lines used tested negative for mycoplasma contamination |
| Commonly misidentified lines<br>(See ICLAC register) | *Name any commonly misidentified cell lines used in the study and provide a rationale for their use.* |

# Palaeontology and Archaeology

Specimen provenance

*Provide provenance information for specimens and describe permits that were obtained for the work (including the name of the issuing authority, the date of issue, and any identifying information). Permits should encompass collection and, where applicable, export.*

Specimen deposition

*Indicate where the specimens have been deposited to permit free access by other researchers.*

Dating methods

*If new dates are provided, describe how they were obtained (e.g. collection, storage, sample pretreatment and measurement), where they were obtained (i.e. lab name), the calibration program and the protocol for quality assurance OR state that no new dates are provided.*

☐ Tick this box to confirm that the raw and calibrated dates are available in the paper or in Supplementary Information.

Ethics oversight

*Identify the organization(s) that approved or provided guidance on the study protocol, OR state that no ethical approval or guidance was required and explain why not.*

Note that full information on the approval of the study protocol must also be provided in the manuscript.

# Animals and other research organisms

Policy information about studies involving animals; ARRIVE guidelines recommended for reporting animal research, and Sex and Gender in Research

Laboratory animals

*For laboratory animals, report species, strain and age OR state that the study did not involve laboratory animals.*

Wild animals

*Provide details on animals observed in or captured in the field; report species and age where possible. Describe how animals were caught and transported and what happened to captive animals after the study (if killed, explain why and describe method; if released, say where and when) OR state that the study did not involve wild animals.*

Reporting on sex

*Indicate if findings apply to only one sex; describe whether sex was considered in study design, methods used for assigning sex. Provide data disaggregated for sex where this information has been collected in the source data as appropriate; provide overall numbers in this Reporting Summary. Please state if this information has not been collected. Report sex-based analyses where performed, justify reasons for lack of sex-based analysis.*

Field-collected samples

*For laboratory work with field-collected samples, describe all relevant parameters such as housing, maintenance, temperature, photoperiod and end-of-experiment protocol OR state that the study did not involve samples collected from the field.*

Ethics oversight

*Identify the organization(s) that approved or provided guidance on the study protocol, OR state that no ethical approval or guidance was required and explain why not.*

Note that full information on the approval of the study protocol must also be provided in the manuscript.

# Clinical data

Policy information about clinical studies
All manuscripts should comply with the ICMJE guidelines for publication of clinical research and a completed CONSORT checklist must be included with all submissions.

Clinical trial registration

*Provide the trial registration number from ClinicalTrials.gov or an equivalent agency.*

Study protocol

*Note where the full trial protocol can be accessed OR if not available, explain why.*

Data collection

*Describe the settings and locales of data collection, noting the time periods of recruitment and data collection.*

Outcomes

*Describe how you pre-defined primary and secondary outcome measures and how you assessed these measures.*

# Dual use research of concern

Policy information about dual use research of concern

## Hazards

Could the accidental, deliberate or reckless misuse of agents or technologies generated in the work, or the application of information presented in the manuscript, pose a threat to:

| No | Yes | |
|---|---|---|
| ☒ | ☐ | Public health |
| ☒ | ☐ | National security |
| ☒ | ☐ | Crops and/or livestock |
| ☒ | ☐ | Ecosystems |
| ☒ | ☐ | Any other significant area |

## Experiments of concern

Does the work involve any of these experiments of concern:

| No | Yes | |
|---|---|---|
| ☒ | ☐ | Demonstrate how to render a vaccine ineffective |
| ☒ | ☐ | Confer resistance to therapeutically useful antibiotics or antiviral agents |
| ☒ | ☐ | Enhance the virulence of a pathogen or render a nonpathogen virulent |
| ☒ | ☐ | Increase transmissibility of a pathogen |
| ☒ | ☐ | Alter the host range of a pathogen |
| ☒ | ☐ | Enable evasion of diagnostic/detection modalities |
| ☒ | ☐ | Enable the weaponization of a biological agent or toxin |
| ☒ | ☐ | Any other potentially harmful combination of experiments and agents |

# Plants

| | |
|---|---|
| Seed stocks | *Report on the source of all seed stocks or other plant material used. If applicable, state the seed stock centre and catalogue number. If plant specimens were collected from the field, describe the collection location, date and sampling procedures.* |
| Novel plant genotypes | *Describe the methods by which all novel plant genotypes were produced. This includes those generated by transgenic approaches, gene editing, chemical/radiation-based mutagenesis and hybridization. For transgenic lines, describe the transformation method, the number of independent lines analyzed and the generation upon which experiments were performed. For gene-edited lines, describe the editor used, the endogenous sequence targeted for editing, the targeting guide RNA sequence (if applicable) and how the editor was applied.* |
| Authentication | *Describe any authentication procedures for each seed stock used or novel genotype generated. Describe any experiments used to assess the effect of a mutation and, where applicable, how potential secondary effects (e.g. second site T-DNA insertions, mosiacism, off-target gene editing) were examined.* |

# ChIP-seq

## Data deposition

☐ Confirm that both raw and final processed data have been deposited in a public database such as GEO.

☐ Confirm that you have deposited or provided access to graph files (e.g. BED files) for the called peaks.

| | |
|---|---|
| Data access links<br>*May remain private before publication.* | *For "Initial submission" or "Revised version" documents, provide reviewer access links. For your "Final submission" document, provide a link to the deposited data.* |
| Files in database submission | *Provide a list of all files available in the database submission.* |
| Genome browser session<br>(e.g. UCSC) | *Provide a link to an anonymized genome browser session for "Initial submission" and "Revised version" documents only, to enable peer review. Write "no longer applicable" for "Final submission" documents.* |

## Methodology

| | |
|---|---|
| Replicates | *Describe the experimental replicates, specifying number, type and replicate agreement.* |
| Sequencing depth | *Describe the sequencing depth for each experiment, providing the total number of reads, uniquely mapped reads, length of reads and whether they were paired- or single-end.* |
| Antibodies | *Describe the antibodies used for the ChIP-seq experiments; as applicable, provide supplier name, catalog number, clone name, and lot number.* |
| Peak calling parameters | *Specify the command line program and parameters used for read mapping and peak calling, including the ChIP, control and index files used.* |

| Data quality | *Describe the methods used to ensure data quality in full detail, including how many peaks are at FDR 5% and above 5-fold enrichment.* |
|---|---|
| Software | *Describe the software used to collect and analyze the ChIP-seq data. For custom code that has been deposited into a community repository, provide accession details.* |

# Flow Cytometry

## Plots

Confirm that:

☒ The axis labels state the marker and fluorochrome used (e.g. CD4-FITC).

☐ The axis scales are clearly visible. Include numbers along axes only for bottom left plot of group (a 'group' is an analysis of identical markers).

☐ All plots are contour plots with outliers or pseudocolor plots.

☒ A numerical value for number of cells or percentage (with statistics) is provided.

## Methodology

| Sample preparation | In case of Strand-seq experiments, at the end of the photolabeling phase, cells were stained for one hour with Hoechst 33342. Cells were harvested with 0.25% Trypsin (Gibco) and resuspended in buffer (8% FBS in 1X PBS, supplemented with Hoechst 33342 5µg/ml and BrdU 40 µM). |
|---|---|
| Instrument | BD FACSAria |
| Software | BD FACSDiva |
| Cell population abundance | Due to limited sample material, post-sort purities were not re-assessed using flow cytometry. |
| Gating strategy | We employed the following gating strategy: we selected first the general population in forward and side scatter and we excluded doublets. Then, cells were sub-gated for photolabeled cells as shown in Figure 1G for H2B-Dendra2 or Figure S2G for DACT-1. When employing Strand-seq, the singlet population was further filtered to select cells with a quenched Hoechst signal that had thus incorporated BrdU. Cells harvested from control slides were used to optimally adjust gates to exclude false positives. |

☐ Tick this box to confirm that a figure exemplifying the gating strategy is provided in the Supplementary Information.

# Magnetic resonance imaging

## Experimental design

| Design type | *Indicate task or resting state; event-related or block design.* |
|---|---|
| Design specifications | *Specify the number of blocks, trials or experimental units per session and/or subject, and specify the length of each trial or block (if trials are blocked) and interval between trials.* |
| Behavioral performance measures | *State number and/or type of variables recorded (e.g. correct button press, response time) and what statistics were used to establish that the subjects were performing the task as expected (e.g. mean, range, and/or standard deviation across subjects).* |

## Acquisition

| Imaging type(s) | *Specify: functional, structural, diffusion, perfusion.* |
|---|---|
| Field strength | *Specify in Tesla* |
| Sequence & imaging parameters | *Specify the pulse sequence type (gradient echo, spin echo, etc.), imaging type (EPI, spiral, etc.), field of view, matrix size, slice thickness, orientation and TE/TR/flip angle.* |
| Area of acquisition | *State whether a whole brain scan was used OR define the area of acquisition, describing how the region was determined.* |

Diffusion MRI     ☐ Used     ☐ Not used

## Preprocessing

| Preprocessing software | *Provide detail on software version and revision number and on specific parameters (model/functions, brain extraction, segmentation, smoothing kernel size, etc.).* |
|---|---|

| Normalization | *If data were normalized/standardized, describe the approach(es): specify linear or non-linear and define image types used for transformation OR indicate that data were not normalized and explain rationale for lack of normalization.* |
|---|---|
| Normalization template | *Describe the template used for normalization/transformation, specifying subject space or group standardized space (e.g. original Talairach, MNI305, ICBM152) OR indicate that the data were not normalized.* |
| Noise and artifact removal | *Describe your procedure(s) for artifact and structured noise removal, specifying motion parameters, tissue signals and physiological signals (heart rate, respiration).* |
| Volume censoring | *Define your software and/or method and criteria for volume censoring, and state the extent of such censoring.* |

## Statistical modeling & inference

| Model type and settings | *Specify type (mass univariate, multivariate, RSA, predictive, etc.) and describe essential details of the model at the first and second levels (e.g. fixed, random or mixed effects; drift or auto-correlation).* |
|---|---|
| Effect(s) tested | *Define precise effect in terms of the task or stimulus conditions instead of psychological concepts and indicate whether ANOVA or factorial designs were used.* |

Specify type of analysis: ☐ Whole brain ☐ ROI-based ☐ Both

| Statistic type for inference<br><br>(See Eklund et al. 2016) | *Specify voxel-wise or cluster-wise and report all relevant parameters for cluster-wise methods.* |
|---|---|
| Correction | *Describe the type of correction and how it is obtained for multiple comparisons (e.g. FWE, FDR, permutation or Monte Carlo).* |

## Models & analysis

| n/a | Involved in the study |
|---|---|
| ☐ | Functional and/or effective connectivity |
| ☐ | Graph analysis |
| ☐ | Multivariate modeling or predictive analysis |

| Functional and/or effective connectivity | *Report the measures of dependence used and the model details (e.g. Pearson correlation, partial correlation, mutual information).* |
|---|---|
| Graph analysis | *Report the dependent variable and connectivity measure, specifying weighted graph or binarized graph, subject- or group-level, and the global and/or node summaries used (e.g. clustering coefficient, efficiency, etc.).* |
| Multivariate modeling and predictive analysis | *Specify independent variables, features extraction and dimension reduction, model, training and evaluation metrics.* |

