## [Peer Review File · Nature]

Origins of chromosome instability unveiled by coupled imaging and genomics

Corresponding Author: Professor Jan Korbel

Version 1:

Reviewer comments:

Referee #1

(Remarks to the Author)

Summary:

In this paper, Cosenza et al. introduce a new approach called MAGIC for automated identification of cultured cells with micronuclei by microscopy and photolabeling them. They then perform single-cell Strand-seq profiling of MAGIC-labeled cells to characterize spontaneously occurring chromosomal aberrations (CAs) that are enriched in micronuclei-containing cells. This would ordinarily have been laborious due to the challenge in identifying in picking rare cells with micronuclei, and MAGIC makes this process more efficient by capturing ~5-fold more of these cells.

The authors focus their experiments on two cell lines: MCF10A and RPE-1. They find different spontaneous rates and distributions of CAs in the two cell lines, and an increase in CA frequency in the absence of P53. They develop a computational model to estimate the per-cell division rate of CA formation in different types of mitoses (normal, lagging chromosome, bridge-containing mitoses) in these cell lines and find that p53 loss increases CAs not by changing these rates, but rather the fraction of cells with nuclear atypias and abnormal mitosis types. Finally, they explore the effect of inducing double-strand breaks (DSBs) in sub-centromeric, middle arm, and sub-telomeric regions of chromosomes 2 and 7, and they discover that DSBs targeted to different regions give rise to different types of CAs that are known to occur in cancer. For example, they find that subcentromeric DSBs can give rise to isodicentric chromosomes and whole-arm CAs, and they find that central-arm and subtelomeric DSBs are more prone to the bridge fusion breakage cycle.

General comments:

The paper is distinguished by its ability to streamline the workflow for profiling rare cells with micronuclei, which are enriched for CAs, and this in turn allows greater statistical power and experimental flexibility to study patterns of CA formation in cultured cells. The authors find a mechanistic basis for the formation of two types of CAs, isodicentric and isoacentric chromosomes, they find that the location of DSBs has a strong influence on the types of CAs formed, they show that p53 affects CA formation rates, and they present the first landscape of spontaneously-occurring CAs, albeit in cell lines. The combination of targeted CRISPR-induced DSBs + MAGIC + Strand-seq was the most powerful demonstration of the potential of the method, because it suggests that many other types of experimental manipulations will be able to systematically dissect mechanisms of CA formation. The CA patterns they identified by targeted CRISPR cutting may also allow engineering of new cell line models with specific CAs that would not be possible to be engineered otherwise.

Key limitations of the study are:

a) MAGIC seems only applicable to cultured cells, so it cannot be used to explore in vivo CAs rates.

b) The manual picking of micronuclei-containing cells in prior studies was laborious, but those studies could correlate the cell image/morphology with subsequent genetic data. However, the automated MAGIC platform as presented cannot connect the image of the cell structure/morphology to the subsequent sequencing data. Therefore, correlations cannot be made between nuclei/micronuclei structure and the DNA sequencing data.

c) The experiments involved two cell lines that already showed differences in their CA patterns, so more cell lines and types will need to be profiled before the community can determine the generalizability of these findings.

d) The use of Strand-seq provides homolog-specific profiling, which is important for CA characterization, but the throughput is limited to hundreds of cells. The authors demonstrated MAGIC combined with single-cell cloning and FISH, but it would have been illuminating and expand its impact to combine it with other higher-throughput single-cell methods (for example, scRNA-seq or scATAC-seq). However, I realize this may require a fair amount of work. But if it is feasible and not a big burden, I would encourage the authors to consider this.

e) MAGIC is limited to profiling micronuclei-containing cells, but CAs can also arise in the absence of micronuclei. So the view of CA processes exposed here may be biased as it is restricted to only those types of cells.

Overall, the study is technically rigorous and well-presented, the authors make some notable discoveries regarding formation of isodicentric chromosomes and acentric amplified fragments, and the method will allow the field to systematically explore influences on CA formation, which is important for understanding cancer initiation. So I support publication, assuming the below specific comments are addressed.

Introduction and abstract:

1. The authors say they used 'non-transformed' cell lines, but then say that RPE-1 cells have hTERT. Doesn't hTERT make it a transformed cell line?

MAGIC:

1. The scale of the number of cells that can be interrogated per experiment should be described. It is mentioned in the Discussion, but more suited here.
2. Could the dyes used cause nuclear atypias and CAs, or perhaps a change in CA patterns? Was this evaluated in a control experiments?
3. Is it known whether brdU labeling can cause CAs? For example, by triggering DSBs due to DNA repair that recognizes the brdU?
4. Automated identification of sister cell pairs from Strand-seq data is very interesting, but only two sister cell pairs were identified in the first experiment. Why is the fraction so low? Could magic be configured to automatically label sister sell pairs when one of the two cells is detected to have a micronucleus?
5. Could MAGIC be combined with an automated cell picking instrument to allow linking of the cell image with single-cell sequencing? For example, the CellCelector instrument. If so, this possibility could be mentioned, even though out of scope of this paper.

Landscape of spontaneous CA formation

1. Regarding the nonsignificant enrichment of CAs in the RPE cell line compared to MCF cell line: could this be due to differences in performance of the MAGIC machine-learning model in recognizing micronuclei of the two cell lines? In general, can the authors compare performance of recognizing micronuclei cells in each cell line?
2. The number of sister cell pairs identified is again quite small, limiting the biological insights to anecdotes about each sister cell pair rather than large-scale statistics.
3. The RPE cell line is more stable and also has different CA patterns than the MCF cell line. But this diversity is explored in only two cell lines, and this raises the question of how generalizable these conclusions are to other cell lines, and potentially healthy tissues. Can the authors comment on this?
4. The authors show a difference in CAs between clonally expanded micronuclei-containing cells versus those selected directly for Strand-seq profiling to suggest that clonal expansion selects against some CAs. But even with direct profiling by Strand-Seq, there is required at least one cell division for the brdU to incorporate. Could selection already be acting in that one cell division so that even the CA profile of direct Strand-seq profiling already has a selection bias?
5. Can Strand-seq detect circularization events? If this is feasible in principle, it would be interesting to quantify in this and the other study's experiments. If not, then this should be commented on in the Discussion, since these events are important types of structural events in cancer.

Prevalence of dicentric-mediated CAs and chromothripsis

1. Just to commend the authors that detection of chromothripsis in sister cell pairs is impressive.

Chromosomal location determinants for CA formation

1. The section on FISH should describe in the text more quantitatively how many cells and how many karyotypes were analyzed.
2. CRISPR cutting will generally occur on both sister chromatids, which can lead to some of the phenomena that the authors observe. But in the absence of CRISPR, what is the probability that randomly occurring DSBs will occur on both sister chromatids both in the sub-centromeric region, for example? Presumably this is significantly more rare, and therefore, would the isodicentric and isoacentric events observed relatively frequently here would be more rare in cancer and in healthy cells? The authors should comment on this, perhaps in the discussion and how this artificial CRISPR system may relate to what happens in vivo.

Discussion:

1. The authors should discuss whether MAGIC could be coupled with other higher-throughput droplet-based methods, such as scATAC-seq or scRNA-seq. What would these methods be able to detect (CNVs, phenotypic changes) vs Strand-seq?

2. The discussion is quite long and can be shortened.
3. Another limitation that should be mentioned is that Strand-seq cannot resolve the base-level sequences at the breakpoints.
4. “as well as the identification of elevated de novo CA formation in clinical samples. This could help identify high-risk precancerous lesions and might improve patient outcomes through timely interventions in the future.”: it is not clear to me how MAGIC could be applied to clinical samples. Either further details should be provided or this should be removed.
5. “but also reveal the basal rate at which CAs emerge”: the authors should qualify this to say this is the rate in two cell lines.

Methods: No comments

Figures:

1. Figs. 3A-E would be easier to interpret with side-by-side comparisons of WT vs p53 cells
2. Fig. 3G y-axis label should make more clear this rate is estimated from a model, not directly measured
3. Figs. 5C,D would benefit from some schematic showing the structure of the exhibited CA. Also, if feasible, the authors should reduce the size of each of these panels to allow showing one karyotype example for each type of CA.

(Remarks on code availability)

Referee #2

(Remarks to the Author)

In this manuscript, the authors present a platform called MAGIC, which couples automated microscopy in live cells, evaluation of nuclear abnormalities via machine learning, targeted cell photolabeling and cell sorting. The isolated cells are then subjected to single-cell sequencing or phenotypic analyses. This approach has been applied to two non-transformed near-diploid cell lines, MCF10A and RPE1. The authors profiled mitotic errors in the unperturbed cell lines using MAGIC followed by Strand-seq, thereby mapping spontaneous chromosomal aberrations (CAs) that arise from micronuclei. Interestingly, this enabled the authors to reconstruct chromosomal aberrations arising across successive cell cycles. A comparison of p53-deficient vs. p53-proficient MCF10A and RPE1 cells confirms that p53 inactivation increases the rate of micronucleus formation and chromosomal aberrations, with more complex CAs emerging in micronucleated p53-null RPE1 cells. CRISPR-Cas9-based induction of double strand breaks (DSBs) revealed that certain CA processes were influenced by the initiating DSB chromosomal location (for example, sub-centromeric DSBs induced isochromosome formation), and FISH was used to validate the Strand-seq results and further interrogate these CAs.

This is an interesting and timely paper, which provides some insights into the de novo formation of chromosomal aberrations. The integration of automated imaging and genomic profiling is powerful. Nonetheless, the study remains largely descriptive, and to justify publication in Nature, I'd propose extending it beyond the technology development in at least one of the following directions:

- (1) Making the MAGIC pipeline and data as accessible and easy-to-adopt as possible: the application of the proposed pipeline in other labs is far from trivial, and the authors need to make an effort to make all steps – experimental and computational alike – accessible and easy to use. The authors have already made the code available, which is valuable. Detailed protocols and tutorials for the experimental aspects of the work should be added as well. In addition, please provide the single-cell CA data for the 2,036 analyzed single cells in a format that will facilitate additional analysis of the data.
- (2) The manuscript currently focuses on spontaneous aberrations that arise in two non-transformed cell lines. Would the findings hold true for micronucleation induced by perturbing chromosome segregation? In other words, is there a difference between spontaneous micronucleation, and micronucleation that is induced by common methods to induce CIN, such as MPS1 inhibition? Such comparison would be very valuable for the field as it could reveal whether the common methods for CIN induction mimic the “natural” CIN that exists in these cells.
- (3) The authors begin to address the question of de novo formation of CAs vs. their selection. Looking deeper into selection would be of much interest. (I'd like to leave leeway for the authors to address this point in whatever ways they see fit.)

Below are a few specific suggestions for additional analyses that could strengthen the main conclusions of the study:

- Can the observed CA formation be linked to the CNA signatures reported by Steele et al. and Drews et al. (Nature 2022)? Please evaluate and discuss.
- CAs were strongly enriched in micronucleated MCF10A cells, but not in micronucleated RPE1 cells. This is quite surprising. Is this also true if micronucleation is induced in these cells by perturbing chromosome segregation?
- In MCF10A, CAs were enriched for chromosome 19. MCF10A harbors several clonal arm-level chromosomal gains, presumably due to unbalanced translocations – are those events not enriched for CAs? In RPE1, CAs were enriched in specific chromosomes – does the rate of CAs correlate the rate of chromosome mis-segregation previously reported in these cells (Klaasen et al. Nature 2022)?
- The results may help explain recent findings of large chromosomal losses following CRISPR-Cas9 genome editing

(Nahmad et al. Nat Biotechnol 2022; Tsuchida et al. Cell 2023). This should be discussed.

- In RPE1 and MCF10A cells, previous studies found that whole-chromosome monosomies were viable only on a p53-deficient background. Was there a difference in the prevalence of gains and losses upon p53 inactivation? This should also be mentioned in the discussion of this topic.

(Remarks on code availability)

Referee #3

(Remarks to the Author)

This work introduces MAGIC, a novel platform to integrate live-cell imaging, machine learning and single-cell genomics.

My comments are limited to the image analysis aspects of this work, including the software provided by the authors.

Image segmentation and classification can be approached in many ways. The strategy here combines conventional image processing with pixel and object classification using XGBoost. This is not in itself conceptually novel, but the details are adapted for the application and it represents a suitable pragmatic approach. The primary question is whether it can achieve sufficient accuracy to support the downstream analysis.

Main comments:

- Accurately segmenting nuclei and micronuclei is a key step. The approach here is to first generate a custom semantic segmentation with XGBoost and Gaussian/Hessian features, and then post-process to generate instances. Based on the strategy and examples provided, one expects there to be issues with nuclei being missed, fragmented, or clumped together.
 - Was the accuracy of the segmentation quantified? This could be using exhaustively annotated images from the study, or a publicly-available benchmark (e.g. from the Broad Bioimage Benchmark Collection).
 - Were other approaches considered? These could either be simpler (e.g. image processing without machine learning), or more complex (e.g. an established deep learning strategy like StarDist or CellPose).

- The manuscript describes 'real-time' detection of micronuclei. The term 'real-time' is ambiguous, but the actual detection takes at least a few seconds (both on my computer, and reported in the paper) - which seems quite slow.

- To what extent is fast processing critical for the application?

- I would guess that the Hessian features are contributing substantially to the processing time; were these shown to be necessary?

- Was GPU acceleration considered?

- Classification models were trained to achieve at least 90% precision and 50% recall.

- Do these values reflect classification *only*, or are errors accumulated during the segmentation also incorporated in some way?

- I think that precision and recall per-image would be most relevant - i.e. considering all objects, and not restricted to only those that were correctly segmented.

- The values suggest a low rate of false positives, but a rather high rate of false negatives. Is it possible to determine that there is not something 'systematically different' about the false negatives that may influence downstream results?

- For example, lower intensity, different morphology, too close to other cells.

- XGBoost is justified as 'performing similarly to a deep learning-based convolutional neural network'. However the details of the CNN architecture depicted in Supplementary Table 3 are unclear, and the choice seems quite unconventional. Is there a justification for using this custom architecture, rather than something more established (e.g. a ResNet, MobileNet)?

- Given the possibility of both segmentation and classification errors, were the 'pre-' and 'post-experiment' images used to assess the 'real-world' performance of the system, ensuring that it achieves sufficient precision and recall across all relevant conditions?

- I find it quite difficult to follow the image analysis workflow, specifically around when a nucleus and micronuclei are considered as a single object and when they are considered/classified separately. I hoped that 'object_classification_example.ipynb' would resolve this, but I remain unclear. I think that the images with contours at the bottom are supposed to show nuclei with associated micronuclei, but I am not sure whether these should all be considered true positives - nor how they compare with cells that were rejected.

(Remarks on code availability)

It is commendable that the software is made available under an open-source license, and is provided with documentation.

Three example fields of view are also provided, along with several training images.

Because it is not possible for me to test the system with a real microscope, I tried to follow the instructions given under 'Testing micronuclei detection plugin' on the 'magic_automation' repository.

This was quite difficult, because important instructions are split across two repositories. It was necessary to add a Fiji update site, install an extra file to Fiji, set up a conda environment, and run a server.

However when I tried this on macOS, I encountered further errors because I needed to carry out further non-obvious steps that were only described later, such as running 'set_server_configuration.py' (which must be done after the server is running), and creating a new 'logs/images' directory.

It would be helpful to try to streamline the process further, to ensure that others can run the code to test the method more easily.

It would also help to provide more images to give a better sense of performance, and to more clearly show in the Jupyter notebooks the classification results - including true/false positives/negatives. In the low-resolution images, particularly for 'micronuclei candidates', it is difficult to interpret what is depicted.

Minor comments:

- The software has been written for use with a Zeiss microscope. Would it be feasible for the software to work on different imaging systems? Do you envisage others using the software in its current form, or is the purpose rather to show proof-of-concept?

- The file 'example_training-set_object-classification.csv' does not appear to be a CSV.

Referee #4

(Remarks to the Author)

Summary:

The paper proposes a platform named MAGIC, which combines confocal live-cell imaging, machine learning-based image processing, and single-cell genomics for the analysis of chromosomal abnormalities (CAs). The study focuses on two non-transformed cell lines, MCF10A and RPE-1, to investigate de novo CAs and their underlying mechanisms. The platform workflow involves live-cell confocal microscopy imaging, manual annotation of cells with micronuclei, training a machine learning algorithm (XGBoost) for automatic identification of micronucleated cells, targeted photolabeling, cell sorting, and ultimately, single-cell genomic characterization of the sorted cells.

One highlight of the paper is the integration of machine learning with targeted photolabeling and cell sorting to enable efficient, large-scale selection of cells with micronuclei. Using MAGIC, the authors analyzed up to 80,000 cells per experiment and isolated over 2,000 single cells for downstream analysis. The study finds that dicentric chromosomes are significant contributors to CA formation, leading to chromosomal gains and losses. Additionally, the paper shows that TP53 deficiency doubles the baseline rate of CA formation, suggesting that p53 mutation plays a critical role in promoting CA propagation in cancer genomics. Furthermore, the authors identified distinct classes of CAs and demonstrated that the types of CAs can be influenced by inducing double-strand breaks (DSBs) at targeted locations. Overall, the study exemplifies how interdisciplinary techniques can advance our understanding of genomic instability. Notably, the real-time feedback loop for automated cell identification and targeted microscopy illumination based on nuclear morphology enhances the scalability of single-cell analysis.

However, I have a few questions and comments as detailed below:

Machine learning:

General Machine Learning Pipeline:

According to the manuscript, the machine learning algorithm for identifying candidate cells can be generally divided into three steps: (1) semantic segmentation to separate nuclei foreground from the background, (2) instance segmentation from the nuclei mask generated in step 1 using morphological operations and watershed, and (3) object classification that uses object-level features to classify objects as normal nuclei or micronuclei. Steps 1 and 3 employ separate XGBoost classifiers. While these classical morphological operations, watershed, and intensity-based features are straightforward, they may lack robustness. They could result in fragmented segmentations, as seen in the example provided in the notebook, `semantic_segmentation_example.ipynb` (also see exemplary segmentation result in the attachment). If these fragmented segmentations can affect steps 2 and 3, how do they impact overall performance? The manuscript reports only precision and recall for step 3 as 0.9 and 0.5, respectively. It would be beneficial to examine the potential error rates of steps 1 and 2 as well.

Annotation Difficulty:

The proposed pipeline requires two types of annotations: foreground/background annotation for semantic segmentation (step 1) and normal nuclei/micronuclei annotation for object classification (step 3). While annotations for step 1 seem relatively straightforward, the ones for step 3 can be challenging based on the example images provided on GitHub. I am curious about the workload and inter-expert variability for these annotations. If the annotation process is not straightforward, how much effort would users need to employ this platform effectively?

Precision/Recall:

The manuscript claims that the ML framework achieves a precision of at least 90% and a recall of 50%, as shown in Figure

S2. However, the object classification notebook reports a validation precision of 0.5 and a recall of 0.8 (see attachment for screenshot). Could the authors clarify this discrepancy? In the context of the MAGIC platform, where the selection of micronucleated cells for downstream genomic analysis is critical, would it be more important to prioritise high precision or high recall at this stage?

Model Generalizability:

I am somewhat skeptical about the robustness and generalizability of the model. The data provided in the study was tested in a single lab environment. How would the model perform on images from other labs or images acquired with different microscopy setups? The authors trained an XGBoost model for each cell line, which suggests potential challenges in model generalization across different cell lines.

Suggestions:

I recommend considering Cellpose (Stringer et al., 2021 and also Cellpose2/Cellpose3), a state-of-the-art nuclei segmentation tool, as a potential solution for nuclei instance segmentation. This could help address potential segmentation issues in steps 1 and 2.

Code:

I attempted to run the code in the Magic_tool notebook, but it was unsuccessful. While installation appeared to be fine, the magic_tool import did not work on both my MacBook and Google Colab. I suggest that the authors prepare a ready-to-use Google Colab notebook for testing, which would avoid issues related to operating system compatibility during software installation.

Line 190-191:

"Utilizing FACS, we observe distinct populations that represent photolabeled cells with either dye (Fig. 1G, S1G), confirming that target cells are efficiently sorted."

Figure S1 does not include a panel G. Did the authors mean to refer to Figure S2G here?

Main Figure 2:

In panel (C), which depicts mitotic history reconstruction for reciprocal CAs in (B), sister A has only one copy number (indicating a loss), while sister B has three copies. Could the authors also provide an illustration for the shared CA where both A and B exhibit a single copy number loss?

(Remarks on code availability)

As above, although I was not able to run the code (see screenshot of errors in the attachment), I checked the main image processing and machine learning pipeline and results produced in the python notebook. But authors need to resolve the installation issue and can provide a ready-to-use google colab notebook to avoid ambiguity for the code installation.

Version 2:

Reviewer comments:

Referee #1

(Remarks to the Author)

The authors addressed well all of my comments, including additional key experiments that expand the generalizability of their findings. I have no further comments.

(Remarks on code availability)

Referee #2

(Remarks to the Author)

The authors have addressed my comments in a satisfactory manner. I have no further comments.

(Remarks on code availability)

Referee #3

(Remarks to the Author)

I thank the authors for their detailed replies to my comments on the image analysis aspects of this work, and for the additional clarifications incorporated into the manuscript.

While I appreciate that the segmentation strategy using XGBoost resembles the approach found in ilastik, I believe ilastik was released around 14 years ago. This was long before the widespread use of deep learning models, which dramatically improved the accuracy of nucleus segmentation - especially since the Kaggle 2018 Data Science Bowl (<https://doi.org/10.1038/s41592-019-0612-7>). Therefore I welcome the inclusion of a statement about the potential use of

other methods in the manuscript.

I also appreciate the clarification that the deep learning architecture tested was based on ConvNeXt, that CellPose was not available at the time this work was started, and that StarDist's star-convex assumption was inappropriate here. Given its near ubiquity for bioimage analysis over some years, I think that a U-Net would have been the natural comparison - and perhaps this might be considered as a drop-in replacement for the current strategy in the future.

(Remarks on code availability)

Referee #4

(Remarks to the Author)

Many thanks to the authors for providing the revised manuscript along with detailed responses to reviewer comments. The revisions address most of my previous concerns; however, I have two remaining points for further clarification:

Precision and Recall

It may indeed be acceptable to achieve high precision with relatively lower recall in subpopulation selection. However, I recommend assessing whether the selected subpopulation accurately represents a random and unbiased subset. It is important to verify that cells missed by the image processing pipeline do not disproportionately belong to a particular subtype, as this could significantly bias subsequent analyses.

Code and Documentation

We (my PhD co-reviewer and I) successfully tested the provided Docker container on a Windows system. However, the pre-built version failed when tested on a macOS system by another colleague. If compatibility issues exist, it is crucial to clearly mention system dependencies or limitations on your website. Additionally, providing detailed instructions tailored to users with different operating systems would significantly enhance accessibility.

Furthermore, the GitLab repository ([magic_automation](https://git.embl.de/tweber/magic_automation)) currently states: "A pre-built Docker container (<https://git.embl.de/tweber/magic-container>) is available for running the full suite of MAGIC tools, including strandtools, magic_tools, and magic_automation." However, the Docker container we tested contains only strandtools (see attached screenshot). Clarifying or correcting this description would help users accurately understand the available tools.

(Remarks on code availability)

See above comments to the authors

Version 3:

Reviewer comments:

Referee #4

(Remarks to the Author)

Dear authors, all my comments are well addressed in the rebuttal and revision. For Fig. R1, I agree with the authors it does not need to put as a supplementary figure since it is already discussed in the discussion.

(Remarks on code availability)

I am also happy with the updated code instructions.

To all four reviewers:

We thank the reviewers for their positive and constructive feedback on our study, and for recognizing its implications in understanding chromosome aberrations (CAs) and genomic instability. The reviewers have acknowledged several unique advancements of our study:

- **Reviewers #1, #2, #3, and #4** commend the novelty of the MAGIC platform for machine-learning assisted identification of cells bearing micronuclei by high-throughput imaging and photolabeling involving an on-the-fly adaptive-feedback microscopy, coupled to single-cell sequencing of CAs. MAGIC automates previously laborious tasks to vastly enhance the scalability of single-cell genomic analysis, and therefore enable studying *de novo* CAs with much enhanced throughput and statistical power (**Reviewers #1 and #4**). They moreover stress how MAGIC enables reconstructing CA processes across successive cell cycles, and that our study exemplifies how interdisciplinary techniques can advance our understanding of genomic instability (**Reviewer #2**).
- **Reviewers #1, #2 and #4** praise our ability to estimate per-cell division CA rates in culture based on MAGIC, enabled through an agent-based model providing insights into *de novo* CA formation in *TP53* deficient and proficient cells. They emphasise how our study unveils, for the first time, the genomic landscape of *de novo* CAs - despite the present limitation to in vitro conditions (**Reviewer #1**), and highlight the achievement of our study to elucidate how *TP53*-loss results in a doubling of the CA rate (**Reviewer #4**), paralleled by an increase in micronucleation (**Reviewer #1, #2**).
- **Reviewers #1, #2, and #4** recognise the utility of coupling MAGIC with targeted CRISPR-induced double-strand break (DSB) generation, noting its potential to unravel mechanisms underlying CA formation events. They highlight our findings that DSBs targeted to different chromosomal regions give rise to different CA classes known to occur in cancer – e.g. with subcentromeric DSBs triggering isodicentrics, whereas central-arm and subtelomeric DSBs are prone to result in breakage fusion bridge cycles. **Reviewer #1** also stresses that our data provide a mechanistic basis for the formation of two classes of CA – isochromosomes and isoacentrics – that the coupling of MAGIC and CRISPR may in the future allow isolating new cell line models with specific CAs that would not be possible to be engineered otherwise, and that the MAGIC platform will allow the field to systematically explore influences on CA formation, which is important for understanding cancer initiation.

Encouraged by these reviews, and taking into account requests for additional clarification and experimentation we have worked to further revise our manuscript. Below we respond to each comment made by the reviewers point-by-point. We have, particularly, identified three key areas in the reviewers' comments:

(1) *Enhancement of the generalizability of our key biological findings.* We have, during manuscript revision, generated data for two additional human cell lines, and performed additional experiments employing biochemical perturbation of chromosome segregation. Our manuscript now comprises a total of 22 experimental conditions – spanning different cell lines, *TP53* statuses, and phenotypic contexts – in which MAGIC was successfully established (see **Table R1** below). In relation to single cell genomics, our revised manuscript now demonstrates the application of MAGIC for isolating 2,898 single cells, followed by

single-cell genomic sequencing in 2,192 cells. This represents a substantial advancement compared to prior studies in the field. As alluded to below, the new data generated allow further generalizing our findings, and provide further insight into spontaneous CA formation.

(2) *Broadening the application range of MAGIC.* Showcasing its versatility, while preparing for manuscript revision, we have coupled MAGIC with single-cell transcriptomics (scRNA-seq). This proof-of-principle application reveals that the p53 pathway is highly upregulated in micronucleated cells, consistent with a key role of *TP53* in determining the cellular outcomes of micronucleation.

(3) *More comprehensive description of our methodology.* We have, as requested, substantially extended the description of the MAGIC platform, to ensure its broad accessibility, including with respect to its embedded machine-learning (ML) components. We are committed to making MAGIC a valuable resource. To ensure this, we have developed a Docker image to foster broader use of MAGIC, and ensure its reproducibility across computational platforms. We have further conducted additional analyses to address reviewer comments surrounding the computer vision and ML methodology. These enhancements clarify our approach and ensure transparency in the model's design and applications.

We once again thank the reviewers for their constructive feedback, allowing us to further strengthen our manuscript.

Table R1 | Summary of experimental conditions MAGIC is applied to in our study

1. MCF10A WT micronuclei (MNI)	12. RPE1 WT CTRL reversine
2. MCF10A WT normal cells (CTRL)	13. BJ5 MNI
3. RPE1 WT MNI	14. BJ5 CTRL
4. RPE1 WT CTRL	15. IMR90 MNI
5. MCF10A TP53 ^{-/-} MNI	16. IMR90 CTRL
6. MCF10A TP53 ^{-/-} CTRL	17. MCF10A WT MNI chr2 centromeric
7. RPE1 TP53 ^{-/-} MNI	18. MCF10A WT MNI chr2 middle
8. RPE1 TP53 ^{-/-} CTRL	19. MCF10A WT MNI chr2 telomeric
9. MCF10A WT MNI reversine	20. MCF10A WT MNI chr7 centromeric
10. MCF10A WT CTRL reversine	21. MCF10A WT MNI chr7 middle
11. RPE1 WT MNI reversine	22. MCF10A WT MNI chr7 telomeric

Referee expertise:

Referee #1: genomics, sequencing technology

Referee #2: genomic instability

Referee #3: biomedical imaging analysis

Referee #4: biomedical image processing

Referees' comments:

Referee #1 (Remarks to the Author):

Summary: In this paper, Cosenza et al. introduce a new approach called MAGIC for automated identification of cultured cells with micronuclei by microscopy and photolabeling them. They then perform single-cell Strand-seq profiling of MAGIC-labeled cells to characterize spontaneously occurring chromosomal aberrations (CAs) that are enriched in micronuclei-containing cells. This would ordinarily have been laborious due to the challenge in identifying in picking rare cells with micronuclei, and MAGIC makes this process more efficient by capturing ~5-fold more of these cells.

The authors focus their experiments on two cell lines: MCF10A and RPE-1. They find different spontaneous rates and distributions of CAs in the two cell lines, and an increase in CA frequency in the absence of P53. They develop a computational model to estimate the per-cell division rate of CA formation in different types of mitoses (normal, lagging chromosome, bridge-containing mitoses) in these cell lines and find that p53 loss increases CAs not by changing these rates, but rather the fraction of cells with nuclear atypias and abnormal mitosis types. Finally, they explore the effect of inducing double-strand breaks (DSBs) in sub-centromeric, middle arm, and sub-telomeric regions of chromosomes 2 and 7, and they discover that DSBs targeted to different regions give rise to different types of CAs that are known to occur in cancer. For example, they find that subcentromeric DSBs can give rise to isodicentric chromosomes and whole-arm CAs, and they find that central-arm and subtelomeric DSBs are more prone to the bridge fusion breakage cycle.

General comments: The paper is distinguished by its ability to streamline the workflow for profiling rare cells with micronuclei, which are enriched for CAs, and this in turn allows greater statistical power and experimental flexibility to study patterns of CA formation in cultured cells. The authors find a mechanistic basis for the formation of two types of CAs, isodicentric and isoacentric chromosomes, they find that the location of DSBs has a strong influence on the types of CAs formed, they show that p53 affects CA formation rates, and they present the first landscape of spontaneously-occurring CAs, albeit in cell lines. The combination of targeted CRISPR-induced DSBs + MAGIC + Strand-seq was the most powerful demonstration of the potential of the method, because it suggests that many other types of experimental manipulations will be able to systematically dissect mechanisms of CA formation. The CA patterns they identified by targeted CRISPR cutting may also allow

engineering of new cell line models with specific CAs that would not be possible to be engineered otherwise.

We thank the reviewer for their encouraging comments and for emphasising the utility and potential broader impact of the methodology presented in our manuscript.

Comment #1.01: *Key limitations of the study are: a) MAGIC seems only applicable to cultured cells, so it cannot be used to explore in vivo CA rates.*

Indeed, our methodology does not currently capture CA rates *in vivo*. Our revised discussion clarifies that its application, presently, is confined to cultured cells.

Comment #1.02: *b) The manual picking of micronuclei-containing cells in prior studies was laborious, but those studies could correlate the cell image/morphology with subsequent genetic data. However, the automated MAGIC platform as presented cannot connect the image of the cell structure/morphology to the subsequent sequencing data. Therefore, correlations cannot be made between nuclei/micronuclei structure and the DNA sequencing data.*

MAGIC currently does not support tracing a sequenced single-cell genome back to the original single-cell image. However, for many applications, we believe that this feature is not strictly necessary. Indeed, as long as the phenotypic criteria are defined precisely enough, all selected cells can be considered equivalent concerning the specific research question. With this in mind, our approach prioritises selecting well-defined phenotypic profiles to enable scalable genomic analysis of cells with targeted phenotypes (*i.e.* micronuclei). Nevertheless, we acknowledge that certain future research directions could benefit from image-to-genome traceability, a possibility we explicitly address in the updated Discussion section. Coupling MAGIC with an automated cell-picking platform (As suggested in **Comment #1.11**) could achieve such integration in the future, albeit most likely at the cost of reduced throughput.

Comment #1.03: *c) The experiments involved two cell lines that already showed differences in their CA patterns, so more cell lines and types will need to be profiled before the community can determine the generalizability of these findings.*

Our initial choice of the RPE-1 and MCF10A cell lines was motivated by their extensive use in foundational studies within the chromosomal instability field¹⁻⁵. Extending MAGIC to additional cell lines necessitates substantial additional effort, including the acquisition of new training image sets and the development of cell-line-specific machine learning models for accurate micronuclei classification. Nonetheless, to enhance the robustness and generalisability of our conclusions, we have now generated new datasets from two additional non-transformed, near-diploid cell lines: BJ-5ta and IMR-90. We sequenced 126 single-cell genomes from BJ-5ta and 130 from IMR-90, including both normal and spontaneously micronucleated cells. Our new data, shown in **Fig. S16** and detailed below, reinforce the generalizability of our key findings. Notably, the *de novo* CA data obtained from spontaneously micronucleation in now four cell lines (RPE-1, MCF10A, BJ-5ta and IMR-90) reveal notable and highly consistent distinctions to the *de novo* CA landscape induced by biochemical perturbation (e.g., reversine treatment; **Fig. S16**) – as also further elaborated in our responses to **Comments #1.12, #2.03, and #2.07**. Beyond that, we make the following observations based on the newly generated data from BJ-5ta and IMR-90:

- We find that BJ-5ta exhibits a significant accumulation of CAs in spontaneously micronucleated cells compared to normal cells, similar to MCF10A ($P=0.003$), whereas IMR-90 appears to show a much weaker (non-significant) signal ($P=0.2$) and in this regard appears to be more similar to RPE-1.
- The genomic CA landscapes of spontaneously micronucleated cells in BJ-5ta and IMR-90 are clearly dominated by terminal abnormalities (**Fig. S16**), indicating that dicentric chromosomes and acentric fragments are among the main triggers of CA formation in these cell lines. This is consistent with, and thus further generalizes our findings made in RPE-1 and MCF10A.
- As we reported for RPE-1 in our originally submitted manuscript, complex *de novo* CAs are rare in spontaneously micronucleated cells from the BJ-5ta and IMR-90 cell lines (**Fig. S16**). No chromothripsis events are detected in BJ-5ta and IMR-90. Consequently, spontaneously micronucleated MCF10A cells emerge as a somewhat exceptional cell line displaying some similarity to *TP53* knockout cell lines. This could be due to the fact that p16 is silenced in MCF10A, as mentioned in our Discussion.
- Interstitial changes are very rare in BJ-5ta and IMR-90, consistent with and this further bolstering our findings based on RPE-1 and MCF10A.

We have incorporated these additional data into the revised manuscript. Across all cell line models, we have now characterised a broad spectrum of CA accumulation responses, ranging from highly susceptible *TP53*-deficient models to p53 proficient cells with varying degrees of spontaneous CA formation. (A detailed overview of all experimental conditions in which we applied MAGIC is provided further above in **Table R1**.)

Figure S16 - CAs detected per cell in BJ-5ta and IMR-90 cell lines (A) and after reversine treatment in MCF10A and RPE-1 (B). Breakdown of CA classes in the BJ-5ta and IMR-90

cell lines in absence of perturbation (**C-F**), as well as in MCF10A and RPE-1 following perturbation with reversine (**G-J**).

Comment #1.04: *d) The use of Strand-seq provides homolog-specific profiling, which is important for CA characterization, but the throughput is limited to hundreds of cells. The authors demonstrated MAGIC combined with single-cell cloning and FISH, but it would have been illuminating and expand its impact to combine it with other higher-throughput single-cell methods (for example, scRNA-seq or scATAC-seq). However, I realize this may require a fair amount of work. But if it is feasible and not a big burden, I would encourage the authors to consider this.*

We thank the reviewer for encouraging us to explore the optional coupling of MAGIC with an additional single-cell method – ‘if feasible’. To highlight the versatility of MAGIC, we integrated MAGIC with scRNA-seq by adopting the SMART-seq2 protocol⁶, as a proof-of-principle demonstration. We note that SMART-seq2 is plate-based⁶ (like Strand-seq⁷), facilitating efficient integration into our workflows during manuscript revision.

Using SMART-seq2, we performed comparative scRNA-seq on micronucleated vs. normal cells in the MCF10A and RPE-1 cell lines. In total, we sequenced 170 micronucleated and 76 normal cells from MCF10A, and 84 micronucleated and 71 normal cells from RPE-1 (**Fig. S17A**). We observe a somewhat higher quality for the newly generated MCF-10A dataset, resulting in the identification of a higher number of differentially expressed (DE) genes (**Fig. S17A**). Nonetheless, data quality was acceptable in both scRNA-seq experiments, allowing us to infer DE genes using Fast Gene Set Enrichment Analysis (fgsea) – revealing enriched pathways from the “hallmark collection” of the Human Molecular Signatures Database (MSigDB), where each gene set represents specific well-defined biological states⁸.

This analysis provides additional insights into the effects of spontaneous micronucleation in these cell lines: specifically, we observe shared suppression of proliferation markers in both MCF10A and RPE-1 for micronucleated cells, as indicated by the downregulation of hallmark gene sets such as E2F_TARGETS, G2M_CHECKPOINT, and MYC_TARGETS (V1 and V2) (**Fig. S17B**). This finding aligns with our live-cell imaging based data, which show cell cycle arrest in micronucleated cells (**Fig. S1D**). In MCF10A, we detect activation of the P53_PATHWAY hallmark gene set (**Fig. S17B**), consistent with p53-driven cell cycle arrest. This finding is further bolstered by the lack of cycle arrest during live-cell imaging of *TP53* $-/-$ models following micronucleation (**Fig. S8H**). We also observe significant upregulation of *TP53* (logFC: 1.655, adj. p-value: 1.439e-16) in micronucleated RPE-1 cells, implying a similar process is at play in these cells.

Exposure of micronucleus content to the cytoplasm has previously been reported to activate the cGAS-STING pathway, leading to NF- κ B activation and the expression of IFN-gamma response genes and IL-6.⁹ In accordance with this, pathway enrichment highlights activation of INTERFERON_GAMMA_RESPONSE, IL6_JAK_STAT_SIGNALING, and TNFA_SIGNALING_VIA_NFKB pathways in MCF10A cells. (**Fig. S17B**). Gene Ontology (GO) enrichment analysis further corroborates these findings, emphasizing suppression of proliferation and metabolism, particularly through the downregulation of genes related to DNA replication and chromosome segregation (**Fig. S17C**). Cell migration pathways are

uniquely enriched in MCF10A, suggesting additional biological processes specific to this cell line (Fig. S17C).

We have incorporated these new data into our revised manuscript, showcasing the versatility of MAGIC and further bolstering findings with respect to the biology of micronucleated cells.

Figure S17 - Differential expression analysis of micronucleated cells

(A) Volcano plots of genes identified by coupling the MAGIC platform with single-cell RNA sequencing, contrasting normal vs. micronucleated cells in MCF10A (left panel) and RPE-1 (right panel) wild-type cells. Differentially expressed (DE) genes are in blue if down-regulated or red if up-regulated. (B) Enrichment analysis for Hallmark gene sets from the Human Molecular Signatures Database (MSigDB)⁸. Upper panel MCF10A, lower panel RPE-1. (C) Heatmap plot summarizing gene ontology (GO) enrichment for MCF10A (left panel) and

RPE-1 (right panel). Redundant GO terms were clustered by semantic similarity, indicated by the blue to red heatmap gradient.

Comment #1.05: *e) MAGIC is limited to profiling micronuclei-containing cells, but CAs can also arise in the absence of micronuclei. So the view of CA processes exposed here may be biased as it is restricted to only those types of cells.*

While we indeed mainly focus on spontaneous micronucleation and while CAs can arise in the absence of micronuclei, we emphasise that MAGIC is not limited to the profiling of micronucleated cells *per se*. In the current study for example, we extensively apply MAGIC to profile both micronucleated and normal (round nucleus) cells. This approach enabled, among other analyses, the development of the agent-based model for estimating the de novo chromosomal aberration (CA) rate (**Fig. 3**).

Given the versatility of MAGIC, future studies could apply this platform to other nuclear atypia types serving as ‘potential imaging biomarker for cells at risk for CA formation’ – such as enlarged nuclei or chromatin strings, thereby facilitating the identification of CAs independent of micronucleus formation. However, our systematic live-cell imaging analyses have revealed micronuclei to be, by far, the most frequent form of nuclear atypia observed (**Fig. 1D**) – supporting our decision to prioritise this class of nuclear abnormalities in our study. While exploring additional nuclear atypia would represent an interesting avenue for future work, it falls outside the scope of this paper. Taking this reviewer's comment into account, we have clarified in the revised Discussion section that future evaluations of CAs arising in association with other forms of nuclear atypia are feasible based on our platform.

Overall, the study is technically rigorous and well-presented, the authors make some notable discoveries regarding formation of isodicentric chromosomes and acentric amplified fragments, and the method will allow the field to systematically explore influences on CA formation, which is important for understanding cancer initiation. So I support publication, assuming the below specific comments are addressed.

Thank you for being supportive of our work. We address all specific comments below.

Introduction and abstract:

Comment #1.06: *1. The authors say they used ‘non-transformed’ cell lines, but then say that RPE-1 cells have hTERT. Doesn’t hTERT make it a transformed cell line?*

It is our understanding that ‘immortalisation’ refers to a cell’s unlimited proliferative potential, while ‘transformation’ denotes the process by which cells gain malignant properties, in particular, tumour formation. With respect to this terminology, we would like to refer to a recent “Cancer Research Landmarks” review article¹⁰ by Puleo and Polyak, which we now cite in our revised manuscript. The hTERT RPE-1 and MCF10A cell lines, derived from normal tissues, are immortal due to telomerase (hTERT) expression (introduced by genetic engineering means, or spontaneously), allowing indefinite culture. However, these cell lines are both non-transformed, since they do not form tumours in mice^{10,11}. While outside the scope of this current manuscript, we do anticipate future MAGIC experimental setups allowing to extend research into primary, non-immortalized cells. We have amended the Discussion to highlight the potential of MAGIC for such future application.

MAGIC:

Comment #1.07: 1. *The scale of the number of cells that can be interrogated per experiment should be described. It is mentioned in the Discussion, but more suited here.*

We thank the reviewer for this suggestion, and have revised the introduction to clarify the scale of the MAGIC system to readers. Specifically, the revised introduction now states that via automated imaging-based cell selection hundreds of target cells can be precisely isolated from a heterogeneous population of tens of thousands of cells.

Comment #1.08: 2. *Could the dyes used cause nuclear atypias and CAs, or perhaps a change in CA patterns? Was this evaluated in a control experiments?*

Thank you for this insightful comment. To address this point, we have performed additional control experiments during manuscript revision – measuring the frequency of micronucleated cells in MCF10A wild-type (WT; *i.e.* p53-competent) cells, with H2B-Dendra2 expression as well as after DACT-1 exposure, and compared these to frequency measurements made without any of these dyes ('no marker'; see **Fig. S1F**). Quantification was performed on fixed cells stained with Hoechst, allowing micronuclei to be detected using a single stain across all conditions. Importantly, based on our systematic analysis of almost 3,000 cells, we find that neither dye significantly affects micronucleus frequencies (**Fig. S1F**). These new control experiments are now referred to in the revised Supplementary Methods section). We further emphasize that MAGIC analyses performed across all cell lines yielded consistent patterns of *de novo* CAs under both labeling conditions (**Fig. 2, 3, S4, S9, S16**). Collectively, these data argue against any confounding effects of these dyes on CA formation.

Fig. S1F. Frequency of micronuclei in untreated MCF10A wild-type cells (no marker), compared with MCF10A wild-type cells that either express H2B-Dendra2 or were stained with DACT-1. No significant (n.s.) differences were observed (based on a $P=0.05$ significance threshold; Fisher's exact test).

Comment #1.09: 3. *Is it known whether brdU labeling can cause CAs? For example, by triggering DSBs due to DNA repair that recognizes the brdU?*

We thank the reviewer for raising this important point. It was previously reported that the relatively low BrdU concentrations utilised in Strand-seq do not significantly alter the occurrence of SCEs, implying that these BrdU levels do not cause chromosomal instability¹³. Our revised Methods section now refers to this prior report. Strand-seq – a single-cell technology involving BrdU labeling – provides a proxy marker for double-strand break (DSB) occurrence through the quantification of sister-chromatid exchanges (SCEs)¹². If the BrdU concentration employed in our study had induced DSBs, we would expect an increased frequency of SCEs, particularly at well-documented BrdU fragile sites. In our Supplementary Methods section, we previously mentioned that we did not detect SCE or breakpoint enrichment on fragile sites – supporting the conclusion that the BrdU concentrations used do not promote chromosomal instability. To further clarify this point, we have introduced two additional figure panels explicitly focused on BrdU fragile sites (see **Fig. S5A**).

Fig. S5A (panel detail) - Neither SCEs nor the inferred boundaries ('breakpoints') of CAs co-occur with previously reported BrdU fragile sites. Distribution derived from permutation of breakpoint location against BrdU fragile sites.

Comment #1.10: 4. *Automated identification of sister cell pairs from Strand-seq data is very interesting, but only two sister cell pairs were identified in the first experiment. Why is the fraction so low? Could magic be configured to automatically label sister cell pairs when one of the two cells is detected to have a micronucleus?*

We appreciate the interest in our novel approach for automated sister cell detection. We first would like to emphasise that our core data-driven approach – systematically comparing the spectrum of CAs in tens to hundreds of genome sequenced micronucleated vs. normal cells – does not rely on sister cell analyses. In our study, the analysis of sister cells provides confirmatory evidence, but it is not essential for our main conclusions. Consequently, our experimental design does not explicitly enrich for sister cells – their inclusion is stochastic.

Nonetheless, MAGIC could be amended in the future to allow increasing the collection of sister cells should this be deemed necessary, for instance, by extending the time between photolabeling and harvest or through cell synchronisation. We now elaborate on this point in the revised Supplementary Methods, where our approach to detecting sister cells from Strand-seq data is detailed.

Finally, with respect to the first experiment shown in **Fig. 1**, we find that the fraction of sister cells detected following targeted DSB induction is indeed low compared to our spontaneous micronucleation-based data (see **Fig. S4A**). CRISPR application is likely to impose particular

stress on the cells, which may ultimately lead to more frequent cell cycle arrest (possibly due to the sustained action of Cas9), and thus could have reduced the number of sister cells captured when compared to our experiments in spontaneously micronucleated cells.

Comment #1.11: 5. *Could MAGIC be combined with an automated cell picking instrument to allow linking of the cell image with single-cell sequencing? For example, the CellCelector instrument. If so, this possibility could be mentioned, even though out of scope of this paper.*

We refer to our response to **Comment #1.02** above – where we have covered this aspect in detail already. (Our revised Discussion section now alludes to the possibility to combine MAGIC with automated cell picking in the future.)

Landscape of spontaneous CA formation

Comment #1.12: 1. *Regarding the nonsignificant enrichment of CAs in the RPE cell line compared to MCF cell line: could this be due to differences in performance of the MAGIC machine-learning model in recognizing micronuclei of the two cell lines? In general, can the authors compare performance of recognizing micronuclei cells in each cell line?*

We emphasise that we developed cell line-specific ML models to ensure comparable performance across different cell lines – thereby adjusting prediction parameters so that the ML models achieve similar precision and recall. Additionally, the performance of micronuclei photolabeling in each experiment was visually assessed post-experiment, to ensure data quality. We stress that there are several lines of evidence supporting that we can effectively detect micronuclei across cell lines, using our ML models – including from new experiments added during revision stage:

- During the revision process, we utilised the same ML models to examine the CA spectrum after reversine treatment (inducing chromosome missegregation¹⁴) both for MCF10A and RPE-1 (for details, see response to **Comment #2.03** and **Fig. S16** further below). These new results, now referenced in our Discussion section, show that upon reversine treatment most micronucleated cells exhibit aneuploidies in both cell lines, in further support of our conclusion that our ML models effectively detect micronuclei in RPE-1.
- We further note that there is supporting evidence from sister cell pairs that the RPE-1 cell line acquires *de novo* CAs at a comparably low rate: Particularly, out of 11 sister cell pairs sequenced from WT RPE-1 cells, only 3 (27%) showed evidence for *de novo* CA formation – compared to 8/12 (67%) for WT MCF10A cells (see main text).
- Based on new experiments performed during manuscript revision we find varying extents of CA enrichment in micronucleated vs. normal BJ-5ta and IMR-90 cells (see our response to **Comment #1.03**). Thereby, BJ-5ta cells are more akin to MCF10A, while IMR-90 cells rather resemble RPE-1, supporting the notion of a spectrum in CA susceptibility across the cell lines assessed.
- We also emphasise that the low rate of CAs seen in RPE-1 cells undergoing spontaneous micronucleation may be explained with our data on the activation of the p53 pathway in micronucleated cells (see response to **Comment #1.04**). As we highlight in our Discussion, MCF10A cells have undergone p16 silencing during spontaneous immortalization¹⁵. This may have created a more permissive

environment for DSB-induced CA formation in MCF10A when compared to RPE-1. Interestingly, in our *TP53*^{-/-} cell line models, it appears that these phenotypic differences diminish, with both *TP53*^{-/-} cell lines (RPE-1 and MCF10A) showing much increased CA formation (**Fig. 3C**) consistent with p53-impaired cells proceeding through the cell cycle more efficiently^{4,16}.

Collectively, these data are in support of our conclusion that the lack of significant CA enrichment in RPE-1 cells reflects an inherent biological characteristic of this cell line (rather than a technical artifact from scoring micronuclei).

Comment #1.13: 2. *The number of sister cell pairs identified is again quite small, limiting the biological insights to anecdotes about each sister cell pair rather than large-scale statistics.*

We refer to our responses to **Comment #1.10 and #1.12** above, emphasising that our core findings – particularly, the CA spectrum characterization – have been performed without relying on sister cell analyses, that is, from a data-driven approach systematically comparing the spectrum of CAs in tens to hundreds of genome sequenced micronucleated vs. normal cells. During paper revision, we replicated and further generalised this approach by subjecting two additional cell lines to MAGIC (see our response to **Comment #1.03**).

Additionally, while we do not perform enrichment of sister cells in our experiments, we emphasise that our revised manuscript describes 23 sister pairs in connection with spontaneous micronucleation of wild-type MCF10A and RPE-1 cells. This exceeds pairs published in prior key studies in the field – for example Zhang *et al.*,⁴ a study analysing 9 sister cell pairs with micronuclei, as well as Umbreit *et al.*,¹⁸ which analysed 20 sister cell pairs following bridge breakage (see e.g. Fig. S7 in Umbreit *et al.*, for a summary). Therefore, even though we did not specifically enrich sister cells, the sister cell data acquired in our study compares favorably with the state of the art in the field. And while methods advancements that could further bring up sister cell numbers are outside of the scope of our current study, our revised Discussion section now highlights how additional future methods optimisations could enable the preferential enrichment of sister cell pairs based on MAGIC.

Comment #1.14: 3. *The RPE cell line is more stable and also has different CA patterns than the MCF cell line. But this diversity is explored in only two cell lines, and this raises the question of how generalizable these conclusions are to other cell lines, and potentially healthy tissues. Can the authors comment on this?*

To enhance the generalizability of our findings, we have now generated data from two additional non-transformed cell lines – BJ-5ta and IMR-90. These new data are largely consistent with and thus verify the findings reported in our initially submitted manuscript (see our detailed response to **Comment #1.03**). And while the rates of different CA processes in healthy tissues remain unknown, we emphasise that our study provides an important conceptual advance by examining these processes in four non-transformed cell lines originating from different cell types (which could be regarded as “models” for cellular behavior in precancerous contexts). Looking ahead, we foresee that future refinements of the MAGIC platform — including illumination and photolabeling in 3D, which could e.g. be

achieved using two-photon excitation microscopy¹⁹ — could offer deeper insights into chromosomal instability patterns in *ex vivo* model systems of a healthy tissue.

Comment #1.15: 4. *The authors show a difference in CAs between clonally expanded micronuclei-containing cells versus those selected directly for Strand-seq profiling to suggest that clonal expansion selects against some CAs. But even with direct profiling by Strand-Seq, there is required at least one cell division for the brdU to incorporate. Could selection already be acting in that one cell division so that even the CA profile of direct Strand-seq profiling already has a selection bias?*

We thank the reviewer for this insightful comment. While it is true that Strand-seq can be applied only to cells that have divided, in our experimental setup, BrdU incorporation does not necessarily happen after micronucleus generation. Indeed, when performing MAGIC in combination with Strand-seq, BrdU incorporation can occur in two different moments: (1) during the cell cycle prior to micronucleus formation; (2) during the cell cycle the micronucleus is present. This allows characterization of micronucleated cells regardless of the timing of BrdU incorporation.

Furthermore, with respect to selection, we emphasise that we do not observe substantial cell death in our MAGIC experiments, which argues against strong selective effects. But we acknowledge that occasional negative selection acting already during the first cell division cycle following micronucleation cannot be ruled out. This discussion point has been incorporated into our revised Discussion section, where we now also propose future methods advancements that could help investigate effects of negative selection.

Comment #1.16: 5. *Can Strand-seq detect circularization events? If this is feasible in principle, it would be interesting to quantify in this and the other study's experiments. If not, then this should be commented on in the Discussion, since these events are important types of structural events in cancer.*

We appreciate the reviewer's comment, which we understand refers to the formation of circular extrachromosomal DNA (ecDNA)^{20,21}. While Strand-seq does not directly detect circularization events, it does allow identifying high-level amplifications that frequently occur as a consequence of the propagation of ecDNAs²¹. We did not observe high-level amplifications in our spontaneous micronucleation based dataset, yet we cannot exclude early-stage ecDNA formation preceding amplification. At the same time, we note that while the isocentric chromosomes described in our manuscript retain chromosomal features such as telomeres (**Fig. 5A,C**), they appear to share with ecDNA the characteristic of asymmetric segregation into daughter cells (**Fig. 4G,H**).

In addressing this reviewer comment, we have revised our Discussion, which now alludes to the fact that future methods advancements – e.g. coupling MAGIC with single-cell sequencing techniques like scEC&T-seq²² – might in the future enable robust MAGIC-based ecDNA capture. We note that this would, once realised, further broaden the scope of MAGIC to include both chromosomal and extrachromosomal contributors to genome instability.

Prevalence of dicentric-mediated CAs and chromothripsis

Comment #1.17: 1. *Just to commend the authors that detection of chromothripsis in sister cell pairs is impressive.*

Thank you for this positive comment.

Chromosomal location determinants for CA formation

Comment #1.18: 1. *The section on FISH should describe in the text more quantitatively how many cells and how many karyotypes were analyzed.*

We have revised the respective figure legend (see revised **Figure 5**).

Comment #1.19: 2. *CRISPR cutting will generally occur on both sister chromatids, which can lead to some of the phenomena that the authors observe. But in the absence of CRISPR, what is the probability that randomly occurring DSBs will occur on both sister chromatids both in the sub-centromeric region, for example? Presumably this is significantly more rare, and therefore, would the isodicentric and isoacentric events observed relatively frequently here would be more rare in cancer and in healthy cells? The authors should comment on this, perhaps in the discussion and how this artificial CRISPR system may relate to what happens in vivo.*

We thank the reviewer for this insightful comment. In unperturbed cells, the probability of two DSBs occurring simultaneously on both sister chromatids at the same genetic locus is indeed very low. However, isochromosomes have been proposed to originate through a U-type exchange mechanism²³, which is triggered by a single DSB. This process would involve a single DSB in the G1 phase, subsequent DNA replication, and the fusion of the resulting broken sister chromatid ends²³. We anticipate that depending on which ends get fused, U-type exchanges could give rise to either isodicentric and isoacentric chromosomes, and could similarly result in the formation of a dicentric chromosome. Our revised Discussion section emphasises that the U-type exchange mechanism is simpler and thus more parsimonious than alternative models requiring simultaneous DSBs in two sister chromatids. The data presented in our manuscript are compatible with the U-type exchange mechanism.

Indeed, we find that the observed frequency of patterns compatible with isoacentric and isochromosome formation determined using Strand-seq is lower in conjunction with spontaneous micronucleation when compared to our CRISPR experiments (**Fig. S12G,H,I**). In the context of our targeted DSB experiments, we observe isoacentric formation to be ~10-fold elevated compared to the baseline observed in spontaneously micronucleated cells in MCF10A and RPE-1 lines (**Fig. S12F-H**). This implies that unrepaired DSBs arising within the interior of chromosomes are not among the most common triggers of spontaneous CAs in these cell lines, whereas our MAGIC system, when targeted DSBs are introduced, is capable of enriching for these specific CA processes. Irrespective of this, we repeatedly observe both isoacentrics and isochromosomes after spontaneous micronucleation (**Fig. S12F-I**). We also emphasise that isochromosomes – while arising more rarely spontaneously than after CRISPR application – form recurrently in several cancer types, such as medulloblastoma and lung cancer^{24,25}, e.g., with isochromosome i(17q) seen in >20% of medulloblastomas (**Fig. S14E**).

In addressing this reviewer comment, our revised Discussion section now emphasises that the U-type exchange process is the most parsimonious explanation for our data when

compared to alternative mechanistic models. Our revised manuscript also refers to the frequency of both isoacentrics (2.5%) and isochromosomes (1.8%) after spontaneous micronucleation in comparison to our CRISPR experiments in which the respective frequencies are about 9 and 8-fold increased, respectively (**Fig. S12H,I**).

Discussion:

Comment #1.20: 1. *The authors should discuss whether MAGIC could be coupled with other higher-throughput droplet-based methods, such as scATAC-seq or scRNA-seq. What would these methods be able to detect (CNVs, phenotypic changes) vs Strand-seq?*

The versatility of MAGIC means that this platform could be coupled with a range of single cell methods. For example, as described further above, our revised manuscript now comprises scRNA-seq data for micronucleated and control cells generated with the plate-based SMART-seq2 protocol. Our new generated data demonstrate, for example, activation of the p53 pathway in spontaneously micronucleated cells (see response to **Comment #1.04**). Similarly, one could envision coupling MAGIC with other single cell methods, as emphasised in the revised **Discussion** section. We have not yet implemented droplet-based methods into MAGIC. If realised, droplet-based methods could drive down experimental costs per cell, yet are anticipated to result in a reduced sensitivity per cell when compared to plate-based protocols²⁶ like Strand-seq and SMART-seq2.

Comment #1.21: 2. *The discussion is quite long and can be shortened.*

We appreciate the emphasis on conciseness. We have incorporated new suggested points by the reviewers while maintaining a clear and streamlined discussion section. We are, of course, open to additional shortening as per the editor's discretion.

Comment #1.22: 3. *Another limitation that should be mentioned is that Strand-seq cannot resolve the base-level sequences at the breakpoints.*

We have updated the Discussion accordingly.

Comment #1.23: 4. *“as well as the identification of elevated de novo CA formation in clinical samples. This could help identify high-risk precancerous lesions and might improve patient outcomes through timely interventions in the future.”: it is not clear to me how MAGIC could be applied to clinical samples. Either further details should be provided or this should be removed.*

We were envisioning future developments of MAGIC via incorporating two-photon excitation microscopy¹⁹, which would allow precise photolabeling in 3D tissue models, such as patient-derived organoids or ex vivo clinical samples. This enhancement would have the potential to capture spatially resolved molecular signatures in tumor tissues and precancerous lesions, potentially offering important insights into heterogeneity and chromosomal instability. However, since this functionality is contingent on future work in our laboratory, and given the need for conciseness, we have removed the respective statement as proposed by the reviewer.

Comment #1.24: 5. *“but also reveal the basal rate at which CAs emerge”: the authors should qualify this to say this is the rate in two cell lines.*

We agree, and have amended the sentence in the Discussion, to clarify that we have determined the basal rate at which CAs emerge in cell culture.

Figures:

Comment #1.25 1. *Figs. 3A-E would be easier to interpret with side-by-side comparisons of WT vs p53 cells*

While we appreciate the reviewer's suggestion to enhance clarity through a side-by-side comparison, the narrative structure of our manuscript prevents us from positioning the data from Figs. 2 and 3 onto the same main display item. Nevertheless, to address this reviewer comment, we have modified **Fig. 3** — which in our view particularly benefits from direct comparative visualisation — by incorporating a panel from **Fig. S9**, explicitly depicting fold changes for each of the four conditions, thereby directly comparing *TP53*^{-/-} to WT cells.

Comment #1.26: 2. *Fig. 3G y-axis label should make more clear this rate is estimated from a model, not directly measured*

We have changed the y-axis label to “estimated rate of SV generation”.

Comment #1.27: 3. *Figs. 5C,D would benefit from some schematic showing the structure of the exhibited CA. Also, if feasible, the authors should reduce the size of each of these panels to allow showing one karyotype example for each type of CA.*

We thank the reviewer for these thoughtful suggestions. We have now integrated a scheme of the chromosome structure alongside each FISH example shown in **Figs. 5** and **S13**, as suggested. In **Fig. 5**, these depicted structures include one example of a normal chromosome 7, one example of an isochromosome (isodicentric), one example of an isoacentric, and one example of amplified isoacentrics.

Referee #2 (Remarks to the Author):

In this manuscript, the authors present a platform called MAGIC, which couples automated microscopy in live cells, evaluation of nuclear abnormalities via machine learning, targeted cell photolabeling and cell sorting. The isolated cells are then subjected to single-cell sequencing or phenotypic analyses. This approach has been applied to two non-transformed near-diploid cell lines, MCF10A and RPE1. The authors profiled mitotic errors in the unperturbed cell lines using MAGIC followed by Strand-seq, thereby mapping spontaneous chromosomal aberrations (CAs) that arise from micronuclei. Interestingly, this enabled the authors to reconstruct chromosomal aberrations arising across successive cell cycles. A comparison of p53-deficient vs. p53-proficient MCF10A and RPE1 cells confirms that p53 inactivation increases the rate of micronucleus formation and chromosomal aberrations, with more complex CAs emerging in micronucleated p53-null RPE1 cells. CRISPR-Cas9-based induction of double strand breaks (DSBs) revealed that certain CA processes were influenced by the initiating DSB chromosomal location (for example, sub-centromeric DSBs induced isochromosome formation), and FISH was used to validate the Strand-seq results and further interrogate these CAs.

We thank the reviewer for outlining the main advances made by our study and the constructive comments.

Comment #2.01 *This is an interesting and timely paper, which provides some insights into the de novo formation of chromosomal aberrations. The integration of automated imaging and genomic profiling is powerful. Nonetheless, the study remains largely descriptive, and to justify publication in Nature, I'd propose extending it beyond the technology development in at least one of the following directions:*

We thank the reviewer for acknowledging the novelty and relevance of our approach, and particularly also for providing optional recommendations for extending our work. Our detailed point-by-point responses to these reviewer propositions can be found below.

We also want to take the opportunity to briefly respond to the point about our study being descriptive: While we do concur that our data-driven imaging-and-genomic study has a strong descriptive component, there are in our view notable findings that we believe yield important biological insights:

- We perform modeling to derive the first estimate of the per-cell division CA rate in p53-competent and mutant cells, and demonstrate that *TP53*-loss effectively doubles the rate of *de novo* CAs per cell division. (This is also highlighted as a particularly significant achievement by **Reviewers #1** and **#4**).
- Our discovery that targeted DSB lesions in distinct chromosomal regions result in specific types of CAs also seen in cancer genomes – such as subcentromeric DSBs, which tend to trigger isodicentrics. In doing so, as noted by **Reviewer #1**, our study offers a mechanistic basis for the formation of two types of CA – *i.e.*, isodicentric and isoacentric formation, processes resulting in the coordinated and asymmetric segregation of DNA material in multiples of two.

Comment #2.02 (1) *Making the MAGIC pipeline and data as accessible and easy-to-adopt as possible: the application of the proposed pipeline in other labs is far from trivial, and the authors need to make an effort to make all steps – experimental and computational alike – accessible and easy to use. The authors have already made the code available, which is valuable. Detailed protocols and tutorials for the experimental aspects of the work should be added as well. In addition, please provide the single-cell CA data for the 2,036 analyzed single cells in a format that will facilitate additional analysis of the data.*

We agree on the importance of making MAGIC widely accessible. During manuscript revision, we worked on further extending the protocol descriptions, to simplify its implementation elsewhere. A detailed protocol can be found in the following repository: https://git.embl.de/cosenza/magic_automation/-/blob/main/docs/MAGIC_protocol.md.

Additionally, as mentioned in **Comments #3.12** and **#4.07**, we have simplified the setup process for our computational tools by developing a Docker container.

Finally, while we previously already released our single-cell raw data, we have made the count data for all single-cell experiments available to facilitate data re-analysis. Here below is a link to the zenodo entry draft:

https://zenodo.org/records/15262424?preview=1&token=eyJhbGciOiJIUzUxMiJ9.eyJpZCI6ImQxN2I5MmU1LWVhZGZlZGEzZTdhYTA2MDAyMTZkNjA4OGMxOGM2OWRjOWQwNCJ9.uQ7dwY-m7F60cg4z8Vg3Ovm8EeTgdw-Pq3wYyzl_8GN48t51dXzBmqMAgc4AySg24D4loe79h_zhak3IID7ACA

Comment #2.03 (2) *The manuscript currently focuses on spontaneous aberrations that arise in two non-transformed cell lines. Would the findings hold true for micronucleation induced by perturbing chromosome segregation? In other words, is there a difference between spontaneous micronucleation, and micronucleation that is induced by common methods to induce CIN, such as MPS1 inhibition? Such comparison would be very valuable for the field as it could reveal whether the common methods for CIN induction mimic the “natural” CIN that exists in these cells.*

We thank the reviewer for this comment, and agree that such a comparison would be valuable for the field. We have therefore conducted additional experiments using reversine — a well-established MPS1 inhibitor¹⁴ — to induce chromosome missegregation in WT MCF10A and WT RPE1 cell lines. These cells were treated with reversine for 24 hours, followed by washout and release in BrdU and isolation via MAGIC about 24 hours after release. We employed MAGIC to collect both normal and micronucleated cells. To allow comprehensive analyses with the data generated, we sequenced 194 single cell genomes from these experiments, with 1,009 CAs detected in total.

Several noteworthy observations emerge from these data: First, reversine leads to the extensive accumulation of whole chromosome aneuploidies in both MCF10A and RPE1. Nearly all cells are affected – whereby we find slightly higher numbers of whole chromosome aneuploidies in micronucleated cells compared to cells with normal nuclei (**Fig. S16B**; shown in connection to **comment #1.03**). These high numbers of aneuploidies are in line with previous studies and the known role of MPS1 in regulating chromosome attachment to the spindle and spindle assembly checkpoint (SAC), with MPS1 inhibition causing extensive chromosome missegregation events^{1,14,17}. Thereby, MPS1 inhibition results in chromosome missegregation, via the stabilization of improper kinetochore-microtubule attachments, leading to segregation defects and lagging chromosomes with and without micronucleus formation¹. This results in a CA spectrum dominated by whole chromosome aneuploidies, revealing notable differences to the CA spectrum of spontaneously micronucleated cells (compare **Fig. S16G-J** to **Fig. 2A,D,E**).

Notably, we find that reversine-driven CAs accumulate in high numbers even in RPE-1 cells (**Fig. S16B,I,J**), which do not significantly accumulate CAs when micronuclei originate spontaneously (**Fig. 2A**). Furthermore, under MPS1 inhibition, whole chromosome gains and losses occur at roughly equal numbers in a p53-proficient background – whereas across all our experiments from spontaneously micronucleated cells, a chromosome loss enrichment over chromosome gains is clearly apparent and high significant ($P < 1e-05$; see **Supplementary Table 10** and **Fig. S16G-J**). As MPS1 inhibition can result in segregation defects both in the presence and the absence of micronucleus formation¹, these new data suggest that the fate of micronuclei might mediate the strong bias towards chromosome losses seen after spontaneous micronucleation.

We conclude it is very likely that the notable differences in the CA spectrum are grounded in the different CA triggering mechanisms. In the case of MPS1 inhibition, chromosome missegregation can occur without activating the SAC and results in aneuploid cells with the ability to further proliferate^{14,27}. On the other hand, our analysis of single cell genomes sequenced from spontaneously micronucleated cells reveals dicentric chromosomes as one

of the most common sources of *de novo* CAs (**Fig. 2A,B,C,D,E**). When compared to chromosome missegregation, these lesions involve DSB formation, and thus imply an increased probability of cell cycle arrest via the *TP53* pathway (**Fig. S1D**, see also **Fig. S17** our response to **Comment #1.04**). When disrupting *TP53* in RPE-1 cells, the *de novo* CA prevalence is substantially increased in conjunction with spontaneous micronucleation (**Fig. 3**), consistent with p53-impaired cells proceeding through the cell cycle more efficiently^{4,16}.

In summary, our experimental data reveal distinct CA landscapes between chemically induced chromosomal missegregation and spontaneous micronucleation. This distinction likely originates from differences in the initial CA-inducing events, and highlights DSBs-triggered CA formation as being strongly influenced by the cell line's *TP53* status, the latter of which can influence the cell cycle arrest probability in cells subject to DSBs. This also underscores the capability of MAGIC to elucidate the underlying mechanisms responsible for *de novo* CA. We discuss these data comprehensively in the revised Discussion section of the manuscript, with further details provided in the Supplement.

Comment #2.04 (3) *The authors begin to address the question of de novo formation of CAs vs. their selection. Looking deeper into selection would be of much interest. (I'd like to leave leeway for the authors to address this point in whatever ways they see fit.)*

We thank the reviewer for this comment, prompting us to examine patterns of selection more closely, and we also thank the reviewer for leaving it to us how to address this point. In our initially submitted manuscript, we reported a particularly low tendency of clone survival in micronucleated RPE-1 cells compared to MCF10A cells (**Fig. S7D**). To further explore potential selective pressures contributing to this observation, we have now performed low-pass whole-genome sequencing on 11 single-cell-derived clones originating from WT RPE-1 cells that survived spontaneous micronucleation. As we described in our originally submitted manuscript, our analogous low-pass sequencing based analysis of single-cell derived clones expanded from micronucleated MCF10A cells provide robust evidence of CA propagation, and also highlight a potential example of positive selection (*i.e.* an enrichment of 7q-arm loss events; see **Fig. S7F**). Interestingly, while we find somatic CAs in 9 out of 18 (50%) single-cell derived clones from MCF10-A micronucleated cells, our new data reveal that RPE-1-derived clones show complete absence (0/11; 0%) of clonally maintained *de novo* CAs (**Fig. S7E, left panel**). This discrepancy between MCF10A and RPE-1 is significant ($P < 0.0052$, Fisher's exact test), and suggests a markedly reduced tendency of RPE-1 cells to maintain newly formed CAs during culturing. Additionally, our reanalysis of all sister cell pairs generated following spontaneous micronucleation suggests that MCF10A sister cell pairs are ~2.4-fold more likely to harbor *de novo* CAs than RPE-1 sister cell pairs (**Fig. S7E**; see right panel) – in further support of our observation that *de novo* CAs arise only rarely in RPE-1 upon spontaneous micronucleation.

Collectively, our data indicate that RPE-1 cells exhibit a reduced ability to tolerate both the initial formation, and the subsequent propagation, of *de novo* CAs – implying particularly strong selective constraints acting in this cell line. These data also show that micronucleation does not always lead to CAs, as nuclear atypia can be resolved without karyotype change. We have integrated these new data into our revised manuscript.

Fig. S7D Fraction of surviving clones after MAGIC isolation.

Fig. S7E - *De novo* CA formation in two cell lines measured from chromosomally abnormal propagated clones as well as sister cell pairs. Left, percentage of clones carrying at least one *de novo* CA, originating from micronucleated cells. Right, frequency of micronucleated sister cell pairs carrying at least one *de novo* CA.

Comment #2.05 *Below are a few specific suggestions for additional analyses that could strengthen the main conclusions of the study.*

We are grateful to the reviewer for proposing avenues that could further strengthen the main conclusions of our study. We have addressed each point thoroughly, as detailed in the reviewer responses provided below.

Comment #2.06 • *Can the observed CA formation be linked to the CNA signatures reported by Steele et al. and Drews et al. (Nature 2022)? Please evaluate and discuss.*

We agree that it is of interest to compare the observed CA formation processes with CNA (CIN) signatures from prior studies. We emphasise that the studies by Steele et al. and Drews et al. were conducted using data from advanced cancers, bearing highly evolved

karyotypes, and therefore typically more aberrant CAs when compared to the non-transformed cell lines from our study. Over time, signatures of discrete CA events are likely to be overlaid in highly evolved cancer genomes, potentially masking their actual mechanistic origin^{28,29}. We thus explored different methodologies in search of one that would allow for a meaningful and fair comparison between models representing different stages of CIN (non-transformed cell lines vs. advanced cancers), as well as between data obtained through different technologies (Strand-seq vs. bulk whole-genome sequencing). While we initially considered both the methodologies from Steele et al. and Drews et al., we emphasise that both methods come with challenges in relation to comparisons with our study. Below, we first clarify observations we made with respect to comparisons across methods and sample types. We then detail the steps we followed to adapt features from the Drews et al. method, ensuring suitability for comparative analysis with our approach:

In particular, the CNA signatures described by Steele et al. are derived from three main features: [1] the length of copy-number (CN) segments, [2] the total estimated CN, and [3] the presence or absence of loss-of-heterozygosity (LOH) events. While these features provide critical insights when applied to advanced cancers, their utility for analyzing *de novo* CAs presents some limitations. This is due to the fact that only the length of CN segments (feature [1])²⁹ is universally applicable for (*i.e.* comprehensively represented in) our single-cell based *de novo* CA data. LOH events in our data are limited exclusively to cases of single-copy losses (CN=1), and the overall diversity of CN states observed in the non-transformed cell lines in our study is substantially lower compared to those reported in advanced (*i.e.* highly evolved) cancers^{28,29}. As a consequence, application of the Steele et al. feature set to our data would provide a relatively small additional informative value, which we reasoned would making novel or interesting observations less likely.

By comparison, the Drews et al. signatures are derived from more features: [1] the length of non-diploid segments, [2] the density of changepoints per 10 Mb, [3] the density of changepoints per chromosome arm, [4] the magnitude of CN changes, and [5] the length of oscillating chains of CN changes. These features are hence more informative (*i.e.* more universally applicable) when applied to our data, as they contribute a broader range of populated values. Furthermore, the fact that ‘total CN’ is not employed as a feature by Drews et al. enables comparisons across samples with varying average ploidy levels. Nonetheless, we noted certain aspects of the Drews et al. signatures that limit their direct applicability to *de novo* CA analysis in our data – resulting in the need for some further adaptation:

- We realised that the Drews et al. method, in its published version, is not designed for analysing CA patterns arising during the earliest stages of somatic evolution, as samples harboring fewer than 20 CAs are not considered by Drews et al. – these sample are, indeed, by default filtered out automatically. If we kept this default filter from Drews et al., this would exclude the vast majority of our data.
- When relaxing the requirement of minimum number of CAs per sample, we noticed a general tendency of the Drews et al. method to report a larger contribution from the signature “CX1” in genomes with fewer and simpler CAs, implying certain limitations in directly extrapolating the method by Drews et al. to early-stage somatic evolution patterns. This appears to be due to the fact that when the number of CA breakpoints in a sample resource is low, the Drews et al. method, when applied to samples carrying particularly few CAs, tends to automatically report CX1. Since most *de novo* CAs observed in a given MCF-10A genome have relatively few breakpoints and

affect one or few chromosomes, this behaviour would therefore have resulted in a marked enrichment of CX1 in our data – which we concluded is less informative with respect to the molecular processes driving CA formation events.

After careful consideration and recognising that the Drews et al. method was calibrated for advanced tumors – not for samples suspect to early stage somatic evolution – we therefore chose to make the following amendments to the methodology (described in detail in the revised **Supplemental Material**):

- [i] We performed the analysis per chromosome instead of per sample/genome. This prevented *de novo* CAs from becoming diluted within otherwise uneventful genomes where most chromosomes do not contain a single CA (which would otherwise lead to the CX1 enrichment mentioned above).
- [ii] We adapted the hard filter introduced in Drews et al. to remove chromosomes harbouring fewer than 2 CAs (lowest possible threshold within the framework of Drews et al.), instead of the original filter that removes genomes harbouring fewer than 20 CAs.
- [iii] We performed the comparison at the level of the features introduced by Drews et al. instead of using the signatures directly. Computation of CNA features is an intermediate step in signature extraction in the Drews et al. study. As such, we reasoned CNA features would not only be sufficient for comparisons by chromosome, but also enhance interpretability as each feature represents a clear discernible signal relevant to the CA landscape.

The following analysis results emerged from applying this amended approach: when jointly clustering our MCF10A single cell genomic data with the bulk cancer genomic data from PCAWG, we find five clusters, four of which contain both MCF10A and PCAWG data (see **Supplemental Material; Fig. S15**). Two clusters (clusters 1 and 2; containing 43% of PCAWG chromosomes) are notably enriched in MCF10A. These clusters are characterized by relatively simple alteration patterns likely to arise during early somatic karyotype evolution. The remaining clusters represent patterns of CAs that are either (i) below the limit of detection of Strand-seq (<200 kb in length), or that (ii) are likely to be derived from several CAs overlapping one another, and thus compatible with later stages of cancer karyotype evolution. From these comparisons, we conclude that although there are some remaining challenges in applying CNA signature methodologies to CA patterns at early stages of tumour evolution, the *de novo* CA patterns we observe in association with spontaneous micronucleation are generally compatible with those documented in advanced cancers.

Comment #2.07 *CAs were strongly enriched in micronucleated MCF10A cells, but not in micronucleated RPE1 cells. This is quite surprising. Is this also true if micronucleation is induced in these cells by perturbing chromosome segregation?*

We thank the reviewer for raising this question, and in this regard also refer to our responses to **Comments #2.03** and **#2.04** above. We note that there is evidence also from sister cell pairs that the RPE1 cell line acquires *de novo* CAs at a rather low rate: out of 11 sister cell pairs sequenced from WT RPE1 cells, 3 showed evidence for *de novo* CA formation – compared to 8/12 for MCF-10A (**Fig. S7E**). Furthermore, we find that clones derived from micronucleated MCF10A cells frequently retain newly generated CAs (seen in 9/18 single-cell derived clones from MCF10-A micronucleated cells). In contrast, we find that clones derived from micronucleated RPE-1 cells show absence (0/11) of detectable CAs

after micronucleation ($P < 0.0052$, Fisher's exact test; see **Fig. S7E**). This suggests a substantially reduced tendency of RPE-1 cells to clonally propagate newly formed CAs, as discussed in more detail in our response to **Comment #2.04** above.

Nonetheless, we find the reviewer's suggestion to investigate RPE-1 cells with MAGIC following perturbation of chromosome segregation very interesting. We thus employed the MAGIC platform to investigate the CA spectrum in response to perturbation with reversine, an MPS1 inhibitor that induces whole chromosome missegregation¹⁴ (to avoid redundancy, full details are in our response to **Comment #2.03**, and in **Fig. S16**; see below). These new results reveal that RPE-1 cells are indeed very susceptible to CA accumulation via whole-chromosome missegregation. Specifically, reversine results in a CA spectrum characterized by whole-chromosome aneuploidies, consistent with perturbation of chromosome segregation. This CA spectrum shows substantial differences to the CA spectrum seen after spontaneous micronucleation, where DSB/dicentrics mediate CAs formation (see response to **Comment #2.03**). As inhibition of MPS1 can induce segregation errors both with and without micronucleus formation¹, it is plausible that micronucleus formation specifically, rather than chromosome missegregation *per se*, constraints CA formation in RPE-1 cells—a point that we now explicitly address in our revised Discussion.

Collectively, our data suggests that the lack of significant *de novo* CA enrichment in RPE1 cells reflects a genuine biological characteristic of this cell line. Notably, MCF10A but not RPE-1 underwent p16 silencing during immortalization¹⁵. This may have created a more permissive environment for DSB-induced CA formation, and as such could explain some of the observed phenotypic differences of these cell lines. Furthermore, the finding that *TP53* knock-out fosters CA accumulation in both MCF10A and RPE1 (**Fig. 3**) is consistent with a role of DNA damage surveillance in limiting CAs originating from DSBs. These findings reinforce the value of MAGIC in investigating distinct molecular triggers of CAs.

Comment #2.08 *In MCF10A, CAs were enriched for chromosome 19. MCF10A harbors several clonal arm-level chromosomal gains, presumably due to unbalanced translocations – are those events not enriched for CAs? In RPE1, CAs were enriched in specific chromosomes – does the rate of CAs correlate the rate of chromosome mis-segregation previously reported in these cells (Klaasen et al. Nature 2022)?*

We thank the reviewer for these insightful questions. Regarding the MCF10A cell line, we confirm that we do detect the previously described clonal CAs in this cell line, including isochromosome 1q or chromosome 8 gain. Importantly, our statistical approach for identifying *de novo* CAs explicitly controls for pre-existing clonal and subclonal alterations. Thus, these pre-existing clonal events are not misclassified as *de novo* unless accompanied by additional, newly arising CAs. Although we do not detect enrichment of *de novo* CAs colocalizing with these clonal CAs in MCF10A, we find evidence that the derivative chromosome der(X) t(X;10), clonally propagated in RPE-1 cells, exhibits an elevated frequency of *de novo* CAs (**Fig. 2G**).

We have further specifically addressed the reviewer's request concerning the chromosome-specific alterations in RPE-1 cells, by comparing our data to the chemical micronucleus induction results described by Klaasen et al. (2022). In our study, we do observe a tendency for CAs to affect larger chromosomes in RPE-1 cells, such as

chromosomes 2, 6, 10, and X (**Fig. 2G**). Notably, these chromosomes have likewise been documented by Klaasen et al. (2022) as frequently undergoing alterations after chemically-induced micronucleation. Accordingly, we measure a significant correlation ($P = 0.027$) between chromosome-specific *de novo* CA frequencies in our study and those observed by Klaasen et al. (2022), illustrated in the newly added **Fig. S5D** in our revised manuscript.

Fig. S5D. Percentage of aneuploid chromosomes in RPE-1 across studies. Correlation in aneuploid chromosome percentages in RPE-1 cells as measured from spontaneous micronucleation (our study) when compared to previously generated data based on induced missegregation¹⁷. Dots represent chromosome-specific aberration frequencies in both studies.

Comment #2.09 *The results may help explain recent findings of large chromosomal losses following CRISPR-Cas9 genome editing (Nahmad et al. Nat Biotechnol 2022; Tsuchida et al. Cell 2023). This should be discussed.*

We thank the reviewer for this insightful comment. We have included both citations in our revised Discussion.

Comment #2.10 • *In RPE1 and MCF10A cells, previous studies found that whole-chromosome monosomies were viable only on a p53-deficient background. Was there a difference in the prevalence of gains and losses upon p53 inactivation? This should also be mentioned in the discussion of this topic.*

We appreciate this comment, and clarify that we find a higher prevalence of chromosome losses over gains both in p53 proficient and *TP53*^{-/-} cells (multiple-testing adjusted $P < 0.05$ seen both for some p53-proficient and -deficient conditions; see **Supplementary Table S10**, highlighted in our response to **Comment #1.03**). We further emphasize that MAGIC enables investigating the *de novo* CA rate of chromosome gains and losses, which is conceptually different from assessing their post-selection distribution (which prior studies have addressed). The post-selection distribution might be strongly influenced even by subtle viability effects, such as negative selection against monosomy. In contrast, *de novo* CA formation rate estimates are unlikely to be affected substantially by subtle viability effects.

Following this reviewer's suggestion, we have revised our Discussion section to clarify these aspects.

Referee #3 (Remarks to the Author):

This work introduces MAGIC, a novel platform to integrate live-cell imaging, machine learning and single-cell genomics.

My comments are limited to the image analysis aspects of this work, including the software provided by the authors.

Comment #3.01 *Image segmentation and classification can be approached in many ways. The strategy here combines conventional image processing with pixel and object classification using XGBoost. This is not in itself conceptually novel, but the details are adapted for the application and it represents a suitable pragmatic approach. The primary question is whether it can achieve sufficient accuracy to support the downstream analysis.*

We thank the reviewer for acknowledging our approach – combining conventional image processing with pixel and object classification via XGBoost – as pragmatic. We point out that this approach aligns with the rationale used by Ilastik³⁰, a method that favors Random Forest classifiers for their simplicity, robustness, and good generalization performance, qualities that make them particularly well-suited for training by non-experts. As such, MAGIC is by-design a flexible platform for experimenters, which allows users to apply its technology to a variety of experimental objectives beyond isolating micronucleated cells. We anticipate that users are unlikely to have access to large, pre-annotated datasets and will often be working on a single, highly specialized task, typically involving a specific experimental model. Our pipelines are thus designed to enable rapid development of customized detection models for specific applications, allowing the set up of tailored pipelines. We obviously agree in this regard that the accuracy of our approach is of primary interest. Thus, as outlined in our response to **Comment #3.10**, we have added **Fig. S18A** during manuscript revision, where we verify that model performance at training matches performance in “real-world” experiments.

Main comments:

Comment #3.02 - *Accurately segmenting nuclei and micronuclei is a key step. The approach here is to first generate a custom semantic segmentation with XGBoost and Gaussian/Hessian features, and then post-process to generate instances. Based on the strategy and examples provided, one expects there to be issues with nuclei being missed, fragmented, or clumped together.*

-- Was the accuracy of the segmentation quantified? This could be using exhaustively annotated images from the study, or a publicly-available benchmark (e.g. from the Broad Bioimage Benchmark Collection).

We thank the reviewer for raising this point and the opportunity to clarify the philosophy we adopted in developing MAGIC. We assessed segmentation accuracy within the specific scope of our experiments and the cell lines used in this study, rather than against publicly available benchmarks. While we did consider that our approach could be susceptible to segmentation issues, an indication of sufficient quality for segmentation accuracy is implicit

in the global performance of the entire classification pipeline (see **Fig. S2H**), which also matches that of real-world experiments as described in **Fig. S18A** (see **Comment #3.08** below for more details).

We stress that our primary objective was not to create a broadly generalizable model for micronuclei classification, but to rapidly deploy a computer vision pipeline tailored to the requirements of our particular experimental setup. Our segmentation and classification models were thus designed to perform optimally for the experimental questions, with no intent to extend beyond them (see also **Comment #3.01**).

Additionally, we emphasise that our experiments can tolerate a certain level of false negatives, as our strategy involves sampling from a population rather than achieving exhaustive detection. As long as the photolabeled objects are representative of the target population, our analysis results remain robust (see **Comment #3.08** below for a wider discussion on potential model biases).

Comment #3.03 -- *Were other approaches considered? These could either be simpler (e.g. image processing without machine learning), or more complex (e.g. an established deep learning strategy like StarDist or CellPose).*

We acknowledge the potential for simpler, and also for more advanced segmentation models like StarDist or CellPose to be beneficial in certain applications and we did consider this during method development. In our case, we tested StarDist but found it produced suboptimal results. This is because atypical nuclei violate a key assumption of StarDist, which is that nuclei are star-convex. Atypical nuclei, particularly in the region surrounding the micronucleus, can be highly concave, leading to poor segmentation in this critical area, which reflects the main focus of our analysis. In addition, when we started the development of MAGIC, CellPose was not available yet. However, we stress that future users of MAGIC have the option to integrate such state-of-the-art models into their custom pipelines if they believe these models will add value for their specific experiments. This flexibility ensures that MAGIC can be adapted to diverse segmentation needs, while maintaining the practicality and accessibility that our pragmatic approach emphasizes. We do appreciate this comment and have amended the Discussion section to clarify that deep-learning approaches like CellPose could in the future be integrated into MAGIC.

Comment #3.04 - *The manuscript describes 'real-time' detection of micronuclei. The term 'real-time' is ambiguous, but the actual detection takes at least a few seconds (both on my computer, and reported in the paper) - which seems quite slow.*

-- To what extent is fast processing critical for the application?

We thank the reviewer for this feedback, and we do recognize that the term "real-time" can be ambiguous and might suggest inference speeds in the millisecond range. In our case, indeed, the inference occurs over a few seconds. For the biological context examined here, a timeframe extending even into minutes would still be appropriate, as significant cell movement is unlikely in such intervals. Following this reviewer comment, we have modified our manuscript to refer specifically to "on-the-fly" detection of micronuclei, thus providing an improved depiction of our analytical timing.

Comment #3.05 -- *I would guess that the Hessian features are contributing substantially to the processing time; were these shown to be necessary?*

The final selection of which operations to apply was based on feature importance for the XGBoost model and this included Hessian features. In any case, the computation of features is parallelized to reduce processing time. We edited the Supplementary method section to clarify this. While computing these features can be time-consuming depending on the scale of the image, we find that the inference time is well within the required range for the biological context examined in this study (see **Comment #3.04**).

Comment #3.06 -- *Was GPU acceleration considered?*

We did consider GPU acceleration for our computer vision pipelines, but ultimately found it unnecessary for this application and biological context (we also refer to our responses to **Comments #3.04** and **#3.05** in this regard). Given that our models already achieve inference times within a few seconds on modern personal laptops, the added complexity and cost of GPU implementation was not essential. Furthermore, we feel that relying on CPU processing reduces the expertise required to run these tools, making them more accessible to researchers who may not have specialized knowledge in machine learning or access to high-performance computing resources.

Comment #3.07 - *Classification models were trained to achieve at least 90% precision and 50% recall.*

-- *Do these values reflect classification *only*, or are errors accumulated during the segmentation also incorporated in some way?*

--- *I think that precision and recall per-image would be most relevant - i.e. considering all objects, and not restricted to only those that were correctly segmented.*

We thank the reviewer for this question and the opportunity to clarify this point. We agree with the reviewer that precision and recall considering all objects is most important. Indeed, the values reported are for the entire detection pipeline and thus include errors accumulated during segmentation. We realise that this may not have been sufficiently clear in the text and have incorporated changes in the **Supplementary Methods** to better clarify this aspect.

Comment #3.08 -- *The values suggest a low rate of false positives, but a rather high rate of false negatives. Is it possible to determine that there is not something 'systematically different' about the false negatives that may influence downstream results?*

--- *For example, lower intensity, different morphology, too close to other cells.*

We appreciate the reviewer's insightful comments regarding biases in our machine learning model. In the revised analysis, we address the concerns by clarifying the implications of the identified biases, providing context for their impact on key findings, and suggesting avenues for future work.

For this purpose, our investigation utilized SHAP (SHapley Additive exPlanations): a concept from cooperative game theory applied in machine learning to estimate the weight of each input feature in determining a model's output³¹. By looking at the SHAP values difference between false negatives and true positives, we could identify features driving discrepancies between these, as a quantitative fairness metric to investigate model bias. The top 5 features identified are highly correlated with each other and imply the presence of three primary factors characterizing the false negatives cases, which we attribute to segmentation quality, object size and object adjacency (**Fig. S18B, S18C**):

Segmentation Quality: Poor segmentation would affect features such as roundness and Hu moments, which capture the ‘round’ shape of a micronucleus. We validated this by annotating cases of poorly segmented masks, where portions of the nucleus were incorrectly included. We observed that, under equal conditions, recall was significantly higher for correctly segmented micronuclei (**Fig. S18D**). As a result, segmentation-related artifacts can alter feature values in poorly segmented instances (**Fig. S18H**).

Object Size: Features like area-perimeter ratio and magnitude of the object long axis directly correlate with object size. We noticed that model performance decreases for extreme cases, particularly for smaller micronuclei (**Fig. S18E**). Subdividing the dataset confirmed that feature values associated with object size were significantly lower in false negatives, tracking their smaller size (**Fig. S18I**). Additionally, fluorescence intensity was dimmer in these cases, emphasizing that dim, small micronuclei are inherently challenging to detect (**Fig. S18J**), especially when the dataset is dominated by similarly-sized, small objects like debris (**Fig. S18K**). We however emphasise that challenges associated with detecting very small micronuclei in datasets crowded with small, similarly-sized objects like debris are comparable for both manual annotation and machine learning classification methods.

Object adjacency: Objects that are adjacent to the main nucleus pose a difficult challenge in terms of segmentation, compared to those that exist in isolation. The `hist_ng0_mni` feature measures the lowest bin intensity in the immediate surrounding of the micronucleus, sensing if this is adjacent to the parental nucleus or isolated (**Fig. S18L**). This difference, similarly to other segmentation quality issues can be reflected in a relatively lower recall for these micronuclei (**Fig. S18F**).

Confirmation of the role of these three factors in driving the group differences in false negatives and true positives is given by the fact that filtering the dataset based on adjacency to the nucleus, object size or segmentation quality decreases the group difference for the corresponding biased features (**Fig. S18M**).

While these factors provide important insights into model limitations, we stress that they are unlikely to affect our key findings. False negatives from poorly segmented masks are expected, but reflect technical challenges in segmentation and do not systematically impact micronucleus features central to our analysis. The lower recall for adjacent micronuclei is also connected to difficulty in segmentation of micronuclei candidates. However, micronuclei are mobile, shifting between isolated and adjacent positions, reducing the potential for consistent bias. Detection challenges near the microscope’s resolution limit, particularly for small or dim micronuclei with low signal-to-noise ratios, are also of technical nature and inherent to the imaging process. In our case, the wide dynamic range of marker fluorescence can complicate simultaneous detection of dim and bright objects, but this reflects expression levels of the transfected marker rather than biologically relevant signals. Moreover, small, dim micronuclei are difficult to annotate even for human experts and the intrinsic uncertainty of this detection might be reflected in the model.

In the revised Discussion we now address these potential biases and contextualise their likely influences, as well as offer avenues for further methods optimisation in the future. We note, for example, that future iterations of MAGIC might employ alternative segmentation algorithms such as CellPose, which could somewhat improve robustness in the context of

certain “edge cases”. In addition, expanding the dataset to include more diverse and representative examples of small, dim, or adjacent micronuclei could improve the ability of our models to generalize across such challenging instances in the future.

Figure S18 | exploring limitations in micronuclei detection.

Precision-recall curve calculated on image data from an experiment with MCF10A wild-type (WT), micronucleated cells (**A**). Difference in the average SHAP values between false negatives and true positives groups, the bar plot shows the top 5 biased features (**B**). Cross-correlation heatmap for the top 5 biased features (Spearman correlation) (**C**). Difference in recall across the segmentation quality (**D**), micronucleus area (**E**) and nucleus adjacency (**F**) factors. Recall per micronucleus area quantile bins (**G**). Value distribution for features related to segmentation quality (**H**), micronucleus size (**I**) and adjacency (**L**). Mean fluorescence intensity of small micronuclei per prediction outcome (**J**). Area distribution across prediction outcomes (**K**). Difference in average SHAP values between false negatives and true positives groups (**M**), before and after filtering out adjacent (left panel), small (central panel) or poorly segmented (right panel) micronuclei.

Comment #3.09 - *XGBoost is justified as 'performing similarly to a deep learning-based convolutional neural network'. However the details of the CNN architecture depicted in Supplementary Table 3 are unclear, and the choice seems quite unconventional. Is there a justification for using this custom architecture, rather than something more established (e.g. a ResNet, MobileNet)?*

We thank the reviewer to give us the opportunity to clarify this aspect: the CNN architecture used is ConvNeXt (<https://arxiv.org/abs/2201.03545>), which has been shown to outperform previous models like ResNet. We have amended our manuscript to clarify this.

Comment #3.10 - *Given the possibility of both segmentation and classification errors, were the 'pre-' and 'post-experiment' images used to assess the 'real-world' performance of the system, ensuring that it achieves sufficient precision and recall across all relevant conditions?*

We thank the reviewer for this valuable suggestion and agree that real-world performance metrics are essential for evaluating our models. In response to this reviewer comment, we have selected a representative experiment for MCF10A, wild-type, micronucleated cells for an in-depth assessment. We find that the precision and recall performance in real-world is essentially identical to that measured during model training (see **Fig. S18A** and response to **Comment #3.08**). We stress that this example is representative of our entire dataset, as no evidence suggests alterations in micronucleus morphology and composition across conditions, making additional in-depth analysis under alternative conditions unnecessary. We have incorporated this information into the revised manuscript, in the context of a larger methodological description about characterisation of the machine-learning model.

Comment #3.11 - *I find it quite difficult to follow the image analysis workflow, specifically around when a nucleus and micronuclei are considered as a single object and when they are considered/classified separately. I hoped that 'object_classification_example.ipynb' would resolve this, but I remain unclear. I think that the images with contours at the bottom are supposed to show nuclei with associated micronuclei, but I am not sure whether these should all be considered true positives - nor how they compare with cells that were rejected.*

We thank the reviewer for highlighting this point. We recognize that the image analysis workflow is complex, and we have thus improved the explanatory details in the Jupyter notebook, part of the *magic_tools* repository documentation. To clarify, our classification process targets candidate objects that may constitute micronuclei, rather than classifying

entire cells or nuclei. Importantly, to improve classification accuracy, we also incorporate the features of such adjacent parental objects, as each candidate object's feature vector is concatenated with that of the nearest parent (typically a nucleus). This approach helps distinguish relevant structures, especially in cases where parental objects may have highly irregular shapes, such as in mitotic or apoptotic cells, which do not represent interphase nuclei.

In our framework, micronuclei can either be isolated in the cytoplasm (case 1) or appear adjacent to a nucleus (case 2). In case 1, isolated micronuclei are easily identified as they are separated from any nucleus, and we simply select all small objects as candidates for classification. In case 2, where a micronucleus is adjacent to a nucleus, they often end up within the same object mask. Here, we employ morphological operations to approximate the nucleus's primary shape and "cut out" any unusual extensions—similar to detecting convexity defects. These candidate objects, whether isolated or adjacent to nuclei, are then pooled and passed through the classification process.

As mentioned in **Comment #4.04**, we have now provided a larger training set to test the code and we have improved the tutorial commentary, which now includes a comparison between micronuclei detected with high probability and rejected candidate micronuclei. We have ensured that the updated notebook improves the clarity of our image analysis workflow for future readers, and we thank the reviewer for the opportunity to provide clarification.

Comment #3.12 Referee #3 (Remarks on code availability):

It is commendable that the software is made available under an open-source license, and is provided with documentation.

Three example fields of view are also provided, along with several training images.

Because it is not possible for me to test the system with a real microscope, I tried to follow the instructions given under 'Testing micronuclei detection plugin' on the 'magic_automation' repository.

This was quite difficult, because important instructions are split across two repositories. It was necessary to add a Fiji update site, install an extra file to Fiji, set up a conda environment, and run a server.

However when I tried this on macOS, I encountered further errors because I needed to carry out further non-obvious steps that were only described later, such as running 'set_server_configuration.py' (which must be done after the server is running), and creating a new 'logs/images' directory.

It would be helpful to try to streamline the process further, to ensure that others can run the code to test the method more easily.

It would also help to provide more images to give a better sense of performance, and to more clearly show in the Jupyter notebooks the classification results - including true/false

positives/negatives. In the low-resolution images, particularly for 'micronuclei candidates', it is difficult to interpret what is depicted.

We thank the reviewer for their detailed feedback on the setup and documentation process. We appreciate the time they invested in testing the software and regret any frustration caused during code testing. To address the difficulties experienced, we have streamlined and consolidated the setup instructions across repositories in a single document (https://git.embl.de/cosenza/magic_automation/-/blob/main/docs/MAGIC_software_installation.md).

As mentioned in **Comment #3.11 and #4.04**, we have improved the notebook walk-throughs and provided a larger training set to test the code, which now includes a comparison between micronuclei detected with high probability and rejected candidate micronuclei.

In addition, we have now provided a unified Docker-based container (<https://git.embl.de/tweber/magic-container>) for running the full suite of MAGIC tools, including strandtools, magic_tools, and magic_automation. The container includes all required dependencies (ImageJ, JupyterLab, and FastAPI) to ensure a streamlined, reproducible setup process. We chose this approach to offer users a robust, platform-independent solution and circumvent installation and compatibility issues, which could be encountered when managing the multiple software components leveraged.

The Docker container supports three execution modes tailored to different aspects of Strand-seq and MAGIC workflows: (1) running Strand-seq analysis using strandtools, (2) testing and developing image analysis pipelines with magic_tools, and (3) executing complete MAGIC experiments via magic_automation. In the latter mode, a desktop GUI is made available through a browser interface using noVNC, enabling users to interact with ImageJ directly in their browser to configure and run MAGIC experiments. To simplify deployment, the container can be built from scratch using the provided Dockerfile, and a pre-built, ready-to-use image is now also available. We note that the container currently supports the amd64 architecture, which is due to requirements specific to running ImageJ.

We thank the reviewer again for their constructive comments. We are confident these improvements will make the software more accessible and user-friendly for the community.

Minor comments:

Comment #3.13 - *The software has been written for use with a Zeiss microscope. Would it be feasible for the software to work on different imaging systems? Do you envisage others using the software in its current form, or is the purpose rather to show proof-of-concept?*

Integration with other imaging systems largely depends on the openness of the microscope's control software, as different brands offer varying levels of flexibility and user-friendliness. This challenge is non-trivial and often requires the development of customized solutions. We will in the future continue developing and streamlining MAGIC, and expanding compatibility with imaging systems from other brands is indeed part of our future strategy.

Comment #3.14 - *The file 'example_training-set_object-classification.csv' does not appear to be a CSV.*

We thank the reviewer for noticing this. The file is a gzip compressed CSV. To clarify that, we have corrected the file format to .csv.gz.

Referee #4 (Remarks to the Author):

Summary:

The paper proposes a platform named MAGIC, which combines confocal live-cell imaging, machine learning-based image processing, and single-cell genomics for the analysis of chromosomal abnormalities (CAs). The study focuses on two non-transformed cell lines, MCF10A and RPE-1, to investigate de novo CAs and their underlying mechanisms. The platform workflow involves live-cell confocal microscopy imaging, manual annotation of cells with micronuclei, training a machine learning algorithm (XGBoost) for automatic identification of micronucleated cells, targeted photolabeling, cell sorting, and ultimately, single-cell genomic characterization of the sorted cells.

Comment #4.01 *One highlight of the paper is the integration of machine learning with targeted photolabeling and cell sorting to enable efficient, large-scale selection of cells with micronuclei. Using MAGIC, the authors analyzed up to 80,000 cells per experiment and isolated over 2,000 single cells for downstream analysis. The study finds that dicentric chromosomes are significant contributors to CA formation, leading to chromosomal gains and losses. Additionally, the paper shows that TP53 deficiency doubles the baseline rate of CA formation, suggesting that p53 mutation plays a critical role in promoting CA propagation in cancer genomics. Furthermore, the authors identified distinct classes of CAs and demonstrated that the types of CAs can be influenced by inducing double-strand breaks (DSBs) at targeted locations. Overall, the study exemplifies how interdisciplinary techniques can advance our understanding of genomic instability. Notably, the real-time feedback loop for automated cell identification and targeted microscopy illumination based on nuclear morphology enhances the scalability of single-cell analysis.*

We thank the reviewer for this thoughtful and positive summary of our study.

However, I have a few questions and comments as detailed below:

Machine learning:

Comment #4.02 *General Machine Learning Pipeline:*

According to the manuscript, the machine learning algorithm for identifying candidate cells can be generally divided into three steps: (1) semantic segmentation to separate nuclei foreground from the background, (2) instance segmentation from the nuclei mask generated in step 1 using morphological operations and watershed, and (3) object classification that uses object-level features to classify objects as normal nuclei or micronuclei. Steps 1 and 3 employ separate XGBoost classifiers. While these classical morphological operations, watershed, and intensity-based features are straightforward, they may lack robustness. They could result in fragmented segmentations, as seen in the example provided in the notebook, semantic_segmentation_example.ipynb (also see exemplary segmentation result in the attachment). If these fragmented segmentations can affect steps 2 and 3, how do they impact overall performance? The manuscript reports only precision and recall for step 3 as

0.9 and 0.5, respectively. It would be beneficial to examine the potential error rates of steps 1 and 2 as well.

We appreciate this important point raised by the reviewer. Indeed, we fully concur that the performance evaluation needs to reflect the integrated performance of the entire machine learning pipeline, rather than solely individual steps. We therefore clarify explicitly that the performance metrics presented in our manuscript represent the cumulative accuracy of the entire machine learning pipeline, including the steps 1-3 mentioned above by the reviewer. Recognizing that our original wording may have led to misinterpretation, we have revised the manuscript to ensure it is clear that the reported error rate estimates pertain to the full machine learning pipeline.

Comment #4.03 Annotation Difficulty:

The proposed pipeline requires two types of annotations: foreground/background annotation for semantic segmentation (step 1) and normal nuclei/micronuclei annotation for object classification (step 3). While annotations for step 1 seem relatively straightforward, the ones for step 3 can be challenging based on the example images provided on GitHub. I am curious about the workload and inter-expert variability for these annotations. If the annotation process is not straightforward, how much effort would users need to employ this platform effectively?

We thank the reviewer for the opportunity to clarify the annotation process. We engaged multiple experts to annotate different datasets used across the models. Annotating 200-400 fields of view typically takes a couple of days, which provides sufficient data for training models with reasonable accuracy. Regarding inter-expert variability, we proactively addressed this by ensuring thorough discussion among annotators for any ambiguous ("edge") cases. We have found this collaborative approach to be very effective, essentially preventing any substantial inter-expert variability.

With regard to the effective use of the platform, given the range of potential applications we believe it is not feasible to create a model (or a collection of models) that covers every possible use case. This guided our approach to focus on simpler, more adaptable machine learning methods, similar to those used elsewhere³⁰. These can work with a smaller training set, while still achieving an appropriate performance for the experimental needs³⁰. This approach allows users to employ MAGIC effectively with manageable annotation requirements and minimal computational resources. We now clarify these points in the revised Methods section.

Comment #4.04 Precision/Recall:

The manuscript claims that the ML framework achieves a precision of at least 90% and a recall of 50%, as shown in Figure S2. However, the object classification notebook reports a validation precision of 0.5 and a recall of 0.8 (see attachment for screenshot). Could the authors clarify this discrepancy? In the context of the MAGIC platform, where the selection of micronucleated cells for downstream genomic analysis is critical, would it be more important to prioritise high precision or high recall at this stage?

We thank the reviewer for this comment and the opportunity to clarify. The line shown in the screenshot represents only a subset of the full precision-recall table. Specifically, the keys "validation_precision_0.1" and "validation_recall_0.1" correspond to the precision and recall values calculated at a prediction probability threshold of 0.1. For reasons of practicality and

space, the repository initially included a simplified “toy dataset” intended primarily to demonstrate the code's functionality. We apologise if this has led to any confusion. To enhance clarity and provide more comprehensive insights, we have, during manuscript revision, added a precision-recall plot and included a larger, annotated dataset.

In the context of the MAGIC platform, we have intentionally prioritized high precision at the expense of recall. We emphasise that when isolating cells of interest, any experiment inherently represents a sample from a larger population. As such, collecting all available cells is neither feasible nor necessary, making lower recall acceptable. Our primary experimental requirement is ensuring the homogeneity of the selected cell population, since this is crucial for obtaining definitive and biologically meaningful results. Consequently, our methodology emphasizes precision over recall.

Comment #4.05 Model Generalizability:

I am somewhat skeptical about the robustness and generalizability of the model. The data provided in the study was tested in a single lab environment. How would the model perform on images from other labs or images acquired with different microscopy setups? The authors trained an XGBoost model for each cell line, which suggests potential challenges in model generalization across different cell lines.

We thank the reviewer for this comment. As also alluded to in our response to **Comments #4.03**, the classification models were not designed to generalise across microscopy setups or cell lines. We chose a pragmatic approach, employing models that can be quickly deployed and tailored to the specifics of the given experiments geared towards studying the origins of chromosomal instability. We also refer to our response to **Comment #3.01**, emphasising that **Reviewer #3** (an expert in biomedical imaging analysis) comments on our XGBoost as a pragmatic approach the adaptation of which to our given application is seen as suitable. We emphasise that our chosen approach aligns with the wide variety of applications that could be enabled by MAGIC, and addresses practical limitations around training data and annotation resources. We clarify this approach in the revised Methods.

Nevertheless, while outside of the scope of this current study, we agree that further robustness and generalizability of the computer vision approach utilised by the MAGIC platform would be desirable in the future. We are hence working towards further generalising the training of MAGIC, and will embrace systems other than XGBoost in the future.

Comment #4.06 Suggestions:

I recommend considering Cellpose (Stringer et al., 2021 and also Cellpose2/Cellpose3), a state-of-the-art nuclei segmentation tool, as a potential solution for nuclei instance segmentation. This could help address potential segmentation issues in steps 1 and 2.

We thank the reviewer for this recommendation. While Cellpose was not yet available at the time we developed MAGIC and initiated large-scale single-cell genomic data generation for our data-driven biological study, the computational pipelines underlying MAGIC remain sufficiently flexible to allow for the future integration of such advanced methods. We have revised the Discussion section accordingly, highlighting this point in our future outlook.

Comment #4.07 Code:

I attempted to run the code in the Magic_tool notebook, but it was unsuccessful. While installation appeared to be fine, the magic_tool import did not work on both my MacBook and Google Colab. I suggest that the authors prepare a ready-to-use Google Colab notebook for testing, which would avoid issues related to operating system compatibility during software installation.

We thank the reviewer for this comment. To address this reviewer comment, we have streamlined the setup process and developed a Docker image to resolve these issues (as also mentioned in our response to **Comment #3.12**, <https://git.embl.de/tweber/magic-container>). This provides a robust, platform-independent solution, circumventing installation and compatibility issues which could be encountered when managing the multiple software components leveraged in this manuscript.

We emphasise that while platforms like Google Colab are suitable for lightweight, notebook-based workflows, Docker offers several advantages for this use case: it guarantees full reproducibility of the computational environment across different systems, and avoids runtime limitations as well as unpredictable software changes that in our experience are often seen in Google Colab. These features make Docker a more stable solution for executing and sharing our complex workflows. This will facilitate the reproducibility of MAGIC experiments in other research groups, including execution of the entire MAGIC pipeline across diverse computational infrastructures (e.g., local machines, institutional servers, or cloud-based systems). We thank the reviewer again for their helpful suggestions with respect to making our research software more accessible and user-friendly.

Comment #4.08 *Line 190-191:*

"Utilizing FACS, we observe distinct populations that represent photolabeled cells with either dye (Fig. 1G, S1G), confirming that target cells are efficiently sorted."

Figure S1 does not include a panel G. Did the authors mean to refer to Figure S2G here?

We appreciate the reviewer's careful examination of our figures. The reference to Figure S1G was indeed a typo, and we have now corrected it to Figure S2G in the revised manuscript.

Comment #4.09 *Main Figure 2:*

In panel (C), which depicts mitotic history reconstruction for reciprocal CAs in (B), sister A has only one copy number (indicating a loss), while sister B has three copies. Could the authors also provide an illustration for the shared CA where both A and B exhibit a single copy number loss?

We thank the reviewer for these insightful suggestions. To provide further clarification: **Figures 2B,C** depict a mitotic reconstruction specifically for a reciprocal CA. The rationale for focusing on a reciprocal event is that this event allows the accurate inference of the rearrangement process occurring within the most recent cell division cycle. In contrast, a shared CA associated with a single copy-number loss indicates an event that must have occurred in an earlier mitotic cycle—likely resulting from an initial chromosomal breakage. However, since this shared CA did not arise from the most recent mitotic event, we lack sufficient evidence to reliably reconstruct the underlying rearrangement process, which is why we refrained from including an illustrative depiction of this scenario.

Referee #4 (Remarks on code availability):

Comment #4.10 *As above, although I was not able to run the code (see screenshot of errors in the attachment), I checked the main image processing and machine learning pipeline and results produced in the python notebook. But authors need to resolve the installation issue and can provide a ready-to-use google colab notebook to avoid ambiguity for the code installation.*

We appreciate the time the reviewer invested in testing the software and regret any potential issues caused by these installation issues. As mentioned in **Comments #3.12 and #4.07**, in response to the comments of this reviewer and of **Reviewer #3**, we have put work into improving documentation, and have developed a Docker container as a stable, cross-platform solution to ensure computational code reproducibility (<https://git.embl.de/tweber/magic-container>).

References

1. Agustinus, A. S. *et al.* Epigenetic dysregulation from chromosomal transit in micronuclei. *Nature* **619**, 176–183 (2023).
2. Mohr, L. *et al.* ER-directed TREX1 limits cGAS activation at micronuclei. *Mol. Cell* **81**, 724–738.e9 (2021).
3. Harding, S. M. *et al.* Mitotic progression following DNA damage enables pattern recognition within micronuclei. *Nature* **548**, 466–470 (2017).
4. Zhang, C.-Z. *et al.* Chromothripsis from DNA damage in micronuclei. *Nature* **522**, 179–184 (2015).
5. Maciejowski, J., Li, Y., Bosco, N., Campbell, P. J. & de Lange, T. Chromothripsis and kataegis induced by telomere crisis. *Cell* **163**, 1641–1654 (2015).
6. Picelli, S. *et al.* Full-length RNA-seq from single cells using Smart-seq2. *Nat. Protoc.* **9**, 171–181 (2014).
7. Sanders, A. D. *et al.* Single-cell analysis of structural variations and complex rearrangements with tri-channel processing. *Nat. Biotechnol.* **38**, 343–354 (2020).
8. Liberzon, A. *et al.* The Molecular Signatures Database (MSigDB) hallmark gene set collection. *Cell Syst.* **1**, 417–425 (2015).
9. Chen, Q., Sun, L. & Chen, Z. J. Regulation and function of the cGAS-STING pathway of cytosolic DNA sensing. *Nat. Immunol.* **17**, 1142–1149 (2016).
10. Puleo, J. & Polyak, K. The MCF10 Model of Breast Tumor Progression. *Cancer Res.* **81**, 4183–4185 (2021).
11. Jiang, X. R. *et al.* Telomerase expression in human somatic cells does not induce changes associated with a transformed phenotype. *Nat. Genet.* **21**, 111–114 (1999).
12. Falconer, E. *et al.* DNA template strand sequencing of single-cells maps genomic rearrangements at high resolution. *Nat. Methods* **9**, 1107–1112 (2012).
13. van Wietmarschen, N. & Lansdorp, P. M. Bromodeoxyuridine does not contribute to sister chromatid exchange events in normal or Bloom syndrome cells. *Nucleic Acids Res.* **44**, 6787–6793 (2016).

14. Santaguida, S., Tighe, A., D'Alise, A. M., Taylor, S. S. & Musacchio, A. Dissecting the role of MPS1 in chromosome biorientation and the spindle checkpoint through the small molecule inhibitor reversine. *J. Cell Biol.* **190**, 73–87 (2010).
15. Worsham, M. J. *et al.* High-resolution mapping of molecular events associated with immortalization, transformation, and progression to breast cancer in the MCF10 model. *Breast Cancer Res. Treat.* **96**, 177–186 (2006).
16. Mardin, B. R. *et al.* A cell-based model system links chromothripsis with hyperploidy. *Mol. Syst. Biol.* **11**, 828 (2015).
17. Klaasen, S. J. *et al.* Nuclear chromosome locations dictate segregation error frequencies. *Nature* **607**, 604–609 (2022).
18. Umbreit, N. T. *et al.* Mechanisms generating cancer genome complexity from a single cell division error. *Science* **368**, eaba0712 (2020).
19. Helmchen, F. & Denk, W. Deep tissue two-photon microscopy. *Nat. Methods* **2**, 932–940 (2005).
20. Shoshani, O. *et al.* Chromothripsis drives the evolution of gene amplification in cancer. *Nature* **591**, 137–141 (2021).
21. Verhaak, R. G. W., Bafna, V. & Mischel, P. S. Extrachromosomal oncogene amplification in tumour pathogenesis and evolution. *Nat. Rev. Cancer* **19**, 283–288 (2019).
22. Chamorro González, R. *et al.* Parallel sequencing of extrachromosomal circular DNAs and transcriptomes in single cancer cells. *Nat. Genet.* **55**, 880–890 (2023).
23. Barra, V. & Fachinetti, D. The dark side of centromeres: types, causes and consequences of structural abnormalities implicating centromeric DNA. *Nat. Commun.* **9**, 4340 (2018).
24. ICGC/TCGA Pan-Cancer Analysis of Whole Genomes Consortium. Pan-cancer analysis of whole genomes. *Nature* **578**, 82–93 (2020).
25. Mertens, F., Johansson, B. & Mitelman, F. Isochromosomes in neoplasia. *Genes Chromosomes Cancer* **10**, 221–230 (1994).
26. Ziegenhain, C. *et al.* Comparative analysis of single-cell RNA sequencing methods. *Mol.*

- Cell* **65**, 631–643.e4 (2017).
27. Adell, M. A. Y. *et al.* Adaptation to spindle assembly checkpoint inhibition through the selection of specific aneuploidies. *Genes Dev.* **37**, 171–190 (2023).
 28. Drews, R. M. *et al.* A pan-cancer compendium of chromosomal instability. *Nature* **606**, 976–983 (2022).
 29. Steele, C. D. *et al.* Signatures of copy number alterations in human cancer. *Nature* **606**, 984–991 (2022).
 30. Berg, S. *et al.* Ilastik: Interactive machine learning for (bio)image analysis. *Nat. Methods* **16**, 1226–1232 (2019).
 31. Lundberg, S. M. *et al.* From local explanations to global understanding with explainable AI for trees. *Nat. Mach. Intell.* **2**, 56–67 (2020).

Referees' comments:

Referee #1 (Remarks to the Author):

Comment #1.01 - *The authors addressed well all of my comments, including additional key experiments that expand the generalizability of their findings. I have no further comments.*

We are pleased that our response has satisfactorily addressed the referee's concerns, and we would like to thank them for their constructive comments and valuable suggestions.

Referee #2 (Remarks to the Author):

Comment #2.01 - *The authors have addressed my comments in a satisfactory manner. I have no further comments.*

We appreciate the referee's acknowledgement that our responses have satisfactorily addressed their comments. We thank the referee once more for their thoughtful review.

Referee #3 (Remarks to the Author):

Comment #3.01 - *I thank the authors for their detailed replies to my comments on the image analysis aspects of this work, and for the additional clarifications incorporated into the manuscript.*

We thank the referee for their careful review and constructive feedback in improving the manuscript.

Comment #3.02 - *While I appreciate that the segmentation strategy using XGBoost resembles the approach found in ilastik, I believe ilastik was released around 14 years ago. This was long before the widespread use of deep learning models, which dramatically improved the accuracy of nucleus segmentation - especially since the Kaggle 2018 Data Science Bowl (<https://doi.org/10.1038/s41592-019-0612-7>). Therefore I welcome the inclusion of a statement about the potential use of other methods in the manuscript.*

We acknowledge the revolutionary contribution of deep learning models in nucleus segmentation. We have edited the text of our manuscript accordingly to refer to the Kaggle 2018 Data Science Bowl.

Comment #3.03 - *I also appreciate the clarification that the deep learning architecture tested was based on ConvNeXt, that CellPose was not available at the time this work was started, and that StarDist's star-convex assumption was inappropriate here. Given its near ubiquity for bioimage analysis over some years, I think that a U-Net would have been the natural comparison - and perhaps this might be considered as a drop-in replacement for the current strategy in the future.*

We thank the reviewer for this valuable suggestion. We are indeed planning CellPose integration in future versions of MAGIC. In the meantime we acknowledge this possibility in the discussion section of our manuscript by mentioning CellPose explicitly:

"Community challenges have demonstrated the power of deep learning for robust and generalizable nucleus segmentation⁸⁸, highlighting opportunities for integrating new cell segmentation tools like CellPose⁸⁹ into MAGIC in the future to support a wider array of classification tasks."

Referee #4 (Remarks to the Author):

Many thanks to the authors for providing the revised manuscript along with detailed responses to reviewer comments. The revisions address most of my previous concerns; however, I have two remaining points for further clarification:

Comment #4.01 - Precision and Recall

It may indeed be acceptable to achieve high precision with relatively lower recall in subpopulation selection. However, I recommend assessing whether the selected subpopulation accurately represents a random and unbiased subset. It is important to verify that cells missed by the image processing pipeline do not disproportionately belong to a particular subtype, as this could significantly bias subsequent analyses.

We thank the reviewer for this valuable suggestion. To evaluate whether the detected micronuclei constitute a representative subset of the overall population, we applied UMAP-based dimensionality reduction to the full feature set. In the resulting embedding, true positives and false negatives appeared broadly interspersed, suggesting no overt morphological bias in detection (see **Figure R1** below). To quantify this, we trained a random forest classifier to distinguish between true positives and false negatives based on UMAP coordinates, yielding a ROC AUC of 0.654. Given that an AUC of 1 indicates perfect separability and 0.5 indicates no separation (i.e. “full randomness”), this result points to at most a very weak detection bias.

We would also like to highlight that we previously undertook a comprehensive characterization of the technical limitations inherent in our microscopy-based micronuclei detection approach. This was summarized in the **Supplementary Information** and **Figure S18**, as part of our response to Comment #3.08 from Reviewer #3 in an earlier review round. In that analysis, we employed SHAP (SHapley Additive exPlanations) values to assess how individual input features influenced the model’s output probability¹. A comparative analysis between true positives and false negatives identified three primary contributors to false negative predictions: reduced segmentation quality, small object size, and close proximity of the micronucleus to the parental nucleus. Although these findings shed light on detection challenges, we wish to underscore that they do not affect the core results of our study, as these limitations appear technical in nature and not indicative of biologically distinct subtypes of micronuclei.

These points are addressed in the Discussion section of our paper, where we also outline opportunities for further methodological refinement. Finally, we opted not to incorporate **Figure R1** in the manuscript, considering that **Figure S18** is already extensive, but would be open to providing it as an additional Supplementary Figure at the editor’s or reviewer’s discretion.

Fig. R1 | Dimensionality reduction on computed features

Computed features used for micronuclei detection were scaled and a two-dimensional UMAP embedding was then generated using the scaled feature matrix to visualize the distribution of true positives (TP) and false negatives (FN).

Comment #4.02 - Code and Documentation

We (my PhD co-reviewer and I) successfully tested the provided Docker container on a Windows system. However, the pre-built version failed when tested on a macOS system by another colleague. If compatibility issues exist, it is crucial to clearly mention system dependencies or limitations on your website. Additionally, providing detailed instructions tailored to users with different operating systems would significantly enhance accessibility.

We thank the reviewers for bringing this compatibility matter to our attention. We have addressed macOS compatibility, and enhanced our Docker Compose installation documentation. The Docker container is built for linux/amd64 architecture due to Fiji/ImageJ limitations on ARM64 systems. Comprehensive testing of the container was done on Windows (AMD64), Linux (AMD64 & ARM64) & MacOS (ARM64) systems. Our documentation has been updated to clearly specify platform requirements and provide detailed Docker installation prerequisites to ensure proper deployment across all supported systems.

Comment #4.03 - Furthermore, the GitLab repository (magic_automation) currently states: "A pre-built Docker container (<https://git.embl.de/tweber/magic-container>) is available for running the full suite of MAGIC tools, including strandtools, magic_tools, and magic_automation." However, the Docker container we tested contains only strandtools (see

attached screenshot). Clarifying or correcting this description would help users accurately understand the available tools.

We appreciate this comment. For clarification, the fact that the container "contains only strandtools" reflects an intentional design choice rather than a functional limitation, as we have envisaged different deployment modes depending on the intended use case.

In particular, the Magic Container can operate in three distinct execution modes:

1. MODE=strandtools > JupyterLab interface from /app/strandtools directory
2. MODE=magic_tools > JupyterLab interface from /app/magic_tools + FastAPI server + dashboard
3. MODE=magic_automation > VNC desktop with Fiji/ImageJ

Each container instance executes a single mode per deployment. The screenshot displaying only strandtools content indicates the container was operating in strandtools mode, which serves as the default configuration. All three tool suites remain embedded within the container image and become accessible through their respective execution modes. In case concurrent access to multiple tools would be needed, we recommend deploying our provided docker-compose configuration, which instantiates separate service containers for each operational mode. Complete implementation examples are detailed in our documentation to facilitate proper multi-service deployment.

Referee #4 (Remarks on code availability):

See above comments to the authors

We once again thank the referee for their constructive review.

Reference:

1. Lundberg, S. M. et al. From local explanations to global understanding with explainable AI for trees. *Nat. Mach. Intell.* 2, 56–67 (2020)

Exemplary fragmented segmentation.

```
'validation_precision_0.1': 0.5, 'validation_recall_0.1': 0.8148148148148148,
```

Precision/recall of the validation set in the notebook.

Error after installing Magic tool

```
-----  
KeyError                                Traceback (most recent call last)  
Cell In[23], line 16  
    14 from magic_tools.analysis.utils import reorder_axis  
    15 import magic_tools.analysis.image as image  
 16 import magic_tools.analysis.features as features  
    17 from magic_tools.ml.classify import Training  
    18 import magic_tools.ml.classify as classify  
  
File c:\users\yu\magic_tools\magic_tools\analysis\features.py:21  
    19 from .utils import clip_bbox, resize_bbox, roi_resize  
    20 from .curvature import smoother, resample_tau, curvature_angles, scale_coordinates  
 21 from ..ml.classify import Classifier  
    24 #####  
    25 #   W O R K E R   #  
    26 #####  
    28 class Worker():  
  
File c:\users\yu\magic_tools\magic_tools\ml\classify.py:13  
    10 import os  
    11 from itertools import product  
 13 if 'jupyter' in os.environ['_']:  
    14     import shap  
    15     shap.initjs()  
  
File <frozen os>:679, in __getitem__(self, key)  
KeyError: '_'
```